# Accelerating Value Iteration with Anchoring

**Jongmin Lee**[1]     **Ernest K. Ryu**[1,2]

[1]Department of Mathematical Science, Seoul National University
[2]Interdisciplinary Program in Artificial Intelligence, Seoul National University

## Abstract

Value Iteration (VI) is foundational to the theory and practice of modern reinforcement learning, and it is known to converge at a $\mathcal{O}(\gamma^k)$-rate, where $\gamma$ is the discount factor. Surprisingly, however, the optimal rate in terms of Bellman error for the VI setup was not known, and finding a general acceleration mechanism has been an open problem. In this paper, we present the first accelerated VI for both the Bellman consistency and optimality operators. Our method, called Anc-VI, is based on an *anchoring* mechanism (distinct from Nesterov's acceleration), and it reduces the Bellman error faster than standard VI. In particular, Anc-VI exhibits a $\mathcal{O}(1/k)$-rate for $\gamma \approx 1$ or even $\gamma = 1$, while standard VI has rate $\mathcal{O}(1)$ for $\gamma \geq 1 - 1/k$, where $k$ is the iteration count. We also provide a complexity lower bound matching the upper bound up to a constant factor of $4$, thereby establishing optimality of the accelerated rate of Anc-VI. Finally, we show that the anchoring mechanism provides the same benefit in the approximate VI and Gauss–Seidel VI setups as well.

## 1  Introduction

Value Iteration (VI) is foundational to the theory and practice of modern dynamic programming (DP) and reinforcement learning (RL). It is well known that when a discount factor $\gamma < 1$ is used, (exact) VI is a contractive iteration in the $\|\cdot\|_\infty$-norm and therefore converges. The progress of VI is measured by the Bellman error in practice (as the distance to the fixed point is not computable), and much prior work has been dedicated to analyzing the rates of convergence of VI and its variants.

Surprisingly, however, the optimal rate in terms of Bellman error for the VI setup was not known, and finding a general acceleration mechanism has been an open problem. The classical $\mathcal{O}(\gamma^k)$-rate of VI is inadequate as many practical setups use $\gamma \approx 1$ or $\gamma = 1$ for the discount factor. (Not to mention that VI may not converge when $\gamma = 1$.) Moreover, most prior works on accelerating VI focused on the Bellman consistency operator (policy evaluation) as its linearity allows eigenvalue analyses, but the Bellman optimality operator (control) is the more relevant object in modern RL.

**Contribution.**  In this paper, we present the first accelerated VI for both the Bellman consistency and optimality operators. Our method, called Anc-VI, is based on an "anchoring" mechanism (distinct from Nesterov's acceleration), and it reduces the Bellman error faster than standard VI. In particular, Anc-VI exhibits a $\mathcal{O}(1/k)$-rate for $\gamma \approx 1$ or even $\gamma = 1$, while standard VI has rate $\mathcal{O}(1)$ for $\gamma \geq 1 - 1/k$, where $k$ is the iteration count. We also provide a complexity lower bound matching the upper bound up to a constant factor of $4$, thereby establishing optimality of the accelerated rate of Anc-VI. Finally, we show that the anchoring mechanism provides the same benefit in the approximate VI and Gauss–Seidel VI setups as well.

37th Conference on Neural Information Processing Systems (NeurIPS 2023).

## 1.1 Notations and preliminaries

We quickly review basic definitions and concepts of Markov decision processes (MDP) and reinforcement learning (RL). For further details, refer to standard references such as [69, 84, 81].

**Markov Decision Process.** Let $\mathcal{M}(\mathcal{X})$ be the space of probability distributions over $\mathcal{X}$. Write $(\mathcal{S}, \mathcal{A}, P, r, \gamma)$ to denote the MDP with state space $\mathcal{S}$, action space $\mathcal{A}$, transition probability $P\colon \mathcal{S} \times \mathcal{A} \to \mathcal{M}(\mathcal{S})$, reward $r\colon \mathcal{S} \times \mathcal{A} \to \mathbb{R}$, and discount factor $\gamma \in (0, 1]$. Denote $\pi\colon \mathcal{S} \to \mathcal{M}(\mathcal{A})$ for a policy, $V^\pi(s) = \mathbb{E}_\pi[\sum_{t=0}^\infty \gamma^t r(s_t, a_t) \,|\, s_0 = s]$ and $Q^\pi(s, a) = \mathbb{E}_\pi[\sum_{t=0}^\infty \gamma^t r(s_t, a_t) \,|\, s_0 = s, a_0 = a]$ for $V$- and $Q$-value functions, where $\mathbb{E}_\pi$ denotes the expected value over all trajectories $(s_0, a_0, s_1, a_1, \dots)$ induced by $P$ and $\pi$. We say $V^\star$ and $Q^\star$ are optimal $V$- and $Q$- value functions if $V^\star = \sup_\pi V^\pi$ and $Q^\star = \sup_\pi Q^\pi$. We say $\pi_V^\star$ and $\pi_Q^\star$ are optimal policies if $\pi_V^\star = \operatorname{argmax}_\pi V^\pi$ and $\pi_Q^\star = \operatorname{argmax}_\pi Q^\pi$. (If argmax is not unique, break ties arbitrarily.)

**Value Iteration.** Let $\mathcal{F}(\mathcal{X})$ denote the space of bounded measurable real-valued functions over $\mathcal{X}$. With the given MDP $(\mathcal{S}, \mathcal{A}, P, r, \gamma)$, for $V \in \mathcal{F}(\mathcal{S})$ and $Q \in \mathcal{F}(\mathcal{S} \times \mathcal{A})$, define the Bellman consistency operators $T^\pi$ as

$$T^\pi V(s) = \mathbb{E}_{a \sim \pi(\cdot \,|\, s), s' \sim P(\cdot \,|\, s, a)} \left[ r(s, a) + \gamma V(s') \right],$$

$$T^\pi Q(s, a) = r(s, a) + \gamma \mathbb{E}_{s' \sim P(\cdot \,|\, s, a), a' \sim \pi(\cdot \,|\, s')} \left[ Q(s', a') \right]$$

for all $s \in \mathcal{S}, a \in \mathcal{A}$, and the Bellman optimality operators $T^\star$ as

$$T^\star V(s) = \sup_{a \in \mathcal{A}} \left\{ r(s, a) + \gamma \mathbb{E}_{s' \sim P(\cdot \,|\, s, a)} \left[ V(s') \right] \right\},$$

$$T^\star Q(s, a) = r(s, a) + \gamma \mathbb{E}_{s' \sim P(\cdot \,|\, s, a)} \left[ \sup_{a' \in \mathcal{A}} Q(s', a') \right]$$

for all $s \in \mathcal{S}, a \in \mathcal{A}$. For notational conciseness, we write $T^\pi V = r^\pi + \gamma \mathcal{P}^\pi V$ and $T^\pi Q = r + \gamma \mathcal{P}^\pi Q$, where $r^\pi(s) = \mathbb{E}_{a \sim \pi(\cdot \,|\, s)}[r(s, a)]$ is the reward induced by policy $\pi$ and $\mathcal{P}^\pi(s)$ and $\mathcal{P}^\pi(s, a)$ defined as

$$\mathcal{P}^\pi(s \to s') = \operatorname{Prob}(s \to s' \,|\, a \sim \pi(\cdot \,|\, s), s' \sim P(\cdot \,|\, s, a))$$

$$\mathcal{P}^\pi((s, a) \to (s', a')) = \operatorname{Prob}((s, a) \to (s', a') \,|\, s' \sim P(\cdot \,|\, s, a), a' \sim \pi(\cdot \,|\, s')),$$

are the transition probabilities induced by policy $\pi$. We define VI for Bellman consistency and optimality operators as

$$V^{k+1} = T^\pi V^k, \quad Q^{k+1} = T^\pi Q^k, \quad V^{k+1} = T^\star V^k, \quad Q^{k+1} = T^\star Q^k \qquad \text{for } k = 0, 1, \dots,$$

where $V^0, Q^0$ are initial points. VI for control, after executing $K$ iterations, returns the near-optimal policy $\pi_K$ as a greedy policy satisfying

$$T^{\pi_K} V^K = T^\star V^K, \quad T^{\pi_K} Q^K = T^\star Q^K.$$

For $\gamma < 1$, both Bellman consistency and optimality operators are contractions, and, by Banach's fixed-point theorem [5], the VIs converge to the unique fixed points $V^\pi$, $Q^\pi$, $V^\star$, and $Q^\star$ with $\mathcal{O}(\gamma^k)$-rate. For notational unity, we use the symbol $U$ when both $V$ and $Q$ can be used. Since $\left\| TU^k - U^k \right\|_\infty \le \left\| TU^k - U^\star \right\|_\infty + \left\| U^k - U^\star \right\|_\infty \le (1 + \gamma) \left\| U^k - U^\star \right\|_\infty$, VI exhibits the rate on the Bellman error:

$$\left\| TU^k - U^k \right\|_\infty \le (1 + \gamma)\gamma^k \left\| U^0 - U^\star \right\|_\infty \qquad \text{for } k = 0, 1, \dots, \tag{1}$$

where $T$ is Bellman consistency or optimality operator, $U^0$ is a starting point, and $U^\star$ is fixed point of $T$. We say $V \le V'$ or $Q \le Q'$ if $V(s) \le V'(s)$ or $Q(s, a) \le Q'(s, a)$ for all $s \in \mathcal{S}$ and $a \in \mathcal{A}$, respectively.

**Fixed-point iterations.** Given an operator $T$, we say $x^\star$ is fixed point if $Tx^\star = x^\star$. Since Banach [5], the standard fixed-point iteration

$$x^{k+1} = Tx^k \qquad \text{for } k = 0, 1, \dots$$

has been commonly used to find fixed points. Note that VI for policy evaluation and control are fixed-point iterations with Bellman consistency and optimality operators. In this work, we also consider the Halpern iteration

$$x^{k+1} = \beta_{k+1} x^0 + (1 - \beta_{k+1}) Tx^k \qquad \text{for } k = 0, 1, \dots,$$

where $x^0$ is an initial point and $\{\beta_k\}_{k \in \mathbf{N}} \in (0, 1)$.

## 1.2 Prior works

**Value Iteration.** Value iteration (VI) was first introduced in the DP literature [8] for finding optimal value function, and its variant approximate VI [11, 30, 56, 32, 19, 90, 81] considers approximate evaluations of the Bellman optimality operator. In RL, VI and approximate VI have served as the basis of RL algorithms such as fitted value iteration [29, 57, 52, 87, 50, 36] and temporal difference learning [80, 89, 41, 94, 54]. There is a line of research that emulates VI by learning a model of the MDP dynamics [85, 83, 62] and applying a modified Bellman operator [7, 33]. Asynchronous VI, another variation of VI updating the coordinate of value function in asynchronous manner, has also been studied in both RL and DP literature [11, 9, 88, 100].

**Fixed-point iterations.** The Banach fixed-point theorem [5] establishes the convergence of the standard fixed-point iteration with a contractive operator. The Halpern iteration [39] converges for *nonexpansive* operators on Hilbert spaces [96] and uniformly smooth Banach spaces [70, 97]. (To clarify, the $\| \cdot \|_\infty$-norm in $\mathbb{R}^n$ is not uniformly smooth.)

The fixed-point residual $\|Tx_k - x_k\|$ is a commonly used error measure for fixed-point problems. In general normed spaces, the Halpern iteration was shown to exhibit $\mathcal{O}(1/\log(k))$-rate for (nonlinear) nonexpansive operators [48] and $\mathcal{O}(1/k)$-rate for linear nonexpansive operators [17] on the fixed-point residual. In Hilbert spaces, [72] first established a $\mathcal{O}(1/k)$-rate for the Halpern iteration and the constant was later improved by [49, 43]. For contractive operators, [65] proved exact optimality of Halpern iteration through an exact matching complexity lower bound.

**Acceleration.** Since Nesterov's seminal work [61], there has been a large body of research on acceleration in convex minimization. Gradient descent [15] can be accelerated to efficiently reduce function value and squared gradient magnitude for smooth convex minimization problems [61, 44, 45, 46, 102, 21, 60] and smooth strongly convex minimization problems [59, 91, 64, 86, 73]. Motivated by Nesterov acceleration, inertial fixed-point iterations [51, 22, 75, 70, 42] have also been suggested to accelerate fixed-point iterations. Anderson acceleration [2], another acceleration scheme for fixed-point iterations, has recently been studied with interest [6, 74, 93, 101].

In DP and RL, prioritized sweeping [55] is a well-known method that changes the order of updates to accelerate convergence, and several variants [68, 53, 95, 3, 18] have been proposed. Speedy Q-learning [4] modifies the update rule of Q-learning and uses aggressive learning rates for acceleration. Recently, there has been a line of research that applies acceleration techniques of other areas to VI: [34, 79, 28, 67, 27, 76] uses Anderson acceleration of fixed-point iterations, [92, 37, 38, 12, 1] uses Nesterov acceleration of convex optimization, and [31] uses ideas inspired by PID controllers in control theory. Among those works, [37, 38, 1] applied Nesterov acceleration to obtain theoretically accelerated convergence rates, but those analyses require certain reversibility conditions or restrictions on eigenvalues of the transition probability induced by the policy.

The *anchor acceleration*, a new acceleration mechanism distinct from Nesterov's, lately gained attention in convex optimization and fixed-point theory. The anchoring mechanism, which retracts iterates towards the initial point, has been used to accelerate algorithms for minimax optimization and fixed-point problems [71, 47, 98, 65, 43, 20, 99, 78], and we focus on it in this paper.

**Complexity lower bound.** With the information-based complexity analysis [58], complexity lower bound on first-order methods for convex minimization problem has been thoroughly studied [59, 23, 25, 13, 14, 24]. If a complexity lower bound matches an algorithm's convergence rate, it establishes optimality of the algorithm [58, 44, 73, 86, 26, 65]. In fixed-point problems, [16] established $\Omega(1/k^{1-\sqrt{2/q}})$ lower bound on distance to solution for Halpern iteration with a nonexpansive operator in $q$-uniformly smooth Banach spaces. In [17], a $\Omega(1/k)$ lower bound on the fixed-point residual for the general Mann iteration with a nonexpansive linear operator, which includes standard fixed-point iteration and Halpern iterations, in the $\ell^\infty$-space was provided. In Hilbert spaces, [65] showed exact complexity lower bound on fixed-point residual for deterministic fixed-point iterations with $\gamma$-contractive and nonexpansive operators. Finally, [37] provided lower bound on distance to optimal value function for fixed-point iterations satisfying span condition with Bellman consistency and optimality operators and we discussed this lower bound in section 4.

## 2 Anchored Value Iteration

Let $T$ be a $\gamma$-contractive (in the $\|\cdot\|_\infty$-norm) Bellman consistency or optimality operator. The *Anchored Value Iteration* (Anc-VI) is

$$U^k = \beta_k U^0 + (1 - \beta_k) T U^{k-1} \qquad \text{(Anc-VI)}$$

for $k = 1, 2, \ldots$, where $\beta_k = 1/(\sum_{i=0}^k \gamma^{-2i})$ and $U^0$ is an initial point. In this section, we present accelerated convergence rates of Anc-VI for *both* Bellman consistency and optimality operators for both $V$- and $Q$-value iterations. For the control setup, where the Bellman optimality operator is used, Anc-VI returns the near-optimal policy $\pi_K$ as a greedy policy satisfying $T^{\pi_K} U^K = T^\star U^K$ after executing $K$ iterations.

Notably, Anc-VI obtains the next iterate as a convex combination between the output of $T$ and the starting point $U^0$. We call the $\beta_k U_0$ term the *anchor term* since, loosely speaking, it serves to pull the iterates toward the starting point $U_0$. The strength of the anchor mechanism diminishes as the iteration progresses since $\beta_k$ is a decreasing sequence.

The anchor mechanism was introduced [39, 72, 49, 65, 17, 48] for general nonexpansive operators and $\|\cdot\|_2$-nonexpansive and contractive operators. The optimal method for $\|\cdot\|_2$-nonexpansive and contractive operators in [65] shares the same coefficients with Anc-VI, and convergence results for general nonexapnsive operators in [17, 48] are applicable to Anc-VI for nonexpansive Bellman optimality and consistency operators. While our anchor mechanism does bear a formal resemblance to those of prior works, our convergence rates and point convergence are neither a direct application nor a direct adaptation of the prior convergence analyses. The prior analyses for $\|\cdot\|_2$-nonexpansive and contractive operators do not apply to Bellman operators, and prior analyses for general nonexpansive operators have slower rates and do not provide point convergence while our Theorem 3 does. Our analyses specifically utilize the structure of Bellman operators to obtain the faster rates and point convergence.

The accelerated rate of Anc-VI for the Bellman *optimality* operator is more technically challenging and is, in our view, the stronger contribution. However, we start by presenting the result for the Bellman *consistency* operator because it is commonly studied in the prior RL theory literature on accelerating value iteration [37, 38, 1, 31] and because the analysis in the Bellman consistency setup will serve as a good conceptual stepping stone towards the analysis in the Bellman optimality setup.

### 2.1 Accelerated rate for Bellman consistency operator

First, for general state-action spaces, we present the accelerated convergence rate of Anc-VI for the Bellman consistency operator.

**Theorem 1.** *Let $0 < \gamma < 1$ be the discount factor and $\pi$ be a policy. Let $T^\pi$ be the Bellman consistency operator for $V$ or $Q$. Then, Anc-VI exhibits the rate*

$$\left\| T^\pi U^k - U^k \right\|_\infty \leq \frac{\left(\gamma^{-1} - \gamma\right)\left(1 + 2\gamma - \gamma^{k+1}\right)}{\left(\gamma^{k+1}\right)^{-1} - \gamma^{k+1}} \left\| U^0 - U^\pi \right\|_\infty$$

$$= \left( \frac{2}{k+1} + \frac{k-1}{k+1}\epsilon + O(\epsilon^2) \right) \left\| U^0 - U^\pi \right\|_\infty \qquad \text{for } k = 0, 1, \ldots,$$

*where $\epsilon = 1 - \gamma$ and the big-$\mathcal{O}$ notation considers the limit $\epsilon \to 0$. If, furthermore, $U^0 \leq T^\pi U^0$ or $U^0 \geq T^\pi U^0$, then Anc-VI exhibits the rate*

$$\left\| T^\pi U^k - U^k \right\|_\infty \leq \frac{\left(\gamma^{-1} - \gamma\right)\left(1 + \gamma - \gamma^{k+1}\right)}{\left(\gamma^{k+1}\right)^{-1} - \gamma^{k+1}} \left\| U^0 - U^\pi \right\|_\infty$$

$$= \left( \frac{1}{k+1} + \frac{k}{k+1}\epsilon + O(\epsilon^2) \right) \left\| U^0 - U^\pi \right\|_\infty \qquad \text{for } k = 0, 1, \ldots.$$

If $\gamma \geq \frac{1}{2}$, both rates of Theorem 1 are strictly faster than the standard rate (1) of VI, since

$$\frac{\left(\gamma^{-1} - \gamma\right)\left(1 + 2\gamma - \gamma^{k+1}\right)}{\left(\gamma^{k+1}\right)^{-1} - \gamma^{k+1}} = \gamma^k \frac{\left(1 - \gamma^2\right)\left(1 + 2\gamma - \gamma^{k+1}\right)}{\left(1 - \gamma^{2k+2}\right)} < \gamma^k(1 + \gamma).$$

The second rate of Theorem 1, which has the additional requirement, is faster than the standard rate (1) of VI for all $0 < \gamma < 1$. Interestingly, in the $\gamma \approx 1$ regime, Anc-VI achieves $\mathcal{O}(1/k)$-rate while VI has a $\mathcal{O}(1)$-rate. We briefly note that the condition $U^0 \leq TU^0$ and $U^0 \geq TU^0$ have been used in analyses of variants of VI [69, Theorem 6.3.11], [77, p.3].

In the following, we briefly outline the proof of Theorem 1 while deferring the full description to Appendix B. In the outline, we highlight a particular step, labeled ▲, that crucially relies on the linearity of the Bellman consistency operator. In the analysis for the Bellman optimality operator of Theorem 2, resolving the ▲ step despite the nonlinearity is the key technical challenge.

*Proof outline of Theorem 1.* Recall that we can write Bellman consistency operator as $T^\pi V = r^\pi + \gamma \mathcal{P}^\pi V$ and $T^\pi Q = r + \gamma \mathcal{P}^\pi Q$. Since $T^\pi$ is a linear operator[1], we get

$$
\begin{aligned}
T^\pi U^k - U^k &= T^\pi U^k - (1 - \beta_k) T^\pi U^{k-1} - \beta_k T^\pi U^\pi - \beta_k (U^0 - U^\pi) \\
&\stackrel{\blacktriangle}{=} \gamma \mathcal{P}^\pi (U^k - (1 - \beta_k) U^{k-1} - \beta_k U^\pi) - \beta_k (U^0 - U^\pi) \\
&= \gamma \mathcal{P}^\pi (\beta_k (U^0 - U^\pi) + (1 - \beta_k)(T^\pi U^{k-1} - U^{k-1})) - \beta_k (U^0 - U^\pi) \\
&= \sum_{i=1}^{k} \left[ (\beta_i - \beta_{i-1}(1 - \beta_i)) \left( \Pi_{j=i+1}^{k}(1 - \beta_j) \right) (\gamma \mathcal{P}^\pi)^{k-i+1} (U^0 - U^\pi) \right] \\
&\quad - \beta_k (U^0 - U^\pi) + \left( \Pi_{j=1}^{k}(1 - \beta_j) \right) (\gamma \mathcal{P}^\pi)^{k+1} (U^0 - U^\pi),
\end{aligned}
$$

where the first equality follows from the definition of Anc-VI and the property of fixed point, while the last equality follows from induction. Taking the $\|\cdot\|_\infty$-norm of both sides, we conclude

$$
\left\| T^\pi U^k - U^k \right\|_\infty \leq \frac{\left( \gamma^{-1} - \gamma \right) \left( 1 + 2\gamma - \gamma^{k+1} \right)}{(\gamma^{k+1})^{-1} - \gamma^{k+1}} \left\| U^0 - U^\pi \right\|_\infty.
$$

$\square$

## 2.2 Accelerated rate for Bellman optimality operator

We now present the accelerated convergence rate of Anc-VI for the Bellman optimality operator.

Our analysis uses what we call the *Bellman anti-optimality operator*, defined as

$$
\hat{T}^\star V(s) = \inf_{a \in \mathcal{A}} \left\{ r(s, a) + \gamma \mathbb{E}_{s' \sim P(\cdot \,|\, s, a)} [V(s')] \right\}
$$

$$
\hat{T}^\star Q(s, a) = r(s, a) + \gamma \mathbb{E}_{s' \sim P(\cdot \,|\, s, a)} \left[ \inf_{a' \in \mathcal{A}} Q(s', a') \right],
$$

for all $s \in \mathcal{S}$ and $a \in \mathcal{A}$. (The sup is replaced with a inf.) When $0 < \gamma < 1$, the Bellman anti-optimality operator is $\gamma$-contractive and has a unique fixed point $\hat{U}^\star$ by the exact same arguments that establish $\gamma$-contractiveness of the standard Bellman optimality operator.

**Theorem 2.** *Let $0 < \gamma < 1$ be the discount factor. Let $T^\star$ and $\hat{T}^\star$ respectively be the Bellman optimality and anti-optimality operators for $V$ or $Q$. Let $U^\star$ and $\hat{U}^\star$ respectively be the fixed points of $T^\star$ and $\hat{T}^\star$. Then, Anc-VI exhibits the rate*

$$
\left\| T^\star U^k - U^k \right\|_\infty \leq \frac{\left( \gamma^{-1} - \gamma \right) \left( 1 + 2\gamma - \gamma^{k+1} \right)}{(\gamma^{k+1})^{-1} - \gamma^{k+1}} \max \left\{ \left\| U^0 - U^\star \right\|_\infty, \left\| U^0 - \hat{U}^\star \right\|_\infty \right\}
$$

*for $k = 0, 1, \dots$. If, furthermore, $U^0 \leq T^\star U^0$ or $U^0 \geq T^\star U^0$, then Anc-VI exhibits the rate*

$$
\left\| T^\star U^k - U^k \right\|_\infty \leq \frac{\left( \gamma^{-1} - \gamma \right) \left( 1 + \gamma - \gamma^{k+1} \right)}{(\gamma^{k+1})^{-1} - \gamma^{k+1}} \left\| U^0 - U^\star \right\|_\infty \quad \text{if } U^0 \leq T^\star U^0
$$

$$
\left\| T^\star U^k - U^k \right\|_\infty \leq \frac{\left( \gamma^{-1} - \gamma \right) \left( 1 + \gamma - \gamma^{k+1} \right)}{(\gamma^{k+1})^{-1} - \gamma^{k+1}} \left\| U^0 - \hat{U}^\star \right\|_\infty \quad \text{if } U^0 \geq T^\star U^0
$$

*for $k = 0, 1, \dots$.*

---

[1] Arguably, $T^\pi$ is affine, not linear, but we follow the convention of [69] say $T^\pi$ is linear.

Anc-VI with the Bellman optimality operator exhibits the same accelerated convergence rate as Anc-VI with the Bellman consistency operator. As in Theorem 1, the rate of Theorem 2 also becomes $\mathcal{O}(1/k)$ when $\gamma \approx 1$, while VI has a $\mathcal{O}(1)$-rate.

*Proof outline of Theorem 2.* The key technical challenge of the proof comes from the fact that the Bellman optimality operator is non-linear. Similar to the Bellman consistency operator case, we have

$$
\begin{aligned}
T^\star U^k - U^k &= T^\star U^k - (1 - \beta_k) T^\star U^{k-1} - \beta_k T^\star U^\star - \beta_k (U^0 - U^\star) \\
&\stackrel{\blacktriangle}{\leq} \gamma \mathcal{P}^{\pi_k} \left( U^k - (1 - \beta_k) U^{k-1} - \beta_k U^\star \right) - \beta_k (U^0 - U^\star) \\
&= \gamma \mathcal{P}^{\pi_k} (\beta_k \left( U^0 - U^\star \right) + (1 - \beta_k)(T^\star U^{k-1} - U^{k-1})) - \beta_k (U^0 - U^\star) \\
&\leq \sum_{i=1}^{k} \left[ (\beta_i - \beta_{i-1}(1 - \beta_i)) \left( \Pi_{j=i+1}^{k} (1 - \beta_j) \right) \left( \Pi_{l=k}^{i} \gamma \mathcal{P}^{\pi_l} \right) (U^0 - U^\star) \right] \\
&\quad - \beta_k (U^0 - U^\star) + \left( \Pi_{j=1}^{k} (1 - \beta_j) \right) \left( \Pi_{l=k}^{0} \gamma \mathcal{P}^{\pi_l} \right) (U^0 - U^\star),
\end{aligned}
$$

where $\pi_k$ is the greedy policy satisfying $T^{\pi_k} U^k = T^\star U^k$, we define $\Pi_{l=k}^{i} \gamma \mathcal{P}^{\pi_l} = \gamma \mathcal{P}^{\pi_k} \gamma \mathcal{P}^{\pi_{k-1}} \cdots \gamma \mathcal{P}^{\pi_i}$, and last inequality follows by induction and monotonicity of Bellman optimality operator. The key step $\blacktriangle$ uses greedy policies $\{\pi_l\}_{l=0,1,\dots,k}$, which are well defined when the action space is finite. When the action space is infinite, greedy policies may not exist, so we use the Hahn–Banach extension theorem to overcome this technicality. The full argument is provided in Appendix B.

To lower bound $T^\star U^k - U^k$, we use a similar line of reasoning with the Bellman anti-optimality operator. Combining the upper and lower bounds of $T^\star U^k - U^k$, we conclude the accelerated rate of Theorem 2. $\qquad\square$

For $\gamma < 1$, the rates of Theorems 1 and 2 can be translated to a bound on the distance to solution:

$$
\left\| U^k - U^\star \right\|_\infty \leq \gamma^k \frac{(1 + \gamma) \left( 1 + 2\gamma - \gamma^{k+1} \right)}{(1 - \gamma^{2k+2})} \left\| U^0 - U^\star \right\|_\infty
$$

for $k = 1, 2, \dots$. This $O(\gamma^k)$ rate is worse than the rate of (classical) VI by a constant factor. Therefore, Anc-VI is better than VI in terms of the Bellman error, but it is not better than VI in terms of distance to solution.

## 3 Convergence when $\gamma = 1$

Undiscounted MDPs are not commonly studied in the DP and RL theory literature due to the following difficulties: Bellman consistency and optimality operators may not have fixed points, VI is a nonexpansive (not contractive) fixed-point iteration and may not convergence to a fixed point even if one exist, and the interpretation of a fixed point as the (optimal) value function becomes unclear when the fixed point is not unique. However, many modern deep RL setups actually do not use discounting,[2] and this empirical practice makes the theoretical analysis with $\gamma = 1$ relevant.

In this section, we show that Anc-VI converges to fixed points of the Bellman consistency and optimality operators of undiscounted MDPs. While a full treatment of undiscounted MDPs is beyond the scope of this paper, we show that fixed points, if one exists, can be found, and we therefore argue that the inability to find fixed points should not be considered an obstacle in studying the $\gamma = 1$ setup.

We first state our convergence result for finite state-action spaces.

**Theorem 3.** *Let $\gamma = 1$. Let $T \colon \mathbb{R}^n \to \mathbb{R}^n$ be the nonexpansive Bellman consistency or optimality operator for $V$ or $Q$. Assume a fixed point exists (not necessarily unique). If, $U^0 \leq TU^0$, then Anc-VI exhibits the rate*

$$
\left\| TU^k - U^k \right\|_\infty \leq \frac{1}{k+1} \left\| U^0 - U^\star \right\|_\infty \qquad \text{for } k = 0, 1, \dots.
$$

---

[2] As a specific example, the classical policy gradient theorem [82] calls for the use of $\nabla J(\theta) = \mathbb{E} \left[ \sum_{t=0}^{\infty} \gamma^t \nabla_\theta \log \pi_\theta(a_t \mid s_t) Q_\gamma^\phi(s_t, a_t) \right]$, but many modern deep policy gradient methods use $\gamma = 1$ in the first instance of $\gamma$ (so $\gamma^t = 1$) while using $\gamma < 1$ in $Q_\gamma^\phi(s_t, a_t)$ [63].

*for any fixed point $U^\star$ satisfying $U^0 \leq U^\star$. Furthermore, $U^k \to U^\infty$ for some fixed point $U^\infty$.*

If rewards are nonnegative, then the condition $U^0 \leq TU^0$ is satisfied with $U^0 = 0$. So, under this mild condition, Anc-VI with $\gamma = 1$ converges with $\mathcal{O}(1/k)$-rate on the Bellman error. To clarify, the convergence $U^k \to U^\infty$ has no rate, i.e., $\|U^k - U^\infty\|_\infty = o(1)$, while $\|TU^k - U^k\|_\infty = \mathcal{O}(1/k)$. In contrast, standard VI does not guarantee convergence in this setup.

We also point out that the convergence of Bellman error does not immediately imply point convergence, i.e., $TU^k - U^k \to 0$ does not immediately imply $U^k \to U^\star$, when $\gamma = 1$. Rather, we show (i) $U^k$ is a bounded sequence, (ii) any convergent subsequence $U^{k_j}$ converges to a fixed point $U^\infty$, and (iii) $U^k$ is elementwise monotonically nondecreasing and therefore has a single limit.

Next, we state our convergence result for general state-action spaces.

**Theorem 4.** *Let $\gamma = 1$. Let the state and action spaces be general (possibly infinite) sets. Let $T$ be the nonexpansive Bellman consistency or optimality operator for $V$ or $Q$, and assume $T$ is well defined.[3] Assume a fixed point exists (not necessarily unique). If $U^0 \leq TU^0$, then Anc-VI exhibits the rate*

$$\left\|TU^k - U^k\right\|_\infty \leq \frac{1}{k+1}\left\|U^0 - U^\star\right\|_\infty \qquad for\ k = 0, 1, \dots$$

*for any fixed point $U^\star$ satisfying $U^0 \leq U^\star$. Furthermore, $U^k \to U^\infty$ pointwise monotonically for some fixed point $U^\infty$.*

The convergence $U^k \to U^\infty$ pointwise in infinite state-action spaces is, in our view, a non-trivial contribution. When the state-action space is finite, pointwise convergence directly implies convergence in $\|\cdot\|_\infty$, and in this sense, Theorem 4 is generalization of Theorem 3. However, when the state-action space is infinite, pointwise convergence does not necessarily imply uniform convergence, i.e., $U^k \to U^\infty$ pointwise does not necessarily imply $U^k \to U^\infty$ in $\|\cdot\|_\infty$.

## 4 Complexity lower bound

We now present a complexity lower bound establishing optimality of Anc-VI.

**Theorem 5.** *Let $k \geq 0$, $n \geq k + 2$, $0 < \gamma \leq 1$, and $U^0 \in \mathbb{R}^n$. Then there exists an MDP with $|\mathcal{S}| = n$ and $|\mathcal{A}| = 1$ (which implies the Bellman consistency and optimality operator for $V$ and $Q$ all coincide as $T\colon \mathbb{R}^n \to \mathbb{R}^n$) such that $T$ has a fixed point $U^\star$ satisfying $U^0 \leq U^\star$ and*

$$\left\|TU^k - U^k\right\|_\infty \geq \frac{\gamma^k}{\sum_{i=0}^k \gamma^i}\left\|U^0 - U^\star\right\|_\infty$$

*for any iterates $\{U^i\}_{i=0}^k$ satisfying*

$$U^i \in U^0 + \mathrm{span}\{TU^0 - U^0, TU^1 - U^1, \dots, TU^{i-1} - U^{i-1}\} \qquad for\ i = 1, \dots, k.$$

*Proof outline of Theorem 5.* Without loss of generality, assume $n = k + 2$ and $U^0 = 0$. Consider the MDP $(\mathcal{S}, \mathcal{A}, P, r, \gamma)$ such that

$$\mathcal{S} = \{s_1, \dots, s_{k+2}\}, \quad \mathcal{A} = \{a_1\}, \quad P(s_i \,|\, s_j, a_1) = \mathbb{1}_{\{i=j=1,\ j=i+1\}}, \quad r(s_i, a_1) = \mathbb{1}_{\{i=2\}}.$$

Then, $T = \gamma \mathcal{P}^\pi U + [0, 1, 0, \dots, 0]^\intercal$, $U^\star = [0, 1, \gamma, \dots, \gamma^k]^\intercal$, and $\|U^0 - U^\star\|_\infty = 1$. Under the span condition, we can show that $(U^k)_1 = (U^k)_{k+2} = 0$. Then, we get

$$TU^k - U^k = \left(0, 1 - (U^k)_2, \gamma(U^k)_2 - (U^k)_3, \dots, \gamma(U^k)_k - (U^k)_{k+1}, \gamma(U^k)_{k+1}\right)$$

and this implies

$$\left(TU^k - U^k\right)_1 + \left(TU^k - U^k\right)_2 + \gamma^{-1}\left(TU^k - U^k\right)_3 + \dots + \gamma^{-k}\left(TU^k - U^k\right)_{k+2} = 1.$$

---

[3]Well-definedness of $T$ requires a $\sigma$-algebra on state and action spaces, expectation with respect to transition probability and policy to be well defined, boundedness and measurability of the output of Bellman operators, etc.

Taking the absolute value on both sides,

$$\left(1 + \cdots + \gamma^{-k}\right) \max_{1 \leq i \leq k+2} \left\{ |TU^k - U^k|_i \right\} \geq 1.$$

Therefore, we conclude

$$\left\| TU^k - U^k \right\|_\infty \geq \frac{\gamma^k}{\sum_{i=0}^k \gamma^i} \left\| U^0 - U^\star \right\|_\infty.$$

$\square$

Note that the case $\gamma = 1$ is included in Theorem 5. When $\gamma = 1$, the lower bound of Theorem 5 *exactly* matches the upper bound of Theorem 3.

Since

$$\frac{\gamma^k}{\sum_{i=0}^k \gamma^i} \leq \frac{\left(\gamma^{-1} - \gamma\right)\left(1 + \gamma - \gamma^{k+1}\right)}{\left(\gamma^{k+1}\right)^{-1} - \gamma^{k+1}} \leq \frac{4\gamma^k}{\sum_{i=0}^k \gamma^i} \qquad \text{for all } 0 < \gamma < 1,$$

the lower bound establishes optimality of the second rates Theorems 1 and 2 up to a constant of factor 4. Theorem 5 improves upon the prior state-of-the-art complexity lower bound established in the proof of [37, Theorem 3] by a factor $1 - \gamma^{k+1}$. (In [37, Theorem 3], a lower bound on the distance to optimal value function is provided. Their result has an implicit dependence on the initial distance to optimal value function $\|U^0 - U^\star\|_\infty$, so we make the dependence explicit, and we translate their result to a lower bound on the Bellman error. Once this is done, the difference between our lower bound of Theorem 5 and of [37, Theorem 3] is a factor of $1 - \gamma^{k+1}$. The worst-case MDP of [37, Theorem 3] and our worst-case MDP primarily differ in the rewards, while the states and the transition probabilities are almost the same.)

The so-called "span condition" of Theorem 5 is arguably very natural and is satisfied by standard VI and Anc-VI. The span condition is commonly used in the construction of complexity lower bounds on first-order optimization methods [59, 23, 25, 13, 14, 65] and has been used in the prior state-of-the-art lower bound for standard VI [37, Theorem 3]. However, designing an algorithm that breaks the lower bound of Theorem 5 by violating the span condition remains a possibility. In optimization theory, there is precedence of lower bounds being broken by violating seemingly natural and minute conditions [40, 35, 98].

## 5 Approximate Anchored Value Iteration

In this section, we show that the anchoring mechanism is robust against evaluation errors of the Bellman operator, just as much as the standard approximate VI.

Let $0 < \gamma < 1$ and let $T^\star$ be the Bellman optimality operator. The *Approximate Anchored Value Iteration* (Apx-Anc-VI) is

$$\begin{aligned}
U_\epsilon^k &= T^\star U^{k-1} + \epsilon^{k-1} \\
U^k &= \beta_k U^0 + (1 - \beta_k) U_\epsilon^k
\end{aligned} \qquad \text{(Apx-Anc-VI)}$$

for $k = 1, 2, \ldots$, where $\beta_k = 1/(\sum_{i=0}^k \gamma^{-2i})$, $U^0$ is an initial point, and the $\{\epsilon^k\}_{k=0}^\infty$ is the error sequence modeling approximate evaluations of $T^\star$.

Of course, the classical Approximate Value Iteration (Apx-VI) is

$$U^k = T^\star U^{k-1} + \epsilon^{k-1} \qquad \text{(Apx-VI)}$$

for $k = 1, 2, \ldots$, where $U^0$ is an initial point.

**Fact 1** (Classical result, [11, p.333])**.** *Let $0 < \gamma < 1$ be the discount factor. Let $T^\star$ be the Bellman optimality for $V$ or $Q$. Let $U^\star$ be the fixed point of $T^\star$. Then Apx-VI exhibits the rate*

$$\left\| T^\star U^k - U^k \right\|_\infty \leq (1 + \gamma)\gamma^k \left\| U^0 - U^\star \right\|_\infty + (1 + \gamma) \frac{1 - \gamma^k}{1 - \gamma} \max_{0 \leq i \leq k-1} \left\| \epsilon^i \right\|_\infty \quad \text{for } k = 1, 2, \ldots.$$

**Theorem 6.** *Let $0 < \gamma < 1$ be the discount factor. Let $T^\star$ and $\hat{T}^\star$ respectively be the Bellman optimality and anti-optimality operators for $V$ or $Q$. Let $U^\star$ and $\hat{U}^\star$ respectively be the fixed points of $T^\star$ and $\hat{T}^\star$. Then Apx-Anc-VI exhibits the rate*

$$\left\|T^\star U^k - U^k\right\|_\infty \leq \frac{\left(\gamma^{-1} - \gamma\right)\left(1 + 2\gamma - \gamma^{k+1}\right)}{(\gamma^{k+1})^{-1} - \gamma^{k+1}} \max\left\{\left\|U^0 - U^\star\right\|_\infty, \left\|U^0 - \hat{U}^\star\right\|_\infty\right\}$$

$$+ \frac{1+\gamma}{1+\gamma^{k+1}} \frac{1-\gamma^k}{1-\gamma} \max_{0 \leq i \leq k-1} \left\|\epsilon^i\right\|_\infty \qquad \text{for } k = 1, 2, \dots.$$

*If, furthermore, $U^0 \geq T^\star U^0$, then* (Apx-Anc-VI) *exhibits the rate*

$$\left\|T^\star U^k - U^k\right\|_\infty \leq \frac{\left(\gamma^{-1} - \gamma\right)\left(1 + \gamma - \gamma^{k+1}\right)}{(\gamma^{k+1})^{-1} - \gamma^{k+1}} \left\|U^0 - \hat{U}^\star\right\|_\infty + \frac{1+\gamma}{1+\gamma^{k+1}} \frac{1-\gamma^k}{1-\gamma} \max_{0 \leq i \leq k-1} \left\|\epsilon^i\right\|_\infty$$

*for $k = 1, 2, \dots$.*

The dependence on $\max \|\epsilon_i\|_\infty$ of Apx-Anc-VI is no worse than that of Apx-VI. In this sense, Apx-Anc-VI is robust against evaluation errors of the Bellman operator, just as much as the standard Apx-VI. Finally, we note that a similar analysis can be done for Apx-Anc-VI with the Bellman consistency operator.

## 6 Gauss–Seidel Anchored Value Iteration

In this section, we show that the anchoring mechanism can be combined with Gauss–Seidel-type updates in finite state-action spaces. Let $0 < \gamma < 1$ and let $T^\star \colon \mathbb{R}^n \to \mathbb{R}^n$ be the Bellman optimality operator. Define $T^\star_{GS} \colon \mathbb{R}^n \to \mathbb{R}^n$ as

$$T^\star_{GS} = T^\star_n \cdots T^\star_2 T^\star_1,$$

where $T^\star_j : \mathbb{R}^n \to \mathbb{R}^n$ is defined as

$$T^\star_j(U) = (U_1, \dots, U_{j-1}, (T^\star(U))_j, U_{j+1}, \dots, U_n)$$

for $j = 1, \dots, n$.

**Fact 2.** *[Classical result, [69, Theorem 6.3.4]] $T^\star_{GS}$ is a $\gamma$-contractive operator and has the same fixed point as $T^\star$.*

The *Gauss–Seidel Anchored Value Iteration* (GS-Anc-VI) is

$$U^k = \beta_k U^0 + (1 - \beta_k) T^\star_{GS} U^{k-1} \qquad \text{(GS-Anc-VI)}$$

for $k = 1, 2, \dots$, where $\beta_k = 1/(\sum_{i=0}^k \gamma^{-2i})$ and $U^0$ is an initial point.

**Theorem 7.** *Let the state and action spaces be finite sets. Let $0 < \gamma < 1$ be the discount factor. Let $T^\star$ and $\hat{T}^\star$ respectively be the Bellman optimality and anti-optimality operators for $V$ or $Q$. Let $U^\star$ and $\hat{U}^\star$ respectively be the fixed points of $T^\star$ and $\hat{T}^\star$. Then GS-Anc-VI exhibits the rate*

$$\left\|T^\star_{GS} U^k - U^k\right\|_\infty \leq \frac{\left(\gamma^{-1} - \gamma\right)\left(1 + 2\gamma - \gamma^{k+1}\right)}{(\gamma^{k+1})^{-1} - \gamma^{k+1}} \max\left\{\left\|U^0 - U^\star\right\|_\infty, \left\|U^0 - \hat{U}^\star\right\|_\infty\right\}$$

*for $k = 0, 1, \dots$. If, furthermore, $U^0 \leq T^\star_{GS} U^0$ or $U^0 \geq T^\star_{GS} U^0$, then GS-Anc-VI exhibits the rate*

$$\left\|T^\star_{GS} U^k - U^k\right\|_\infty \leq \frac{\left(\gamma^{-1} - \gamma\right)\left(1 + \gamma - \gamma^{k+1}\right)}{(\gamma^{k+1})^{-1} - \gamma^{k+1}} \left\|U^0 - U^\star\right\|_\infty \quad \text{if } U^0 \leq T^\star_{GS} U^0$$

$$\left\|T^\star_{GS} U^k - U^k\right\|_\infty \leq \frac{\left(\gamma^{-1} - \gamma\right)\left(1 + \gamma - \gamma^{k+1}\right)}{(\gamma^{k+1})^{-1} - \gamma^{k+1}} \left\|U^0 - \hat{U}^\star\right\|_\infty \quad \text{if } U^0 \geq T^\star_{GS} U^0$$

*for $k = 0, 1, \dots$.*

We point out that GS-Anc-VI cannot be directly extended to infinite action spaces since Hahn–Banach extension theorem is not applicable in the Gauss–Seidel setup. Furthermore, we note that a similar analysis can be carried out for GS-Anc-VI with the Bellman consistency operator.

# 7 Conclusion

We show that the classical value iteration (VI) is, in fact, suboptimal and that the anchoring mechanism accelerates VI to be optimal in the sense that the accelerated rate matches a complexity lower bound up to a constant factor of $4$. We also show that the accelerated iteration provably converges to a fixed point even when $\gamma = 1$, if a fixed point exists. Being able to provide a substantive improvement upon the classical VI is, in our view, a surprising contribution.

One direction of future work is to study the empirical effectiveness of Anc-VI. Another direction is to analyze Anc-VI in a model-free setting and, more broadly, to investigate the effectiveness of the anchor mechanism in more practical RL methods.

Our results lead us to believe that many of the classical foundations of dynamic programming and reinforcement learning may be improved with a careful examination based on an optimization complexity theory perspective. The theory of optimal optimization algorithms has recently enjoyed significant developments [44, 43, 45, 98, 66], the anchoring mechanism being one such example [49, 65], and the classical DP and RL theory may benefit from a similar line of investigation on iteration complexity.

## Acknowledgments and Disclosure of Funding

This work was supported by the the Information & communications Technology Planning & Evaluation (IITP) grant funded by the Korea government(MSIT) [NO.2021-0-01343, Artificial Intelligence Graduate School Program (Seoul National University)] and the Samsung Science and Technology Foundation (Project Number SSTF-BA2101-02). We thank Jisun Park for providing valuable feedback.

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

## A  Preliminaries

For notational unity, we use the symbol $U$ when both $V$ and $Q$ can be used.

**Lemma 1.** *[10, Lemma 1.1.1] Let $0 < \gamma \le 1$. If $U \le \tilde{U}$, then $T^\pi U \le T^\pi \tilde{U}, T^\star U \le T^\star \tilde{U}$.*

**Lemma 2.** *Let $0 < \gamma \le 1$. For any policy $\pi$, $\mathcal{P}^\pi$ is a nonexpansive linear operator such that if $U \le \tilde{U}$, $\mathcal{P}^\pi U \le \mathcal{P}^\pi \tilde{U}$.*

*Proof.* If $r(s,a) = 0$ for all $s \in \mathcal{S}$ and $a \in \mathcal{A}$, $T^\pi = \gamma \mathcal{P}^\pi$. Then by Lemma 1 and $\gamma$-contraction of $T^\pi$, we have the desired result. $\qquad\square$

**Lemma 3.** *Let $0 < \gamma < 1$. Let $T^\star$ and $\hat{T}^\star$ respectively be the Bellman optimality and anti-optimality operators. Let $U^\star$ and $\hat{U}^\star$ respectively be the fixed points of $T^\star$ and $\hat{T}^\star$. Then $\hat{U}^\star \le U^\star$.*

*Proof.* By definition, $\hat{U}^\star = \hat{T}^\star \hat{U}^\star \le T^\star \hat{U}^\star$. Thus, $\hat{U}^\star \le \lim_{m \to \infty} (T^\star)^m \hat{U}^\star = U^\star$. $\qquad\square$

## B  Omitted proofs in Section 2

First, we prove the following lemma by induction.

**Lemma 4.** *Let $0 < \gamma \le 1$, and if $\gamma = 1$, assume a fixed point $U^\pi$ exists. For the iterates $\{U^k\}_{k=0,1,\ldots}$ of Anc-VI,*

$$T^\pi U^k - U^k = \sum_{i=1}^{k} \left[ (\beta_i - \beta_{i-1}(1-\beta_i)) \left( \Pi_{j=i+1}^k (1-\beta_j) \right) (\gamma \mathcal{P}^\pi)^{k-i+1} (U^0 - U^\pi) \right]$$
$$- \beta_k (U^0 - U^\pi) + \left( \Pi_{j=1}^k (1-\beta_j) \right) (\gamma \mathcal{P}^\pi)^{k+1} (U^0 - U^\pi)$$

*where $\left( \Pi_{j=k+1}^k (1-\beta_j) \right) = 1$ and $\beta_0 = 1$.*

*Proof.* If $k = 0$, we have

$$\begin{aligned}
T^\pi U^0 - U^0 &= T^\pi U^0 - U^\pi - (U^0 - U^\pi) \\
&= T^\pi U^0 - T^\pi U^\pi - (U^0 - U^\pi) \\
&= \gamma \mathcal{P}^\pi (U^0 - U^\pi) - (U^0 - U^\pi)
\end{aligned}$$

If $k = m$, since $T^\pi$ is a linear operator,

$$\begin{aligned}
T^\pi U^m - U^m &= T^\pi U^m - (1-\beta_m) T^\pi U^{m-1} - \beta_m U^0 \\
&= T^\pi U^m - (1-\beta_m) T^\pi U^{m-1} - \beta_m U^\pi - \beta_m (U^0 - U^\pi) \\
&= T^\pi U^m - (1-\beta_m) T^\pi U^{m-1} - \beta_m T^\pi U^\pi - \beta_m (U^0 - U^\pi) \\
&= \gamma \mathcal{P}^\pi (U^m - (1-\beta_m) U^{m-1} - \beta_m U^\pi) - \beta_m (U^0 - U^\pi) \\
&= \gamma \mathcal{P}^\pi (\beta_m (U^0 - U^\pi) + (1-\beta_m)(T^\pi U^{m-1} - U^{m-1})) - \beta_m (U^0 - U^\pi) \\
&= (1-\beta_m) \gamma \mathcal{P}^\pi \sum_{i=1}^{m-1} \left[ (\beta_i - \beta_{i-1}(1-\beta_i)) \left( \Pi_{j=i+1}^{m-1} (1-\beta_j) \right) (\gamma \mathcal{P}^\pi)^{m-1-i+1} (U^0 - U^\pi) \right] \\
&\quad - (1-\beta_m) \gamma \mathcal{P}^\pi \beta_{m-1} (U^0 - U^\pi) + (1-\beta_m) \gamma \mathcal{P}^\pi \left( \Pi_{j=1}^{m-1} (1-\beta_j) \right) (\gamma \mathcal{P}^\pi)^m (U^0 - U^\pi) \\
&\quad + \beta_m \gamma \mathcal{P}^\pi (U^0 - U^\pi) - \beta_m (U^0 - U^\pi) \\
&= \sum_{i=1}^{m-1} \left[ (\beta_i - \beta_{i-1}(1-\beta_i)) \left( \Pi_{j=i+1}^{m} (1-\beta_j) \right) (\gamma \mathcal{P}^\pi)^{m-i+1} (U^0 - U^\pi) \right] \\
&\quad - \beta_{m-1}(1-\beta_m) \gamma \mathcal{P}^\pi (U^0 - U^\pi) + \beta_m \gamma \mathcal{P}^\pi (U^0 - U^\pi) \\
&\quad - \beta_m (U^0 - U^\pi) + \left( \Pi_{j=1}^{m} (1-\beta_j) \right) (\gamma \mathcal{P}^\pi)^{m+1} (U^0 - U^\pi)
\end{aligned}$$

$$= \sum_{i=1}^{m} \left[ (\beta_i - \beta_{i-1}(1-\beta_i)) \left( \Pi_{j=i+1}^{m}(1-\beta_j) \right) (\gamma \mathcal{P}^\pi)^{m-i+1} (U^0 - U^\pi) \right]$$
$$- \beta_m (U^0 - U^\pi) + \left( \Pi_{j=1}^{m}(1-\beta_j) \right) (\gamma \mathcal{P}^\pi)^{m+1} (U^0 - U^\pi)$$

$\square$

Now, we prove the first rate of Theorem 1.

*Proof of first rate in Theorem 1.* Taking $\|\cdot\|_\infty$-norm both sides of equality in Lemma 4, we get

$$\left\| T^\pi U^k - U^k \right\|_\infty \leq \sum_{i=1}^{k} |\beta_i - \beta_{i-1}(1-\beta_i)| \left( \Pi_{j=i+1}^{k}(1-\beta_j) \right) \left\| (\gamma \mathcal{P}^\pi)^{k-i+1} (U^0 - U^\pi) \right\|_\infty$$

$$+ \beta_k \left\| U^0 - U^\pi \right\|_\infty + \left( \Pi_{i=1}^{k}(1-\beta_i) \right) \left\| (\gamma \mathcal{P}^\pi)^{k+1} (U^0 - U^\pi) \right\|_\infty$$

$$\leq \left( \sum_{i=1}^{k} \gamma^{k-i+1} |\beta_i - \beta_{i-1}(1-\beta_i)| \left( \Pi_{j=i+1}^{k}(1-\beta_j) \right) + \beta_k + \gamma^{k+1} \Pi_{j=1}^{k}(1-\beta_j) \right)$$

$$\left\| U^0 - U^\star \right\|_\infty$$

$$= \left( \sum_{i=1}^{k} \gamma^{k+i-1} \frac{\left(1-\gamma^2\right)^2}{1-\gamma^{(2k+2)}} + \gamma^{2k} \frac{1-\gamma^2}{1-\gamma^{2k+2}} + \gamma^{k+1} \frac{1-\gamma^2}{1-\gamma^{2k+2}} \right) \left\| U^0 - U^\pi \right\|_\infty$$

$$= \frac{\left(\gamma^{-1} - \gamma\right) \left(1 + 2\gamma - \gamma^{k+1}\right)}{\left(\gamma^{k+1}\right)^{-1} - \gamma^{k+1}} \left\| U^0 - U^\pi \right\|_\infty,$$

where the first inequality comes from triangular inequality, second inequality is from Lemma 2, and equality come from calculations. $\square$

For the second rate of Theorem 1, we introduce following lemma.

**Lemma 5.** *Let $0 < \gamma < 1$. Let $T$ be Bellman consistency or optimality operator. For the iterates $\{U^k\}_{k=0,1,\ldots}$ of Anc-VI, if $U^0 \leq TU^0$, then $U_{k-1} \leq U_k \leq TU_{k-1} \leq TU_k \leq U^\star$ for $1 \leq k$. Also, if $U^0 \geq TU^0$, then $U_{k-1} \geq U_k \geq TU_{k-1} \geq TU_k \geq U^\star$ for $1 \leq k$.*

*Proof.* First, let $U^0 \leq TU^0$. If $k=1$, $U^0 \leq \beta_1 U^0 + (1-\beta_1)TU^0 = U_1 \leq TU^0$ by assumption. Since $U^0 \leq U^1$, $TU^0 \leq TU^1$ by monotonicity of Bellman consistency and optimality operators.

By induction,
$$U^k = \beta_k U^0 + (1-\beta_k)TU^{k-1} \leq TU^{k-1},$$

and since $\beta_k \leq \beta_{k-1}$,
$$\beta_k U^0 + (1-\beta_k)TU^{k-1} \geq \beta_{k-1}U^0 + (1-\beta_{k-1})TU^{k-1}$$
$$\geq \beta_{k-1}U^0 + (1-\beta_{k-1})TU^{k-2}$$
$$= U^{k-1}.$$

Also, $U^{k-1} \leq U^k$ implies $TU^{k-1} \leq TU^k$ by monotonicity of Bellman consistency and optimality operators, and $U^k \leq TU^k$ implies that $U^k \leq \lim_{m\to\infty} (T)^m U^k = U^\star$ for all $k = 0, 1, \ldots$.

Now, suppose $U^0 \geq TU^0$. If $k = 1$, $U^0 \geq \beta_1 U^0 + (1-\beta_1)TU^0 = U_1 \geq TU^0$ by assumption. Since $U^0 \geq U^1$, $TU^0 \geq TU^1$ by monotonicity of Bellman consistency and optimality operators.

By induction,
$$U^k = \beta_k U^0 + (1-\beta_k)TU^{k-1} \geq TU^{k-1},$$

and since $\beta_k \leq \beta_{k-1}$,
$$\beta_k U^0 + (1-\beta_k)TU^{k-1} \leq \beta_{k-1}U^0 + (1-\beta_{k-1})TU^{k-1}$$
$$\leq \beta_{k-1}U^0 + (1-\beta_{k-1})TU^{k-2}$$
$$= U^{k-1}.$$

Also, $U^{k-1} \geq U^k$ implies $TU^{k-1} \geq TU^k$ by monotonicity of Bellman consistency and optimality operators, and $U_k \geq TU_k$ implies that $U^k \geq \lim_{m\to\infty} (T)^m U^k = U^\star$ for all $k = 0, 1, \dots$.     $\square$

Now, we prove following key lemmas.

**Lemma 6.** *Let $0 < \gamma \leq 1$, and assume a fixed point $U^\pi$ exists if $\gamma = 1$. For the iterates $\{U^k\}_{k=0,1,\dots}$ of Anc-VI, if $U^0 \leq U^\pi$,*

$$T^\pi U^k - U^k \leq \sum_{i=1}^k \left[ (\beta_i - \beta_{i-1}(1 - \beta_i)) \left( \Pi_{j=i+1}^k (1 - \beta_j) \right) (\gamma \mathcal{P}^\pi)^{k-i+1} (U^0 - U^\pi) \right]$$
$$- \beta_k (U^0 - U^\pi),$$

*where $\left( \Pi_{j=k+1}^k (1 - \beta_j) \right) = 1$ and $\beta_0 = 1$.*

**Lemma 7.** *Let $0 < \gamma < 1$. For the iterates $\{U^k\}_{k=0,1,\dots}$ of Anc-VI, if $U^0 \geq T^\pi U^0$,*

$$T^\pi U^k - U^k \geq \sum_{i=1}^k \left[ (\beta_i - \beta_{i-1}(1 - \beta_i)) \left( \Pi_{j=i+1}^k (1 - \beta_j) \right) (\gamma \mathcal{P}^\pi)^{k-i+1} (U^0 - U^\pi) \right]$$
$$- \beta_k (U^0 - U^\pi),$$

*where $\left( \Pi_{j=k+1}^k (1 - \beta_j) \right) = 1$ and $\beta_0 = 1$.*

*Proof of Lemma 6.* If $U^0 \leq U^\pi$, we get

$$T^\pi U^k - U^k = \sum_{i=1}^k \left[ (\beta_i - \beta_{i-1}(1 - \beta_i)) \left( \Pi_{j=i+1}^k (1 - \beta_j) \right) (\gamma \mathcal{P}^\pi)^{k-i+1} (U^0 - U^\pi) \right]$$
$$- \beta_k (U^0 - U^\pi) + \left( \Pi_{j=1}^k (1 - \beta_j) \right) (\gamma \mathcal{P}^\pi)^{k+1} (U^0 - U^\pi)$$
$$\leq \sum_{i=1}^k \left[ (\beta_i - \beta_{i-1}(1 - \beta_i)) \left( \Pi_{j=i+1}^k (1 - \beta_j) \right) (\gamma \mathcal{P}^\pi)^{k-i+1} (U^0 - U^\pi) \right] - \beta_k (U^0 - U^\pi),$$

by Lemma 4 and the fact that $\left( \Pi_{j=1}^k (1 - \beta_j) \right) (\gamma \mathcal{P}^\pi)^{k+1} (U^0 - U^\pi) \leq 0$.     $\square$

*Proof of Lemma 7.* If $U^0 \geq TU^0$, $U^0 - U^\pi \geq 0$ by Lemma 5. Hence, by Lemma 4, we have

$$T^\pi U^k - U^k \geq \sum_{i=1}^k \left[ (\beta_i - \beta_{i-1}(1 - \beta_i)) \left( \Pi_{j=i+1}^k (1 - \beta_j) \right) (\gamma \mathcal{P}^\pi)^{k-i+1} (U^0 - U^\pi) \right] - \beta_k (U^0 - U^\pi),$$

since $0 \leq \left( \Pi_{j=1}^k (1 - \beta_j) \right) (\gamma \mathcal{P}^\pi)^{k+1} (U^0 - U^\pi)$.     $\square$

Now, we prove the second rates of Theorem 1.

*Proof of second rates in Theorem 1.* Let $0 < \gamma < 1$. By Lemma 5, if $U^0 \leq T^\pi U^0$, then $U^0 \leq U^\pi$. Hence,

$$0 \leq T^\pi U^k - U^k$$
$$\leq \sum_{i=1}^k \left[ (\beta_i - \beta_{i-1}(1 - \beta_i)) \left( \Pi_{j=i+1}^k (1 - \beta_j) \right) (\gamma \mathcal{P}^\pi)^{k-i+1} (U^0 - U^\pi) \right] - \beta_k (U^0 - U^\pi),$$

by Lemma 6. Taking $\|\cdot\|_\infty$-norm both sides, we have

$$\left\| T^\pi U^k - U^k \right\|_\infty \leq \frac{(\gamma^{-1} - \gamma)(1 + \gamma - \gamma^{k+1})}{(\gamma^{k+1})^{-1} - \gamma^{k+1}} \left\| U^0 - U^\pi \right\|_\infty.$$

Otherwise, if $U^0 \geq TU^0$, $U^k \geq TU^k$ by Lemma 5. Since

$$0 \geq T^\pi U^k - U^k$$
$$\geq \sum_{i=1}^{k} \left[ (\beta_i - \beta_{i-1}(1 - \beta_i)) \left( \Pi_{j=i+1}^{k}(1 - \beta_j) \right) (\gamma \mathcal{P}^\pi)^{k-i+1} (U^0 - U^\pi) \right] - \beta_k (U^0 - U^\pi),$$

by Lemma 7, taking $\|\cdot\|_\infty$-norm both sides, we obtain same rate as before.

Lastly, Taylor series expansion for both rates at $\gamma = 1$ is

$$\frac{\left( \gamma^{-1} - \gamma \right) \left( 1 + 2\gamma - \gamma^{k+1} \right)}{(\gamma^{k+1})^{-1} - \gamma^{k+1}} = \frac{2}{k+1} - \frac{k-1}{k+1}(\gamma - 1) + O((\gamma - 1)^2),$$

$$\frac{\left( \gamma^{-1} - \gamma \right) \left( 1 + \gamma - \gamma^{k+1} \right)}{(\gamma^{k+1})^{-1} - \gamma^{k+1}} = \frac{1}{k+1} - \frac{k}{k+1}(\gamma - 1) + O((\gamma - 1)^2).$$

$\square$

For the analyses of Anc-VI for Bellman optimality operator, we first prove following two lemmas.

**Lemma 8.** *Let $0 < \gamma \leq 1$. If $\gamma = 1$, assume a fixed point $U^\star$ exists. Then, if $0 \leq \alpha \leq 1$ and $U - (1 - \alpha)\tilde{U} - \alpha U^\star \leq \bar{U}$, there exist nonexpansive linear operator $\mathcal{P}_H$ such that*

$$T^\star U - (1 - \alpha)T^\star \tilde{U} - \alpha T^\star U^\star \leq \gamma \mathcal{P}_H \bar{U}.$$

**Lemma 9.** *Let $0 < \gamma < 1$. If $0 \leq \alpha \leq 1$ and $\bar{U} \leq U - (1 - \alpha)\tilde{U} - \alpha \hat{U}^\star$, then there exist nonexpansive linear operator $\hat{\mathcal{P}}_H$ such that*

$$\gamma \hat{\mathcal{P}}_H(\bar{U}) \leq T^\star U - \alpha T^\star \tilde{U} - (1 - \alpha)\hat{T}^\star \hat{U}^\star.$$

*Proof of Lemma 8.* First, let $U = V, \tilde{U} = \tilde{V}, U^\star = V^\star, \bar{U} = \bar{V}$, and $V - (1 - \alpha)\tilde{V} - \alpha V^\star \leq \bar{V}$. If action space is finite,

$$T^\star V - (1 - \alpha)T^\star \tilde{V} - \alpha T^\star V^\star \leq T^\pi V - (1 - \alpha)T^\pi \tilde{V} - \alpha T^\pi V^\star$$
$$= \gamma \mathcal{P}^\pi \left( V - (1 - \alpha)\tilde{V} - \alpha V^\star \right)$$
$$\leq \gamma \mathcal{P}^\pi \bar{V}$$

where $\pi$ is the greedy policy satisfying $T^\pi V = T^\star V$, first inequality is from $T^\pi \tilde{V} \leq T^\star \tilde{V}$ and $T^\pi V^\star \leq T^\star V^\star$, and second inequality comes from Lemma 1. Thus, we can conclude $\mathcal{P}_H = \mathcal{P}^\pi$.

Otherwise, if action space is infinite, define $\mathcal{P}(c\bar{V}) = c \sup_{s \in \mathcal{S}} \bar{V}(s)$ for $c \in \mathbb{R}$ and previously given $\bar{V}$. Let $M$ be linear space spanned by $\bar{V}$ with $\|\cdot\|_\infty$-norm. Then, $\mathcal{P}$ is linear functional on $M$ and $\|\mathcal{P}\|_{\mathrm{op}} \leq 1$ since $\frac{|c \sup_{s \in \mathcal{S}} \bar{V}(s)|}{\|c\bar{V}\|_\infty} \leq 1$. Due to Hahn–Banach extension Theorem, there exist linear functional $\mathcal{P}_h : \mathcal{F}(\mathcal{S}) \to \mathbb{R}$ with $\mathcal{P}_h(\bar{V}) = \sup_{s \in \mathcal{S}} \bar{V}(s)$ and $\|\mathcal{P}_h\|_{\mathrm{op}} \leq 1$. Furthermore, we can define $\mathcal{P}_H : \mathcal{F}(\mathcal{S}) \to \mathcal{F}(\mathcal{S})$ such that $\mathcal{P}_H V(s) = \mathcal{P}_h(V)$ for all $s \in \mathcal{S}$. Then, since $\|\mathcal{P}_H(V)\|_\infty = |\mathcal{P}_h(V)| \leq \|\mathcal{P}_h\|_{\mathrm{op}} \leq 1$ for $\|V\|_\infty \leq 1$, we have $\|\mathcal{P}_H\|_\infty \leq 1$. Therefore, $\mathcal{P}_H$ is

nonexpansive linear operator in $\|\cdot\|_\infty$-norm. Then,

$$T^\star V(s) - (1-\alpha)T^\star \tilde{V}(s) - \alpha T^\star V^\star(s)$$

$$= \sup_{a\in\mathcal{A}} \left\{ r(s,a) + \gamma \mathbb{E}_{s'\sim P(\cdot\,|\,s,a)}\left[V(s')\right] \right\} - \sup_{a\in\mathcal{A}} \left\{ (1-\alpha)r(s,a) + (1-\alpha)\gamma \mathbb{E}_{s'\sim P(\cdot\,|\,s,a)}\left[\tilde{V}(s')\right] \right\}$$

$$\quad - \sup_{a\in\mathcal{A}} \left\{ \alpha r(s,a) + \alpha\gamma \mathbb{E}_{s'\sim P(\cdot\,|\,s,a)}\left[V^\star(s')\right] \right\}$$

$$\leq \sup_{a\in\mathcal{A}} \left\{ r(s,a) + \gamma \mathbb{E}_{s'\sim P(\cdot\,|\,s,a)}\left[V(s')\right] - (1-\alpha)r(s,a) - (1-\alpha)\gamma \mathbb{E}_{s'\sim P(\cdot\,|\,s,a)}\left[\tilde{V}(s')\right] \right\}$$

$$\quad - \sup_{a\in\mathcal{A}} \left\{ \alpha r(s,a) + \alpha\gamma \mathbb{E}_{s'\sim P(\cdot\,|\,s,a)}\left[V^\star(s')\right] \right\}$$

$$\leq \gamma \sup_{a\in\mathcal{A}} \left\{ \mathbb{E}_{s'\sim P(\cdot\,|\,s,a)}\left[V(s') - (1-\alpha)\tilde{V}(s') - \alpha V^\star(s')\right] \right\}$$

$$\leq \gamma \sup_{s'\in\mathcal{S}} \{V(s') - (1-\alpha)\tilde{V}(s') - \alpha V^\star(s')\}$$

$$\leq \gamma \sup_{s'\in\mathcal{S}} \bar{V}(s').$$

for all $s \in \mathcal{S}$. Therefore, we have

$$T^\star V - (1-\alpha)T^\star \tilde{V} - \alpha T^\star V^\star \leq \gamma \mathcal{P}_H(\bar{V}).$$

Similarly, let $U = Q, \tilde{U} = \tilde{Q}, U^\star = Q^\star, \bar{U} = \bar{Q}$, and $Q - (1-\alpha)\tilde{Q} - \alpha Q^\star \leq \bar{Q}$.

If action space is finite,

$$T^\star Q - (1-\alpha)T^\star \tilde{Q} - \alpha T^\star Q^\star \leq \gamma \mathcal{P}^\pi \left( Q - (1-\alpha)\tilde{Q} - \alpha Q^\star \right)$$

$$\leq \gamma \mathcal{P}^\pi \bar{Q}$$

where $\pi$ is the greedy policy satisfying $T^\pi Q = T^\star Q$, first inequality is from $T^\pi \tilde{Q} \leq T^\star \tilde{Q}$ and $T^\pi Q^\star \leq T^\star Q^\star$, and second inequality comes from Lemma 1. Then, we can conclude $\mathcal{P}_H = \mathcal{P}^\pi$.

Otherwise, if action space is infinite, define $\mathcal{P}(c\bar{Q}) = c\sup_{(s',a')\in\mathcal{S}\times\mathcal{A}} \bar{Q}(s',a')$ for $c \in \mathbb{R}$ and previously given $\bar{Q}$. Let $M$ be linear space spanned by $\bar{Q}$ with $\|\cdot\|_\infty$-norm. Then, $\mathcal{P}$ is linear functional on $M$ and $\|\mathcal{P}\|_{\mathrm{op}} \leq 1$. Due to Hahn–Banach extension Theorem, there exist linear functional $\mathcal{P}_h \colon \mathcal{F}(\mathcal{S}\times\mathcal{A}) \to \mathbb{R}$ with $\mathcal{P}_h(\bar{Q}) = \sup_{(s',a')\in\mathcal{S}\times\mathcal{A}} \bar{Q}(s',a')$ and $\|\mathcal{P}_h\|_{\mathrm{op}} \leq 1$. Furthermore, we can define $\mathcal{P}_H \colon \mathcal{F}(\mathcal{S}\times\mathcal{A}) \to \mathcal{F}(\mathcal{S}\times\mathcal{A})$ such that $\mathcal{P}_H Q(s,a) = \mathcal{P}_h(Q)$ for all $(s,a) \in \mathcal{S}\times\mathcal{A}$ and $\|P_H\|_\infty \leq 1$. Therefore, $\mathcal{P}_H$ is nonexpansive linear operator in $\|\cdot\|_\infty$-norm. Then,

$$T^\star Q(s,a) - (1-\alpha)T^\star \tilde{Q}(s,a) - \alpha T^\star Q^\star(s,a)$$

$$= r(s,a) + \gamma \mathbb{E}_{s'\sim P(\cdot\,|\,s,a)}\left[\sup_{a'\in\mathcal{A}} Q(s',a')\right] - (1-\alpha)r(s,a) - (1-\alpha)\gamma \mathbb{E}_{s'\sim P(\cdot\,|\,s,a)}\left[\sup_{a'\in\mathcal{A}} \tilde{Q}(s',a')\right]$$

$$\quad - \alpha r(s,a) - \alpha\gamma \mathbb{E}_{s'\sim P(\cdot\,|\,s,a)}\left[\sup_{a'\in\mathcal{A}} Q^\star(s',a')\right]$$

$$\leq \gamma \mathbb{E}_{s'\sim P(\cdot\,|\,s,a)}\left[\sup_{a'\in\mathcal{A}} \left\{Q(s',a') - (1-\alpha)\tilde{Q}(s',a')\right\}\right] - \gamma \mathbb{E}_{s'\sim P(\cdot\,|\,s,a)}\left[\sup_{a'\in\mathcal{A}} \alpha Q(s',a')\right]$$

$$\leq \gamma \mathbb{E}_{s'\sim P(\cdot\,|\,s,a)}\left[\sup_{a'\in\mathcal{A}} \left\{Q(s',a') - (1-\alpha)\tilde{Q}(s',a') - \alpha Q^\star(s',a')\right\}\right]$$

$$\leq \gamma \sup_{(s',a')\in\mathcal{S}\times\mathcal{A}} \left\{Q(s',a') - (1-\alpha)\tilde{Q}(s',a') - \alpha Q^\star(s',a')\right\},$$

$$\leq \gamma \sup_{(s',a')\in\mathcal{S}\times\mathcal{A}} \bar{Q}(s',a')$$

for all $(s,a) \in \mathcal{S}\times\mathcal{A}$. Therefore, we have

$$T^\star Q - (1-\alpha)T^\star \tilde{Q} - \alpha T^\star Q^\star \leq \gamma \mathcal{P}_H(\bar{Q}).$$

$\square$

*Proof of Lemma 9.* Note that $\hat{T}^\star$ is Bellman anti-optimality operators for $V$ or $Q$, and $\hat{U}^\star$ is the fixed point of $\hat{T}^\star$. First, let $U = V, \tilde{U} = \tilde{V}, \hat{U}^\star = \hat{V}^\star, \bar{U} = \bar{V}$, and $\bar{V} \leq V - (1-\alpha)\tilde{V} - \alpha\hat{V}^\star$. Then,

$$T^\star V(s) - (1-\alpha)T^\star\tilde{V}(s) - \alpha\hat{T}^\star\hat{V}^\star(s)$$

$$= \sup_{a\in\mathcal{A}}\left\{r(s,a) + \gamma\mathbb{E}_{s'\sim P(\cdot\,|\,s,a)}[V(s')]\right\} - \sup_{a\in\mathcal{A}}\left\{(1-\alpha)r(s,a) + (1-\alpha)\gamma\mathbb{E}_{s'\sim P(\cdot\,|\,s,a)}\left[\tilde{V}(s')\right]\right\}$$

$$- \inf_{a\in\mathcal{A}}\left\{\alpha r(s,a) + \alpha\gamma\mathbb{E}_{s'\sim P(\cdot\,|\,s,a)}\left[\hat{V}^\star(s')\right]\right\}$$

$$\geq \inf_{a\in\mathcal{A}}\left\{r(s,a) + \gamma\mathbb{E}_{s'\sim P(\cdot\,|\,s,a)}[V(s')] - (1-\alpha)r(s,a) - (1-\alpha)\gamma\mathbb{E}_{s'\sim P(\cdot\,|\,s,a)}\left[\tilde{V}(s')\right]\right\}$$

$$- \inf_{a\in\mathcal{A}}\left\{\alpha r(s,a) + \alpha\gamma\mathbb{E}_{s'\sim P(\cdot\,|\,s,a)}\left[\hat{V}^\star(s')\right]\right\}$$

$$\geq \gamma\inf_{a\in\mathcal{A}}\left\{\mathbb{E}_{s'\sim P(\cdot\,|\,s,a)}\left[V(s') - (1-\alpha)\tilde{V}(s') - \alpha\hat{V}^\star(s')\right]\right\}.$$

Then, if action space is finite,

$$T^\star V - (1-\alpha)T^\star\tilde{V} - \alpha T^\star V^\star \geq \gamma\mathcal{P}^{\hat{\pi}}\left(V - (1-\alpha)\tilde{V} - \alpha\hat{V}^\star\right)$$

$$\geq \gamma\mathcal{P}^{\hat{\pi}}\bar{V}$$

where $\hat{\pi}$ is the policy satisfying $\hat{\pi}(\cdot\,|\,s) = \operatorname{argmin}_{a\in\mathcal{A}}\mathbb{E}_{s'\sim P(\cdot\,|\,s,a)}\left[V(s') - (1-\alpha)\tilde{V}(s') - \alpha\hat{V}^\star(s')\right]$ and second inequality comes from Lemma 1. Thus, we can conclude $\mathcal{P}_H = \mathcal{P}^\pi$.

Otherwise, if action space is infinite, define $\hat{\mathcal{P}}(c\bar{V}) = c\inf_{s\in\mathcal{S}}\bar{V}(s)$ for $c \in \mathbb{R}$ and previously given $\bar{V}$. Let $M$ be linear space spanned by $\bar{V}$ with $\|\cdot\|_\infty$-norm. Then, $\hat{\mathcal{P}}$ is linear functional on $M$ and $\|\hat{\mathcal{P}}\|_{\mathrm{op}} \leq 1$ since $\frac{|c\inf_{s\in\mathcal{S}}\bar{V}(s)|}{\|c\bar{V}\|_\infty} \leq 1$. Due to Hahn–Banach extension Theorem, there exist linear functional $\hat{\mathcal{P}}_h\colon \mathcal{F}(\mathcal{S}) \to \mathbb{R}$ with $\hat{\mathcal{P}}_h(\bar{V}) = \inf_{s\in\mathcal{S}}\bar{V}(s)$ and $\|\hat{\mathcal{P}}_h\|_{\mathrm{op}} \leq 1$. Furthermore, we can define $\hat{\mathcal{P}}_H\colon \mathcal{F}(\mathcal{S}) \to \mathcal{F}(\mathcal{S})$ such that $\hat{\mathcal{P}}_H V(s) = \hat{\mathcal{P}}_h(V)$ for all $s \in \mathcal{S}$. Then $\|\hat{\mathcal{P}}_H\|_\infty \leq 1$ since $\|\hat{\mathcal{P}}_H(V)\|_\infty = |\hat{\mathcal{P}}_h(V)| \leq \|\hat{\mathcal{P}}_h\|_{\mathrm{op}} \leq 1$ for $\|V\|_\infty \leq 1$. . Thus, $\hat{\mathcal{P}}_H$ is nonexpansive linear operator in $\|\cdot\|_\infty$-norm. Then, we have

$$T^\star V(s) - (1-\alpha)T^\star\tilde{V}(s) - \alpha\hat{T}^\star\hat{V}^\star(s) \geq \gamma\inf_{a\in\mathcal{A}}\left\{\mathbb{E}_{s'\sim P(\cdot\,|\,s,a)}\left[V(s') - (1-\alpha)\tilde{V}(s') - \alpha\hat{V}^\star(s')\right]\right\}$$

$$\geq \gamma\inf_{s'\in\mathcal{S}}\{V(s') - (1-\alpha)\tilde{V}(s') - \alpha\hat{V}^\star(s')\}$$

$$\geq \gamma\inf_{s'\in\mathcal{S}}\{\bar{V}(s')\}$$

for all $s \in \mathcal{S}$. Therefore, we have

$$\gamma\hat{\mathcal{P}}_H(\bar{V}) \leq T^\star V(s) - (1-\alpha)T^\star\tilde{V}(s) - \alpha\hat{T}^\star\hat{V}^\star(s).$$

Similarly, let $U = Q, \tilde{U} = \tilde{Q}, \hat{U}^\star = \hat{Q}^\star, \bar{U} = \bar{Q}$, and $\bar{Q} \leq Q - (1-\alpha)\tilde{Q} - \alpha\hat{Q}^\star$. Then,

$$T^\star Q(s,a) - \alpha T^\star\tilde{Q}(s,a) - (1-\alpha)\hat{T}^\star\hat{Q}^\star(s,a)$$

$$= r(s,a) + \gamma\mathbb{E}_{s'\sim P(\cdot\,|\,s,a)}\left[\sup_{a'\in\mathcal{A}}Q(s',a')\right] - (1-\alpha)r(s,a) - (1-\alpha)\gamma\mathbb{E}_{s'\sim P(\cdot\,|\,s,a)}\left[\sup_{a'\in\mathcal{A}}\tilde{Q}(s',a')\right]$$

$$- \alpha r(s,a) - \alpha\gamma\mathbb{E}_{s'\sim P(\cdot\,|\,s,a)}\left[\inf_{a'\in\mathcal{A}}\hat{Q}^\star(s',a')\right]$$

$$\geq \gamma\mathbb{E}_{s'\sim P(\cdot\,|\,s,a)}\left[\inf_{a'\in\mathcal{A}}\left\{Q(s',a') - (1-\alpha)\tilde{Q}(s',a')\right\}\right] - \gamma\mathbb{E}_{s'\sim P(\cdot\,|\,s,a)}\left[\inf_{a'\in\mathcal{A}}\alpha\hat{Q}(s',a')\right]$$

$$\geq \gamma\mathbb{E}_{s'\sim P(\cdot\,|\,s,a)}\left[\inf_{a'\in\mathcal{A}}\left\{Q(s',a') - (1-\alpha)\tilde{Q}(s',a') - \alpha\hat{Q}^\star(s',a')\right\}\right].$$

Hence, if action space is finite,

$$T^\star Q - (1-\alpha)T^\star \tilde{Q} - \alpha T^\star Q^\star \geq \gamma \mathcal{P}^{\hat{\pi}}\left(Q - (1-\alpha)\tilde{Q} - \alpha Q^\star\right),$$
$$\geq \gamma \mathcal{P}^{\hat{\pi}} \bar{Q},$$

where $\hat{\pi}$ is the policy satisfying $\hat{\pi}(\cdot \,|\, s) = \operatorname{argmin}_{a \in \mathcal{A}} \mathbb{E}_{s' \sim P(\cdot \,|\, s,a)}\left[Q(s') - (1-\alpha)\tilde{Q}(s') - \alpha Q^\star(s')\right]$ and second inequality comes from Lemma 1. Then, we can conclude $\mathcal{P}_H = \mathcal{P}^{\hat{\pi}}$.

Otherwise, if action space is infinite, define $\hat{\mathcal{P}}(c\bar{Q}) = c \inf_{(s',a') \in \mathcal{S} \times \mathcal{A}} \bar{Q}(s',a')$ for $c \in \mathbb{R}^n$ and previously given $\bar{Q}$. Let $M$ be linear space spanned by $\bar{Q}$ with $\|\cdot\|_\infty$-norm. Then, $\mathcal{P}$ is linear functional on $M$ with $\|\hat{\mathcal{P}}\|_{\text{op}} \leq 1$. Due to Hahn–Banach extension Theorem, there exist linear functional $\hat{\mathcal{P}}_h \colon \mathcal{F}(\mathcal{S} \times \mathcal{A}) \to \mathbb{R}$ with $\hat{\mathcal{P}}_h(\bar{Q}) = \inf_{(s',a') \in \mathcal{S} \times \mathcal{A}} \bar{Q}(s',a')$ and $\|\hat{\mathcal{P}}_h\|_{\text{op}} \leq 1$. Furthermore, we can define $\hat{\mathcal{P}}_H \colon \mathcal{F}(\mathcal{S} \times \mathcal{A}) \to \mathcal{F}(\mathcal{S} \times \mathcal{A})$ such that $\mathcal{P}_H Q(s,a) = \hat{\mathcal{P}}_h(Q)$ for all $(s,a) \in \mathcal{S} \times \mathcal{A}$ and $\|\hat{P}_H\|_\infty \leq 1$. Thus $\hat{\mathcal{P}}_H$ is nonexpansive linear operator in $\|\cdot\|_\infty$-norm. Then, we have

$$T^\star Q(s,a) - \alpha T^\star \tilde{Q}(s,a) - (1-\alpha)\hat{T}^\star \hat{Q}^\star(s,a)$$
$$\geq \gamma \mathbb{E}_{s' \sim P(\cdot \,|\, s,a)}\left[\inf_{a' \in \mathcal{A}}\left\{Q(s',a') - (1-\alpha)\tilde{Q}(s',a') - \alpha \hat{Q}^\star(s',a')\right\}\right]$$
$$\geq \gamma \inf_{(s',a') \in \mathcal{S} \times \mathcal{A}}\left\{Q(s',a') - (1-\alpha)\tilde{Q}(s',a') - \alpha \hat{Q}^\star(s',a')\right\}$$
$$\geq \gamma \inf_{(s',a') \in \mathcal{S} \times \mathcal{A}} \bar{Q}(s',a'),$$

for all $(s,a) \in \mathcal{S} \times \mathcal{A}$. Therefore, we have

$$\gamma \hat{\mathcal{P}}_H(\bar{Q}) \leq T^\star Q - (1-\alpha)T^\star \tilde{Q} - \alpha \hat{T}^\star \hat{Q}^\star.$$

$\square$

Now, we present our key lemmas for the first rate of Theorem 2.

**Lemma 10.** *Let $0 < \gamma \leq 1$. If $\gamma = 1$, assume a fixed point $U^\star$ exists. For the iterates $\{U^k\}_{k=0,1,\dots}$ of Anc-VI, there exist nonexpansive linear operators $\{\mathcal{P}^l\}_{l=0,1,\dots,k}$ such that*

$$T^\star U^k - U^k \leq \sum_{i=1}^k \left[(\beta_i - \beta_{i-1}(1-\beta_i))\left(\Pi_{j=i+1}^k (1-\beta_j)\right)\left(\Pi_{l=k}^i \gamma \mathcal{P}^l\right)(U^0 - U^\star)\right]$$
$$- \beta_k(U^0 - U^\star) + \left(\Pi_{j=1}^k(1-\beta_j)\right)\left(\Pi_{l=k}^0 \gamma \mathcal{P}^l\right)(U^0 - U^\star)$$

*where $\Pi_{j=k+1}^k(1-\beta_j) = 1$ and $\beta_0 = 1$.*

**Lemma 11.** *Let $0 < \gamma < 1$. For the iterates $\{U^k\}_{k=0,1,\dots}$ of Anc-VI, there exist nonexpansive linear operators $\{\hat{\mathcal{P}}^l\}_{l=0,1,\dots,k}$ such that*

$$T^\star U^k - U^k \geq \sum_{i=1}^k \left[(\beta_i - \beta_{i-1}(1-\beta_i))\left(\Pi_{j=i+1}^k (1-\beta_j)\right)\left(\Pi_{l=k}^i \gamma \hat{\mathcal{P}}^l\right)(U^0 - \hat{U}^\star)\right]$$
$$- \beta_k(U^0 - \hat{U}^\star) + \left(\Pi_{j=1}^k(1-\beta_j)\right)\left(\Pi_{l=k}^0 \gamma \hat{\mathcal{P}}^l\right)(U^0 - \hat{U}^\star),$$

*where $\Pi_{j=k+1}^k(1-\beta_j) = 1$ and $\beta_0 = 1$.*

We prove previous lemmas by induction.

*Proof of Lemma 10.* If $k = 0$,
$$T^\star U^0 - U^0 = T^\star U^0 - U^\star - (U^0 - U^\star)$$
$$= T^\star U^0 - T^\star U^\star - (U^0 - U^\star)$$
$$\leq \gamma \mathcal{P}^0(U^0 - U^\star) - (U^0 - U^\star).$$

where inequality comes from first inequality in Lemma 8 with $\alpha = 1, U = U^0, \bar{U} = U^0 - U^\star$.
By induction,

$$
\begin{aligned}
&U^k - (1 - \beta_k)U^{k-1} - \beta_k U^\star \\
&= \beta_k \left(U^0 - U^\star\right) + (1 - \beta_k)(T^\star U^{k-1} - U^{k-1}) \\
&\leq (1 - \beta_k)\sum_{i=1}^{k-1}\left[(\beta_i - \beta_{i-1}(1 - \beta_i))\left(\Pi_{j=i+1}^{k-1}(1 - \beta_j)\right)\left(\Pi_{l=k-1}^{i}\gamma\mathcal{P}^l\right)(U^0 - U^\star)\right] \\
&\quad - (1 - \beta_k)\beta_{k-1}(U^0 - U^\star) + (1 - \beta_k)\left(\Pi_{j=1}^{k-1}(1 - \beta_j)\right)\left(\Pi_{l=k-1}^{0}\gamma\mathcal{P}^l\right)(U^0 - U^\star) \\
&\quad + \beta_k(U^0 - U^\star),
\end{aligned}
$$

and let $\bar{U}$ be the entire right hand side of inequality. Then, we have

$$
\begin{aligned}
&T^\star U^k - U^k \\
&= T^\star U^k - (1 - \beta_k)T^\star U^{k-1} - \beta_k U^0 \\
&= T^\star U^k - (1 - \beta_k)T^\star U^{k-1} - \beta_k U^\star - \beta_k(U^0 - U^\star) \\
&= T^\star U^k - (1 - \beta_k)T^\star U^{k-1} - \beta_k T^\star U^\star - \beta_k(U^0 - U^\star) \\
&\leq \gamma\mathcal{P}^k\Bigg((1 - \beta_k)\sum_{i=1}^{k-1}\left[(\beta_i - \beta_{i-1}(1 - \beta_i))\left(\Pi_{j=i+1}^{k-1}(1 - \beta_j)\right)\left(\Pi_{l=k-1}^{i}\gamma\mathcal{P}^l\right)(U^0 - U^\star)\right] \\
&\quad - (1 - \beta_k)\beta_{k-1}(U^0 - U^\star) + (1 - \beta_k)\left(\Pi_{j=1}^{k-1}(1 - \beta_j)\right)\left(\Pi_{l=k-1}^{0}\gamma\mathcal{P}^l\right)(U^0 - U^\star) \\
&\quad + \beta_k(U^0 - U^\star)\Bigg) - \beta_k(U^0 - U^\star) \\
&= \sum_{i=1}^{k-1}\left[(\beta_i - \beta_{i-1}(1 - \beta_i))\left(\Pi_{j=i+1}^{k}(1 - \beta_j)\right)\left(\Pi_{l=k}^{i}\gamma\mathcal{P}^l\right)(U^0 - U^\star)\right] \\
&\quad - \beta_{k-1}(1 - \beta_k)\gamma\mathcal{P}^k(U^0 - U^\star) + \beta_k\gamma\mathcal{P}^k\left(U^0 - U^\star\right) \\
&\quad - \beta_k(U^0 - U^\star) + \left(\Pi_{j=1}^{k}(1 - \beta_j)\right)\left(\Pi_{l=k}^{0}\gamma\mathcal{P}^l\right)(U^0 - U^\star) \\
&= \sum_{i=1}^{k}\left[(\beta_i - \beta_{i-1}(1 - \beta_i))\left(\Pi_{j=i+1}^{k}(1 - \beta_j)\right)\left(\Pi_{l=k}^{i}\gamma\mathcal{P}^l\right)(U^0 - U^\star)\right] \\
&\quad - \beta_k(U^0 - U^\star) + \left(\Pi_{j=1}^{k}(1 - \beta_j)\right)\left(\Pi_{l=k}^{0}\gamma\mathcal{P}^l\right)(U^0 - U^\star).
\end{aligned}
$$

where inequality comes from first inequality in Lemma 8 with $\alpha = \beta_k, U = U^k, \tilde{U} = U^{k-1}$, and previously defined $\bar{U}$. $\qquad\square$

*Proof of Lemma 11.* Note that $\hat{T}^\star$ is Bellman anti-optimality operators for $V$ or $Q$, and $\hat{U}^\star$ is the fixed point of $\hat{T}^\star$. If $k = 0$,

$$
\begin{aligned}
T^\star U^0 - U^0 &= T^\star U^0 - \hat{U}^\star - (U^0 - \hat{U}^\star) \\
&= T^\star U^0 - \hat{T}^\star \hat{U}^\star - (U^0 - \hat{U}^\star) \\
&\geq \gamma\hat{\mathcal{P}}^0(U^0 - \hat{U}^\star) - (U^0 - \hat{U}^\star).
\end{aligned}
$$

where inequality comes from second inequality in Lemma 9 with $\alpha = 1, U = U^0, \bar{U} = U^0 - \hat{U}^\star$.

By induction,

$$U^k - (1 - \beta_k)U^{k-1} - \beta_k \hat{U}^\star$$
$$= \beta_k(U^0 - \hat{U}^\star) + (1 - \beta_k)(T^\star U^{k-1} - U^{k-1})$$
$$\geq (1 - \beta_k) \sum_{i=1}^{k-1} \left[ (\beta_i - \beta_{i-1}(1 - \beta_i)) \left( \Pi_{j=i+1}^{k-1}(1 - \beta_j) \right) \left( \Pi_{l=k-1}^{i} \gamma \hat{\mathcal{P}}^l \right) (U^0 - \hat{U}^\star) \right]$$
$$- (1 - \beta_k)\beta_{k-1}(U^0 - \hat{U}^\star) + (1 - \beta_k) \left( \Pi_{j=1}^{k-1}(1 - \beta_j) \right) \left( \Pi_{l=k-1}^{0} \gamma \hat{\mathcal{P}}^l \right) (U^0 - \hat{U}^\star)$$
$$+ \beta_k(U^0 - \hat{U}^\star),$$

and let $\bar{U}$ be the entire right hand side of inequality. Then, we have

$$T^\star U^k - U^k$$
$$= T^\star U^k - (1 - \beta_k)T^\star U^{k-1} - \beta_k U^0$$
$$= T^\star U^k - (1 - \beta_k)T^\star U^{k-1} - \beta_k \hat{U}^\star - \beta_k(U^0 - \hat{U}^\star)$$
$$= T^\star U^k - (1 - \beta_k)T^\star U^{k-1} - \beta_k \hat{T}^\star \hat{U}^\star - \beta_k(U^0 - \hat{U}^\star)$$
$$\geq \gamma \hat{\mathcal{P}}^k \left( (1 - \beta_k) \sum_{i=1}^{k-1} \left[ (\beta_i - \beta_{i-1}(1 - \beta_i)) \left( \Pi_{j=i+1}^{k-1}(1 - \beta_j) \right) \left( \Pi_{l=k-1}^{i} \gamma \hat{\mathcal{P}}^l \right) (U^0 - \hat{U}^\star) \right] \right.$$
$$- (1 - \beta_k)\beta_{k-1}(U^0 - \hat{U}^\star) + (1 - \beta_k) \left( \Pi_{j=1}^{k-1}(1 - \beta_j) \right) \left( \Pi_{l=k-1}^{0} \gamma \hat{\mathcal{P}}^l \right) (U^0 - \hat{U}^\star)$$
$$\left. + \beta_k(U^0 - \hat{U}^\star) \right) - \beta_k(U^0 - \hat{U}^\star)$$
$$= \sum_{i=1}^{k-1} \left[ (\beta_i - \beta_{i-1}(1 - \beta_i)) \left( \Pi_{j=i+1}^{k}(1 - \beta_j) \right) \left( \Pi_{l=k}^{i} \gamma \hat{\mathcal{P}}^l \right) (U^0 - \hat{U}^\star) \right]$$
$$- \beta_{k-1}(1 - \beta_k)\gamma \hat{\mathcal{P}}^k(U^0 - \hat{U}^\star) + \beta_k \gamma \hat{\mathcal{P}}^k \left( U^0 - \hat{U}^\star \right)$$
$$- \beta_k(U^0 - \hat{U}^\star) + \left( \Pi_{j=1}^{k}(1 - \beta_j) \right) \left( \Pi_{l=k}^{0} \gamma \hat{\mathcal{P}}^l \right) (U^0 - \hat{U}^\star)$$
$$= \sum_{i=1}^{k} \left[ (\beta_i - \beta_{i-1}(1 - \beta_i)) \left( \Pi_{j=i+1}^{k}(1 - \beta_j) \right) \left( \Pi_{l=k}^{i} \gamma \hat{\mathcal{P}}^l \right) (U^0 - \hat{U}^\star) \right]$$
$$- \beta_k(U^0 - \hat{U}^\star) + \left( \Pi_{j=1}^{k}(1 - \beta_j) \right) \left( \Pi_{l=k}^{0} \gamma \hat{\mathcal{P}}^l \right) (U^0 - \hat{U}^\star).$$

where inequality comes from second inequality in Lemma 9 with $\alpha = \beta_k, U = U^k, \tilde{U} = U^{k-1}$, and previously defined $\bar{U}$. $\qquad\square$

Now, we prove the first rate of Theorem 2.

*Proof of first rate in Theorem 2.* Since $B_1 \leq A \leq B_2$ implies $\|A\|_\infty \leq \sup\{\|B_1\|_\infty, \|B_2\|_\infty\}$ for $A, B \in \mathcal{F}(\mathcal{X})$, if we take $\|\cdot\|_\infty$ right side first inequality of Lemma 10, we have

$$
\sum_{i=1}^{k} |\beta_i - \beta_{i-1}(1 - \beta_i)| \left(\Pi_{j=i+1}^{k}(1 - \beta_j)\right) \left\|\left(\Pi_{l=k}^{i}\gamma\mathcal{P}^l\right)(U^0 - U^\star)\right\|_\infty
$$
$$
+ \beta_k \left\|U^0 - U^\pi\right\|_\infty + \left(\Pi_{j=1}^{k}(1 - \beta_j)\right) \left\|\left(\Pi_{l=k}^{0}\gamma\mathcal{P}^l\right)(U^0 - U^\star)\right\|_\infty
$$
$$
\leq \left(\sum_{i=1}^{k} \gamma^{k-i+1} |\beta_i - \beta_{i-1}(1 - \beta_i)| \left(\Pi_{j=i+1}^{k}(1 - \beta_j)\right) + \beta_k + \gamma^{k+1}\Pi_{j=1}^{k}(1 - \beta_j)\right)
$$
$$
\left\|U^0 - U^\star\right\|_\infty
$$
$$
= \frac{\left(\gamma^{-1} - \gamma\right)\left(1 + 2\gamma - \gamma^{k+1}\right)}{\left(\gamma^{k+1}\right)^{-1} - \gamma^{k+1}} \left\|U^0 - U^\star\right\|_\infty,
$$

where the first inequality comes from triangular inequality, second inequality is from nonexpansiveness of $\mathcal{P}^l$, and last equality comes from calculations.

If we take $\|\cdot\|_\infty$ right side of second inequality of Lemma 10, similarly, we have

$$
\sum_{i=1}^{k} |\beta_i - \beta_{i-1}(1 - \beta_i)| \left(\Pi_{j=i+1}^{k}(1 - \beta_j)\right) \left\|\left(\Pi_{l=k}^{i}\gamma\hat{\mathcal{P}}^l\right)(U^0 - \hat{U}^\star)\right\|_\infty
$$
$$
+ \beta_k \left\|U^0 - U^\pi\right\|_\infty + \left(\Pi_{j=1}^{k}(1 - \beta_j)\right) \left\|\left(\Pi_{l=k}^{0}\gamma\hat{\mathcal{P}}^l\right)(U^0 - \hat{U}^\star)\right\|_\infty
$$
$$
\leq \left(\sum_{i=1}^{k} \gamma^{k-i+1} |\beta_i - \beta_{i-1}(1 - \beta_i)| \left(\Pi_{j=i+1}^{k}(1 - \beta_j)\right) + \beta_k + \gamma^{k+1}\Pi_{j=1}^{k}(1 - \beta_j)\right)
$$
$$
= \frac{\left(\gamma^{-1} - \gamma\right)\left(1 + 2\gamma - \gamma^{k+1}\right)}{\left(\gamma^{k+1}\right)^{-1} - \gamma^{k+1}} \left\|U^0 - \hat{U}^\star\right\|_\infty,
$$

where the first inequality comes from triangular inequality, second inequality is from from nonexpansiveness of $\hat{\mathcal{P}}^l$, and last equality comes from calculations. Therefore, we conclude

$$
\left\|T^\star U^k - U^k\right\|_\infty \leq \frac{\left(\gamma^{-1} - \gamma\right)\left(1 + 2\gamma - \gamma^{k+1}\right)}{\left(\gamma^{k+1}\right)^{-1} - \gamma^{k+1}} \max\left\{\left\|U^0 - U^\star\right\|_\infty, \left\|U^0 - \hat{U}^\star\right\|_\infty\right\}.
$$

$\square$

Next, for the second rate in Theorem 2, we prove following lemmas by induction.

**Lemma 12.** *Let $0 < \gamma \leq 1$. If $\gamma = 1$, assume a fixed point $U^\star$ exists. For the iterates $\{U^k\}_{k=0,1,\ldots}$ of Anc-VI, if $T^\star U^0 \leq U^\star$, there exist nonexpansive linear operators $\{\mathcal{P}^l\}_{l=0,1,\ldots,k}$ such that*

$$
T^\star U^k - U^k \leq \sum_{i=1}^{k} \left[(\beta_i - \beta_{i-1}(1 - \beta_i))\left(\Pi_{j=i+1}^{k}(1 - \beta_j)\right)\left(\Pi_{l=k}^{i}\gamma\mathcal{P}^l\right)(U^0 - U^\star)\right] - \beta_k(U^0 - U^\star)
$$

*where $\Pi_{j=k+1}^{k}(1 - \beta_j) = 1$ and $\beta_0 = 1$.*

**Lemma 13.** *Let $0 < \gamma < 1$. For the iterates $\{U^k\}_{k=0,1,\ldots}$ of Anc-VI, if $U^0 \geq T^\star U^0$, there exist nonexpansive linear operators $\{\hat{\mathcal{P}}^l\}_{l=0,1,\ldots,k}$ such that*

$$
T^\star U^k - U^k \geq \sum_{i=1}^{k} \left[(\beta_i - \beta_{i-1}(1 - \beta_i))\left(\Pi_{j=i+1}^{k}(1 - \beta_j)\right)\left(\Pi_{l=k}^{i}\gamma\hat{\mathcal{P}}^l\right)(U^0 - \hat{U}^\star)\right] - \beta_k(U^0 - \hat{U}^\star),
$$

*where $\Pi_{j=k+1}^{k}(1 - \beta_j) = 1$ and $\beta_0 = 1$.*

*Proof of Lemma 12.* If $k = 0$,

$$
T^\star U^0 - U^0 = T^\star U^0 - U^\star - (U^0 - U^\star)
$$
$$
\leq -(U^0 - U^\star)
$$

where the second inequality is from the condition.

By induction,

$$U^k - (1 - \beta_k)U^{k-1} - \beta_k U^\star$$

$$\leq (1 - \beta_k) \sum_{i=1}^{k-1} \left[ (\beta_i - \beta_{i-1}(1 - \beta_i)) \left( \Pi_{j=i+1}^{k-1}(1 - \beta_j) \right) \left( \Pi_{l=k-1}^{i} \gamma \mathcal{P}^l \right) (U^0 - U^\star) \right]$$

$$- (1 - \beta_k)\beta_{k-1}(U^0 - U^\star) + \beta_k(U^0 - U^\star),$$

and let $\bar{U}$ be the entire right hand side of inequality. Then, we have

$$T^\star U^k - U^k$$

$$= T^\star U^k - (1 - \beta_k)T^\star U^{k-1} - \beta_k T^\star U^\star - \beta_k(U^0 - U^\star)$$

$$\leq \gamma \mathcal{P}^k \left( (1 - \beta_k) \sum_{i=1}^{k-1} \left[ (\beta_i - \beta_{i-1}(1 - \beta_i)) \left( \Pi_{j=i+1}^{k-1}(1 - \beta_j) \right) \left( \Pi_{l=k-1}^{i} \gamma \mathcal{P}^l \right) (U^0 - U^\star) \right] \right.$$

$$\left. - (1 - \beta_k)\beta_{k-1}(U^0 - U^\star) + \beta_k(U^0 - U^\star) \right) - \beta_k(U^0 - U^\star)$$

$$= \sum_{i=1}^{k} \left[ (\beta_i - \beta_{i-1}(1 - \beta_i)) \left( \Pi_{j=i+1}^{k}(1 - \beta_j) \right) \left( \Pi_{l=k}^{i} \gamma \mathcal{P}^l \right) (U^0 - U^\star) \right] - \beta_k(U^0 - U^\star),$$

where inequality comes from first inequality in Lemma 8 with $\alpha = \beta_k, U = U^k, \tilde{U} = U^{k-1}$, and previously defined $\bar{U}$. $\qquad \square$

*Proof of Lemma 13.* If $k = 0$,

$$T^\star U^0 - U^0 = T^\star U^0 - \hat{U}^\star - (U^0 - \hat{U}^\star)$$

$$\geq -(U^0 - \hat{U}^\star).$$

where the second inequality is from the fact that $U^0 \geq T^\star U^0$ implies $T^\star U^0 \geq U^\star$ by Lemma 5 and $U^\star \geq \hat{U}^\star$ by Lemma 3.

By induction,

$$U^k - (1 - \beta_k)U^{k-1} - \beta_k \hat{U}^\star$$

$$\geq (1 - \beta_k) \sum_{i=1}^{k-1} \left[ (\beta_i - \beta_{i-1}(1 - \beta_i)) \left( \Pi_{j=i+1}^{k-1}(1 - \beta_j) \right) \left( \Pi_{l=k-1}^{i} \gamma \hat{\mathcal{P}}^l \right) (U^0 - \hat{U}^\star) \right]$$

$$- (1 - \beta_k)\beta_{k-1}(U^0 - \hat{U}^\star) + \beta_k(U^0 - \hat{U}^\star),$$

and let $\bar{U}$ be the entire right hand side of inequality. Then, we have

$$T^\star U^k - U^k$$

$$= T^\star U^k - (1 - \beta_k)T^\star U^{k-1} - \beta_k \hat{T}^\star \hat{U}^\star - \beta_k(U^0 - \hat{U}^\star)$$

$$\geq \gamma \hat{\mathcal{P}}^k \left( (1 - \beta_k) \sum_{i=1}^{k-1} \left[ (\beta_i - \beta_{i-1}(1 - \beta_i)) \left( \Pi_{j=i+1}^{k-1}(1 - \beta_j) \right) \left( \Pi_{l=k-1}^{i} \gamma \hat{\mathcal{P}}^l \right) (U^0 - \hat{U}^\star) \right] \right.$$

$$\left. - (1 - \beta_k)\beta_{k-1}(U^0 - \hat{U}^\star) + \beta_k(U^0 - \hat{U}^\star) \right) - \beta_k(U^0 - \hat{U}^\star)$$

$$= \sum_{i=1}^{k} \left[ (\beta_i - \beta_{i-1}(1 - \beta_i)) \left( \Pi_{j=i+1}^{k}(1 - \beta_j) \right) \left( \Pi_{l=k}^{i} \gamma \hat{\mathcal{P}}^l \right) (U^0 - \hat{U}^\star) \right] - \beta_k(U^0 - \hat{U}^\star),$$

where inequality comes from second inequality in Lemma 9 with $\alpha = \beta_k, U = U^k, \tilde{U} = U^{k-1}$, and previously defined $\bar{U}$. $\qquad \square$

Now, we prove the second rates of Theorem 2.

*Proof of second rates in Theorem 2.* Let $0 < \gamma < 1$. Then, if $U^0 \le T^\star U^0$, then $T^\star U^0 \le U^\star$ and $U^k \le T^\star U^k$ by Lemma 5. Hence, taking $\|\cdot\|_\infty$-norm both sides of first inequality in Lemma 12, we have

$$\left\| T^\star U^k - U^k \right\|_\infty \le \frac{\left(\gamma^{-1} - \gamma\right)\left(1 + \gamma - \gamma^{k+1}\right)}{\left(\gamma^{k+1}\right)^{-1} - \gamma^{k+1}} \left\| U^0 - U^\star \right\|_\infty.$$

Otherwise, if $U^0 \ge TU^0$, $U^k \ge TU^k$ by Lemma 5. taking $\|\cdot\|_\infty$-norm both sides of second inequality in Lemma 13, we have

$$\left\| T^\star U^k - U^k \right\|_\infty \le \frac{\left(\gamma^{-1} - \gamma\right)\left(1 + \gamma - \gamma^{k+1}\right)}{\left(\gamma^{k+1}\right)^{-1} - \gamma^{k+1}} \left\| U^0 - \hat{U}^\star \right\|_\infty.$$

$\square$

## C  Omitted proofs in Section 3

First, we present the following lemma.

**Lemma 14.** *Let $\gamma = 1$. Assume a fixed point $U^\star$ exists. For the iterates $\{U^k\}_{k=0,1,\dots}$ of Anc-VI,* $\left\| U^k - U^\star \right\|_\infty \le \left\| U^0 - U^\star \right\|_\infty.$

*Proof.* If $k = 0$, it is obvious. By induction,

$$\begin{aligned}
\left\| U^k - U^\star \right\|_\infty &= \left\| \beta_k U^0 + (1 - \beta_k) TU^{k-1} - U^\star \right\|_\infty \\
&= \left\| (1 - \beta_k)(TU^{k-1} - U^\star) + \beta_k \left(U^0 - U^\star\right) \right\|_\infty \\
&\le (1 - \beta_k) \left\| TU^{k-1} - U^\star \right\|_\infty + \beta_k \left\| U^0 - U^\star \right\|_\infty \\
&\le (1 - \beta_k) \left\| U^{k-1} - U^\star \right\|_\infty + \beta_k \left\| U^0 - U^\star \right\|_\infty \\
&= \left\| U^0 - U^\star \right\|_\infty
\end{aligned}$$

where the second inequality comes form nonexpansiveness of $T$.  $\square$

Now, we present the proof of Theorem 3.

*Proof of Theorem 3.* First, if $U^0 \le TU^0$, with same argument in proof of Lemma 5, we can show that $U^{k-1} \le U^k \le TU^{k-1} \le TU^k$ for $k = 1, 2, \dots$.

Since fixed point $U^\star$ exists by assumption, Lemma 4 and 10 hold. Note that $\gamma = 1$ implies $\beta_k = \frac{1}{k+1}$ and if we take $\|\cdot\|_\infty$-norm both sides for those inequalities in lemmas, by simple calculation, we have

$$\left\| TU^k - U^k \right\|_\infty \le \frac{2}{k+1} \left\| U^0 - U^\star \right\|_\infty$$

for any fixed point $U^\star$ (since $0 \le T^\star U^k - U^k$, we can get upper bound of $\left\| T^\star U^k - U^k \right\|_\infty$ from Lemma 10).

Suppose that there exist $\{k_j\}_{j=0,1,\dots}$ such that $U^{k_j}$ converges to some $\tilde{U}^\star$. Then, $\lim_{j\to\infty}(T - I)U^{k_j} = (T - I)\tilde{U}^\star = 0$ since $T - I$ is continuous. This implies that $\tilde{U}^\star$ is a fixed point. By Lemma 14 and previous argument, $U^k$ is increasing and bounded sequence in $\mathbb{R}^n$. Thus, $U^k$ has single limit point, some fixed point $\tilde{U}^\star$. Furthermore, the fact that $U^0 \le TU^0 \le \tilde{U}^\star$ implies that Lemma 6 and 12 hold. Therefore, we have

$$\left\| TU^k - U^k \right\|_\infty \le \frac{1}{k+1} \left\| U^0 - \tilde{U}^\star \right\|_\infty.$$

$\square$

Next, we prove the Theorem 4.

*Proof of Theorem 4.* By same argument in the proof of Theorem 3, if $U^0 \leq TU^0$, we can show that $U^{k-1} \leq U^k \leq TU^{k-1} \leq TU^k$ for $k = 1, 2, \ldots$, and

$$\left\|TU^k - U^k\right\|_\infty \leq \frac{2}{k+1}\left\|U^0 - U^\star\right\|_\infty$$

for any fixed point $U^\star$. Since $U^k$ is increasing and bounded by Lemma 14 and previous argument, $U^k$ converges pointwise to some $\tilde{U}^\star$ in general action-state space. We now show that $TU^k$ also converges pointwise to $T\tilde{U}^\star$. First, let $T$ be Bellman consistency operator and $U = V, \tilde{U}^\star = \tilde{V}^\pi$. By monontone convergence theorem,

$$\begin{aligned}
\lim_{k\to\infty} T^\pi V^k(s) &= \lim_{k\to\infty} \mathbb{E}_{a\sim\pi(\cdot\,|\,s)}\left[\mathbb{E}_{s'\sim P(\cdot\,|\,s,a)}\left[r(s,a) + \gamma V^k(s')\right]\right] \\
&= \mathbb{E}_{a\sim\pi(\cdot\,|\,s)}\left[\lim_{k\to\infty} \mathbb{E}_{s'\sim P(\cdot\,|\,s,a)}\left[r(s,a) + \gamma V^k(s')\right]\right] \\
&= \mathbb{E}_{a\sim\pi(\cdot\,|\,s)}\left[\mathbb{E}_{s'\sim P(\cdot\,|\,s,a)}\left[r(s,a) + \gamma \lim_{k\to\infty} V^k(s')\right]\right] \\
&= T^\pi \tilde{V}^\pi(s)
\end{aligned}$$

for any fixed $s \in \mathcal{S}$. With same argument, case $U = Q$ also holds. If $T$ is Bellman optimality operator, we use following lemma.

**Lemma 15.** *Let $W, W^k \in \mathcal{F}(\mathcal{X})$ for $k = 0, 1, \ldots$. If $W^k(x) \leq W^{k+1}(x)$ for all $x \in \mathcal{X}$, and $\{W^k\}_{k=0,1,\ldots}$, converge pointwise to $W$, then $\lim_{k\to\infty}\{\sup_x W^k(x)\} = \sup_x W(x)$.*

*Proof.* $W^k(x) \leq W(x)$ implies that $\sup_x W^k(x) \leq \sup_x W(x)$. If $\sup_x W(x) = a$, there exist $x$ which satisfying $a - W(x) < \frac{\epsilon}{2}$, and by definition of $W$, there exist $W^k$ such that $a - W^k(x) < \epsilon$ for any $\epsilon > 0$. $\square$

If $U = V$ and $\tilde{U}^\star = \tilde{V}^\star$, by previous lemma and monotone convergence theorem, we have

$$\begin{aligned}
\lim_{k\to\infty} T^\star V^k(s) &= \lim_{k\to\infty} \sup_a \left\{\mathbb{E}_{s'\sim P(\cdot\,|\,s,a)}\left[r(s,a) + \gamma V^k(s')\right]\right\} \\
&= \sup_a \left\{\lim_{k\to\infty} \mathbb{E}_{s'\sim P(\cdot\,|\,s,a)}\left[r(s,a) + \gamma V^k(s')\right]\right\} \\
&= \sup_a \left\{\mathbb{E}_{s'\sim P(\cdot\,|\,s,a)}\left[r(s,a) + \gamma \lim_{k\to\infty} V^k(s')\right]\right\} \\
&= T^\star \tilde{U}^\star(s)
\end{aligned}$$

for any fixed $s \in \mathcal{S}$. With similar argument, case $U = Q$ also holds.

Since $TU^k \to T\tilde{U}^\star$ and $U^k \to \tilde{U}^\star$ pointwisely, $TU^k - U^k$ converges pointwise to $T\tilde{U}^\star - \tilde{U}^\star = 0$. Thus, $\tilde{U}^\star$ is indeed fixed point of $T$. Furthermore, the fact that $U^0 \leq TU^0 \leq \tilde{U}^\star$ implies that Lemma 6 and 12 hold. Therefore, we have

$$\left\|TU^k - U^k\right\|_\infty \leq \frac{1}{k+1}\left\|U^0 - \tilde{U}^\star\right\|_\infty.$$

$\square$

# D   Omitted proofs in Section 4

We present the proof of Theorem 5.

*Proof of Theorem 5.* First, we prove the case $U^0 = 0$ for $n \geq k+2$. Consider the MDP $(\mathcal{S}, \mathcal{A}, P, r, \gamma)$ such that

$$\mathcal{S} = \{s_1, \ldots, s_n\}, \quad \mathcal{A} = \{a_1\}, \quad P(s_i\,|\,s_j, a_1) = \mathbb{1}_{\{i=j=1,\,j=i+1\}}, \quad r(s_i, a_1) = \mathbb{1}_{\{i=2\}}.$$

Then, $T = \gamma \mathcal{P}^\pi U + [0, 1, 0, \ldots, 0]^\intercal, U^\star = [0, 1, \gamma, \ldots, \gamma^{n-2}]^\intercal$, and $\left\|U^0 - U^\star\right\|_\infty = 1$. Under the span condition, we can show that $(U^k)_1 = (U^k)_l = 0$ for $k + 2 \leq l \leq n$ by following lemma.

**Lemma 16.** *Let $T \colon \mathbb{R}^n \to \mathbb{R}^n$ be defined as before. Then, under span condition, $\left(U^i\right)_1 = 0$ for $0 \le i \le k$, and $\left(U^i\right)_j = 0$ for $0 \le i \le k$ and $i + 2 \le j \le n$.*

*Proof.* Case $k = 0$ is obvious. By induction, $\left(U^l\right)_1 = 0$ for $0 \le l \le i - 1$. Then $\left(TU^l\right)_1 = 0$ for $0 \le l \le i - 1$. This implies that $\left(TU^l - U^l\right)_1 = 0$ for $0 \le l \le i - 1$. Hence $\left(U^i\right)_1 = \left(U^0\right)_1 = 0$. Again, by induction, $\left(U^l\right)_j = 0$ for $0 \le l \le i-1, l+2 \le j \le n$. Then $\left(TU^l\right)_j = 0$ for $0 \le l \le i-1$, $l + 3 \le j \le n$ and this implies that $\left(TU^l - U^l\right)_j = 0$ for $0 \le l \le i - 1, l + 3 \le j \le n$. Therefore, $\left(U^i\right)_j = 0$ for $i + 2 \le j \le n$. $\qquad\square$

Then, we get

$$TU^k - U^k = \left(0, 1 - \left(U^k\right)_2, \gamma \left(U^k\right)_2 - \left(U^k\right)_3, \ldots, \gamma \left(U^k\right)_k - \left(U^k\right)_{k+1}, \gamma \left(U^k\right)_{k+1}, \underbrace{0, \ldots, 0}_{n-k-2}\right),$$

and this implies

$$\left(TU^k - U^k\right)_2 + \gamma^{-1}\left(TU^k - U^k\right)_3 + \cdots + \gamma^{-k}\left(TU^k - U^k\right)_{k+2} = 1.$$

Taking the absolute value on both sides,

$$\left(1 + \cdots + \gamma^{-k}\right) \max_{1 \le i \le n} \left\{|TU^k - U^k|_i\right\} \ge 1.$$

Therefore, we conclude

$$\|TU^k - U^k\|_\infty \ge \frac{\gamma^k}{\sum_{i=0}^{k} \gamma^i} \|U^0 - U^\star\|_\infty.$$

Now, we show that for any initial point $U^0 \in \mathbb{R}^n$, there exists an MDP which exhibits same lower bound with the case $U^0 = 0$. Denote by MDP(0) and $T_0$ the worst-case MDP and Bellman consistency or opitmality operator constructed for $U^0 = 0$. Define an MDP($U^0$) $(\mathcal{S}, \mathcal{A}, P, r, \gamma)$ for $U^0 \neq 0$ as

$$\mathcal{S} = \{s_1, \ldots, s_n\}, \ \mathcal{A} = \{a_1\}, \ P(s_i \,|\, s_j, a_1) = \mathbb{1}_{\{i=j=1, \, j=i+1\}}, \ r(s_i, a_1) = \left(U^0 - \mathcal{P}^\pi U^0\right)_i + \mathbb{1}_{\{i=2\}}.$$

Then, Bellman consistency or optimality operator $T$ satisfies

$$TU = T_0(U - U^0) + U^0.$$

Let $\tilde{U}^\star$ be fixed point of $T_0$. Then, if $U^\star = \tilde{U}^\star + U^0$, $U^\star$ is fixed point of $T$. Furthermore, if $\{U^i\}_{i=0}^k$ satisfies span condition

$$U^i \in U^0 + span\{TU^0 - U^0, TU^1 - U^1, \ldots, TU^{i-1} - U^{i-1}\}, \qquad i = 1, \ldots, k,$$

$\tilde{U}^i = U^i - U^0$ is a sequence satisfying

$$\tilde{U}^i \in \underbrace{\tilde{U}^0}_{=0} + span\{T_0\tilde{U}^0 - \tilde{U}^0, T_0\tilde{U}^1 - \tilde{U}^1, \ldots, T_0\tilde{U}^{i-1} - \tilde{U}^{i-1}\}, \qquad i = 1, \ldots, k,$$

which is the same span condition in Theorem 5 with respect to $T_0$. This is because

$$TU^i - U^i = T_0(U^i - U^0) - (U^i - U^0) = T\tilde{U}^i - \tilde{U}^i$$

for $i = 0, \ldots, k$. Thus, $\{\tilde{U}^i\}_{i=0}^k$ is a sequence starting from 0 and satisfy the span condition for $T_0$. This implies that

$$\|TU^k - U^k\|_\infty = \left\|T\tilde{U}^k - \tilde{U}^k\right\|_\infty$$

$$\ge \frac{\gamma^k}{\sum_{i=0}^{k} \gamma^i} \left\|\tilde{U}^0 - \tilde{U}^\star\right\|_\infty$$

$$= \frac{\gamma^k}{\sum_{i=0}^{k} \gamma^i} \|U^0 - U^\star\|_\infty.$$

Hence, MDP($U^0$) is indeed our desired worst-case instance. Lastly, the fact that $U^0 - U^\star = \tilde{U}^0 - \tilde{U}^\star = -(0, 1, \gamma, \ldots, \gamma^{n-2})$ implies $U^0 \le U^\star$.

$$\square$$

# E    Omitted proofs in Section 5

First, we prove following key lemma.

**Lemma 17.** *Let $0 < \gamma < 1$. For the iterates $\{U^k\}_{k=0,1,\dots}$ of Anc-VI, there exist nonexpansive linear operators $\{\mathcal{P}^l\}_{l=0,1,\dots,k}$ and $\{\hat{\mathcal{P}}^l\}_{l=0,1,\dots,k}$ such that*

$$T^\star U^k - U^k \leq \sum_{i=1}^{k} \left[ (\beta_i - \beta_{i-1}(1-\beta_i)) \left( \Pi_{j=i+1}^k (1-\beta_j) \right) \left( \Pi_{l=k}^i \gamma \mathcal{P}^l \right) (U^0 - U^\star) \right] - \beta_k (U^0 - U^\star)$$

$$+ \Pi_{j=1}^k (1-\beta_j) \Pi_{l=k}^0 \gamma \mathcal{P}^l (U^0 - U^\star) - \sum_{i=1}^{k} \Pi_{j=i}^k (1-\beta_j) \Pi_{l=k}^{i+1} \gamma \mathcal{P}^l \left( I - \gamma \mathcal{P}^i \right) \epsilon^{i-1},$$

$$T^\star U^k - U^k \geq \sum_{i=1}^{k} \left[ (\beta_i - \beta_{i-1}(1-\beta_i)) \left( \Pi_{j=i+1}^k (1-\beta_j) \right) \left( \Pi_{l=k}^i \gamma \hat{\mathcal{P}}^l \right) (U^0 - \hat{U}^\star) \right] - \beta_k (U^0 - \hat{U}^\star)$$

$$+ \Pi_{j=1}^k (1-\beta_j) \Pi_{l=k}^0 \gamma \hat{\mathcal{P}}^l (U^0 - \hat{U}^\star) - \sum_{i=1}^{k} \Pi_{j=i}^k (1-\beta_j) \Pi_{l=k}^{i+1} \gamma \hat{\mathcal{P}}^l \left( I - \gamma \hat{\mathcal{P}}^i \right) \epsilon^{i-1},$$

*for $1 \leq k$, where $\Pi_{j=k+1}^k (1-\beta_j) = 1$, $\Pi_{l=k}^{k+1} \gamma \mathcal{P}^l = \Pi_{l=k}^{k+1} \gamma \hat{\mathcal{P}}^l = I$, and $\beta_0 = 1$.*

*Proof of Lemma 17.* First, we prove the first inequality in Lemma 17 by induction.

If $k = 1$,

$$U^1 - (1-\beta_1)U^0 - \beta_1 U^\star = (1-\beta_1)\epsilon^0 + \beta_1(U^0 - U^\star) + (1-\beta_1)(T^\star U^0 - U^0)$$
$$\leq (1-\beta_1)\epsilon^0 + (1-\beta_1)\gamma \mathcal{P}^0 (U^0 - U^\star) + (2\beta_1 - 1)(U^0 - U^\star),$$

where inequality comes from Lemma 8 with $\alpha = 1, U = U^0, \bar{U} = U^0 - U^\star$, and let $\bar{U}$ be the entire right hand side of inequality. Then, we have

$$T^\star U^1 - U^1 = T^\star U^1 - (1-\beta_1)T^\star U^0 - \beta_1 U^\star - \beta_1(U^0 - U^\star) - (1-\beta_1)\epsilon^0$$
$$\leq \gamma \mathcal{P}^1 ((1-\beta_1)\epsilon^0 + (1-\beta_1)\gamma \mathcal{P}^0 (U^0 - U^\star) + (2\beta_1 - 1)(U^0 - U^\star)) - \beta_1(U^0 - U^\star)$$
$$- (1-\beta_1)\epsilon^0$$
$$= (1-\beta_1)\gamma \mathcal{P}^1 \gamma \mathcal{P}^0 (U^0 - U^\star) + \gamma \mathcal{P}^1 (2\beta_1 - 1)(U^0 - U^\star) - \beta_1(U^0 - U^\star)$$
$$- (I - \gamma \mathcal{P}^1)(1-\beta_1)\epsilon^0.$$

where inequality comes from Lemma 8 with $\alpha = \beta_1, U = U^1, \tilde{U} = U^0$, and previously defined $\bar{U}$.

By induction,

$$U^k - (1-\beta_k)U^{k-1} - \beta_k U^\star$$
$$= \beta_k \left( U^0 - U^\star \right) + (1-\beta_k)(T^\star U^{k-1} - U^{k-1}) + (1-\beta_k)\epsilon^{k-1}$$
$$\leq (1-\beta_k) \sum_{i=1}^{k-1} \left[ (\beta_i - \beta_{i-1}(1-\beta_i)) \left( \Pi_{j=i+1}^{k-1} (1-\beta_j) \right) \left( \Pi_{l=k-1}^i \gamma \mathcal{P}^l \right) (U^0 - U^\star) \right]$$
$$- (1-\beta_k)\beta_{k-1}(U^0 - U^\star) + (1-\beta_k) \left( \Pi_{j=1}^{k-1} (1-\beta_j) \right) \left( \Pi_{l=k-1}^0 \gamma \mathcal{P}^l \right) (U^0 - U^\star)$$
$$+ \beta_k(U^0 - U^\star) - (1-\beta_k) \sum_{i=1}^{k-1} \Pi_{j=i}^{k-1} (1-\beta_j) \Pi_{l=k-1}^{i+1} \gamma \mathcal{P}^l \left( I - \gamma \mathcal{P}^i \right) \epsilon^{i-1} + (1-\beta_k)\epsilon^{k-1},$$

and let $\bar{U}$ be the entire right hand side of inequality. Then, we have

$$T^\star U^k - U^k$$
$$= T^\star U^k - (1-\beta_k)T^\star U^{k-1} - \beta_k U^0 - (1-\beta_k)\epsilon^{k-1}$$
$$= T^\star U^k - (1-\beta_k)T^\star U^{k-1} - \beta_k T^\star U^\star - \beta_k(U^0 - U^\star) - (1-\beta_k)\epsilon^{k-1}$$
$$\leq \gamma\mathcal{P}^k\Bigg((1-\beta_k)\sum_{i=1}^{k-1}\left[(\beta_i - \beta_{i-1}(1-\beta_i))\left(\Pi_{j=i+1}^{k-1}(1-\beta_j)\right)\left(\Pi_{l=k-1}^{i}\gamma\mathcal{P}^l\right)(U^0 - U^\star)\right]$$
$$- (1-\beta_k)\beta_{k-1}(U^0 - U^\star) + (1-\beta_k)\left(\Pi_{j=1}^{k-1}(1-\beta_j)\right)\left(\Pi_{l=k-1}^{0}\gamma\mathcal{P}^l\right)(U^0 - U^\star)$$
$$+ \beta_k(U^0 - U^\star) - (1-\beta_k)\sum_{i=1}^{k-1}\Pi_{j=i}^{k-1}(1-\beta_j)\Pi_{l=k-1}^{i+1}\gamma\mathcal{P}^l\left(I - \gamma\mathcal{P}^i\right)\epsilon^{i-1} + (1-\beta_k)\epsilon^{k-1}\Bigg)$$
$$- \beta_k(U^0 - U^\star) - (1-\beta_k)\epsilon^{k-1}$$
$$= \sum_{i=1}^{k}\left[(\beta_i - \beta_{i-1}(1-\beta_i))\left(\Pi_{j=i+1}^{k}(1-\beta_j)\right)\left(\Pi_{l=k}^{i}\gamma\mathcal{P}^l\right)(U^0 - U^\star)\right] - \beta_k(U^0 - U^\star)$$
$$+ \Pi_{j=1}^{k}(1-\beta_j)\Pi_{l=k}^{0}\gamma\mathcal{P}^l(U^0 - U^\star) - \sum_{i=1}^{k}\Pi_{j=i}^{k}(1-\beta_j)\Pi_{l=k}^{i+1}\gamma\mathcal{P}^l\left(I - \gamma\mathcal{P}^i\right)\epsilon^{i-1},$$

where inequality comes from Lemma 8 with $\alpha = \beta_k, U = U^k, \tilde{U} = U^{k-1}$, and previously defined $\bar{U}$.

Now, we prove second inequality in Lemma 17 by induction.

If $k = 1$,

$$U^1 - (1-\beta_1)U^0 - \beta_1\hat{U}^\star = (1-\beta_1)\epsilon^0 + \beta_1(U^0 - \hat{U}^\star) + (1-\beta_1)(T^\star U^0 - U^0)$$
$$\geq (1-\beta_1)\epsilon^0 + (1-\beta_1)\gamma\hat{\mathcal{P}}^0(U^0 - \hat{U}^\star) + (2\beta_1 - 1)(U^0 - \hat{U}^\star),$$

where inequality comes from Lemma 9 with $\alpha = 1, U = U^0, \bar{U} = U^0 - \hat{U}^\star$, and let $\bar{U}$ be the entire right hand side of inequality. Then, we have

$$T^\star U^1 - U^1 = T^\star U^1 - (1-\beta_1)T^\star U^0 - \beta_1\hat{U}^\star - \beta_1(U^0 - \hat{U}^\star) - (1-\beta_1)\epsilon^0$$
$$\geq \gamma\hat{\mathcal{P}}^1((1-\beta_1)\epsilon^0 + (1-\beta_1)\gamma\hat{\mathcal{P}}^0(U^0 - \hat{U}^\star) + (2\beta_1 - 1)(U^0 - \hat{U}^\star)) - \beta_1(U^0 - \hat{U}^\star)$$
$$- (1-\beta_1)\epsilon^0$$
$$= (1-\beta_1)\gamma\hat{\mathcal{P}}^1\gamma\hat{\mathcal{P}}^0(U^0 - \hat{U}^\star) + \gamma\hat{\mathcal{P}}^1(2\beta_1 - 1)(U^0 - \hat{U}^\star) - \beta_1(U^0 - \hat{U}^\star)$$
$$- (I - \gamma\hat{\mathcal{P}}^1)(1-\beta_1)\epsilon^0.$$

where inequality comes from Lemma 9 with $\alpha = \beta_1, U = U^1, \tilde{U} = U^0$, and previously defined $\bar{U}$.

By induction,

$$U^k - (1-\beta_k)U^{k-1} - \beta_k\hat{U}^\star$$
$$= \beta_k\left(U^0 - \hat{U}^\star\right) + (1-\beta_k)(T^\star U^{k-1} - U^{k-1}) + (1-\beta_k)\epsilon^{k-1}$$
$$\geq (1-\beta_k)\sum_{i=1}^{k-1}\left[(\beta_i - \beta_{i-1}(1-\beta_i))\left(\Pi_{j=i+1}^{k-1}(1-\beta_j)\right)\left(\Pi_{l=k-1}^{i}\gamma\hat{\mathcal{P}}^l\right)(U^0 - \hat{U}^\star)\right]$$
$$- (1-\beta_k)\beta_{k-1}(U^0 - \hat{U}^\star) + (1-\beta_k)\left(\Pi_{j=1}^{k-1}(1-\beta_j)\right)\left(\Pi_{l=k-1}^{0}\gamma\hat{\mathcal{P}}^l\right)(U^0 - \hat{U}^\star)$$
$$+ \beta_k(U^0 - \hat{U}^\star) - (1-\beta_k)\sum_{i=1}^{k-1}\Pi_{j=i}^{k-1}(1-\beta_j)\Pi_{l=k-1}^{i+1}\gamma\hat{\mathcal{P}}^l\left(I - \gamma\hat{\mathcal{P}}^i\right)\epsilon^{i-1} + (1-\beta_k)\epsilon^{k-1},$$

and let $\bar{U}$ be the entire right hand side of inequality. Then, we have

$$
\begin{aligned}
& T^\star U^k - U^k \\
&= T^\star U^k - (1-\beta_k)T^\star U^{k-1} - \beta_k U^0 - (1-\beta_k)\epsilon^{k-1} \\
&= T^\star U^k - (1-\beta_k)T^\star U^{k-1} - \beta_k T^\star \hat{U}^\star - \beta_k(U^0 - \hat{U}^\star) - (1-\beta_k)\epsilon^{k-1} \\
&\geq \gamma\hat{\mathcal{P}}^k\Bigg((1-\beta_k)\sum_{i=1}^{k-1}\left[(\beta_i - \beta_{i-1}(1-\beta_i))\left(\Pi_{j=i+1}^{k-1}(1-\beta_j)\right)\left(\Pi_{l=k-1}^i\gamma\hat{\mathcal{P}}^l\right)(U^0 - \hat{U}^\star)\right] \\
&\quad - (1-\beta_k)\beta_{k-1}(U^0 - \hat{U}^\star) + (1-\beta_k)\left(\Pi_{j=1}^{k-1}(1-\beta_j)\right)\left(\Pi_{l=k-1}^0\gamma\hat{\mathcal{P}}^l\right)(U^0 - \hat{U}^\star) \\
&\quad + \beta_k(U^0 - \hat{U}^\star) - (1-\beta_k)\sum_{i=1}^{k-1}\Pi_{j=i}^{k-1}(1-\beta_j)\Pi_{l=k-1}^{i+1}\gamma\hat{\mathcal{P}}^l\left(I - \gamma\hat{\mathcal{P}}^i\right)\epsilon^{i-1} + (1-\beta_k)\epsilon^{k-1}\Bigg) \\
&\quad - \beta_k(U^0 - \hat{U}^\star) - (1-\beta_k)\epsilon^{k-1} \\
&= \sum_{i=1}^k\left[(\beta_i - \beta_{i-1}(1-\beta_i))\left(\Pi_{j=i+1}^k(1-\beta_j)\right)\left(\Pi_{l=k}^i\gamma\hat{\mathcal{P}}^l\right)(U^0 - \hat{U}^\star)\right] - \beta_k(U^0 - \hat{U}^\star) \\
&\quad + \Pi_{j=1}^k(1-\beta_j)\Pi_{l=k}^0\gamma\hat{\mathcal{P}}^l(U^0 - \hat{U}^\star) - \sum_{i=1}^k\Pi_{j=i}^k(1-\beta_j)\Pi_{l=k}^{i+1}\gamma\hat{\mathcal{P}}^l\left(I - \gamma\hat{\mathcal{P}}^i\right)\epsilon^{i-1},
\end{aligned}
$$

where inequality comes from Lemma 9 with $\alpha = \beta_k, U = U^k, \tilde{U} = U^{k-1}$, and previously defined $\bar{U}$ $\qquad\square$

Now, we prove the first rate in Theorem 6.

*Proof of first rate in Theorem 6.* Since $B_1 \leq A \leq B_2$ implies $\|A\|_\infty \leq \sup\{\|B_1\|_\infty, \|B_2\|_\infty\}$ for $A, B \in \mathcal{F}(\mathcal{X})$, if we take $\|\cdot\|_\infty$ right side of first inequality in Lemma 17, we have

$$
\begin{aligned}
& \frac{(\gamma^{-1} - \gamma)(1 + 2\gamma - \gamma^{k+1})}{(\gamma^{k+1})^{-1} - \gamma^{k+1}}\left\|U^0 - U^\star\right\|_\infty + (1+\gamma)\sum_{i=1}^k\left(\Pi_{j=i}^k(1-\beta_j)\right)\gamma^{k-i}\left\|\epsilon^{i-1}\right\|_\infty \\
&\leq \frac{(\gamma^{-1} - \gamma)(1 + 2\gamma - \gamma^{k+1})}{(\gamma^{k+1})^{-1} - \gamma^{k+1}}\left\|U^0 - U^\star\right\|_\infty + \frac{1+\gamma}{1+\gamma^{k+1}}\frac{1-\gamma^k}{1-\gamma}\max_{0\leq i\leq k-1}\left\|\epsilon^i\right\|_\infty.
\end{aligned}
$$

If we apply second inequality of Lemma 17 and take $\|\cdot\|_\infty$-norm right side, we have

$$
\frac{(\gamma^{-1} - \gamma)(1 + 2\gamma - \gamma^{k+1})}{(\gamma^{k+1})^{-1} - \gamma^{k+1}}\left\|U^0 - \hat{U}^\star\right\|_\infty + \frac{1+\gamma}{1+\gamma^{k+1}}\frac{1-\gamma^k}{1-\gamma}\max_{0\leq i\leq k-1}\left\|\epsilon^i\right\|_\infty.
$$

Therefore, we get

$$
\begin{aligned}
\left\|T^\star U^k - U^k\right\|_\infty &\leq \frac{(\gamma^{-1} - \gamma)(1 + 2\gamma - \gamma^{k+1})}{(\gamma^{k+1})^{-1} - \gamma^{k+1}}\max\left\{\left\|U^0 - U^\star\right\|_\infty, \left\|U^0 - \hat{U}^\star\right\|_\infty\right\} \\
&\quad + \frac{1+\gamma}{1+\gamma^{k+1}}\frac{1-\gamma^k}{1-\gamma}\max_{0\leq i\leq k-1}\left\|\epsilon^i\right\|_\infty.
\end{aligned}
$$

$\qquad\square$

Now, for the second rate in Theorem 6, we present following key lemma.

**Lemma 18.** *Let $0 < \gamma < 1$. For the iterates $\{U^k\}_{k=0,1,\ldots}$ of Anc-VI, if $U^0 \geq T^\star U^0$, there exist nonexpansive linear operators $\{\mathcal{P}^l\}_{l=0,1,\ldots,k}$ and $\{\hat{\mathcal{P}}^l\}_{l=0,1,\ldots,k}$ such that*

$$T^\star U^k - U^k \leq \Pi_{j=1}^k (1-\beta_j) \Pi_{l=k}^0 \gamma \mathcal{P}^l (U^0 - U^\star) - \sum_{i=1}^k \Pi_{j=i}^k (1-\beta_j) \Pi_{l=k}^{i+1} \gamma \mathcal{P}^l \left( I - \gamma \mathcal{P}^i \right) \epsilon^{i-1}$$

$$- \beta^k (U^0 - U^\star),$$

$$T^\star U^k - U^k \geq \sum_{i=1}^k \left[ (\beta_i - \beta_{i-1}(1-\beta_i)) \left( \Pi_{j=i+1}^k (1-\beta_j) \right) \left( \Pi_{l=k}^i \gamma \hat{\mathcal{P}}^l \right) (U^0 - \hat{U}^\star) \right] - \beta_k (U^0 - \hat{U}^\star)$$

$$- \sum_{i=1}^k \Pi_{j=i}^k (1-\beta_j) \Pi_{l=k}^{i+1} \gamma \hat{\mathcal{P}}^l \left( I - \gamma \hat{\mathcal{P}}^i \right) \epsilon^{i-1},$$

*for $1 \leq k$, where $\Pi_{j=k+1}^k (1-\beta_j) = 1$, $\Pi_{l=k}^{k+1} \gamma \mathcal{P}^l = \Pi_{l=k}^{k+1} \gamma \hat{\mathcal{P}}^l = I$, and $\beta_0 = 1$.*

*Proof of Lemma 18.* If $U^0 \geq T^\star U^0$, $U^0 \geq \lim_{m\to\infty} (T^\star)^m U^0 = U^\star$ by Lemma 1. By Lemma 3, this also implies $U^0 \geq \hat{U}^\star$.

First, we prove first inequality in Lemma 18 by induction. If $k = 1$,

$$U^1 - (1-\beta_1)U^0 - \beta_1 U^\star = (1-\beta_1)\epsilon^0 + \beta_1(U^0 - U^\star) + (1-\beta_1)(T^\star U^0 - U^0)$$
$$\leq (1-\beta_1)\epsilon^0 + (1-\beta_1)\gamma \mathcal{P}^0(U^0 - U^\star) + (2\beta_1 - 1)(U^0 - U^\star)$$
$$\leq (1-\beta_1)\epsilon^0 + (1-\beta_1)\gamma \mathcal{P}^0(U^0 - U^\star),$$

where the second inequality is from the $(2\beta_1 - 1)(U^0 - U^\star) \leq 0$, and first inequality comes from Lemma 8 with $\alpha = 1, U = U^0, \bar{U} = U^0 - U^\star$, and let $\bar{U}$ be the entire right hand side of inequality. Then, we have

$$T^\star U^1 - U^1 = T^\star U^1 - (1-\beta_1)T^\star U^0 - \beta_1 U^\star - \beta_1(U^0 - U^\star) - (1-\beta_1)\epsilon^0$$
$$\leq \gamma \mathcal{P}^1((1-\beta_1)\epsilon^0 + (1-\beta_1)\gamma \mathcal{P}^0(U^0 - U^\star)) - \beta_1(U^0 - U^\star) - (1-\beta_1)\epsilon^0$$
$$= (1-\beta_1)\gamma \mathcal{P}^1 \gamma \mathcal{P}^0(U^0 - U^\star) - \beta_1(U^0 - U^\star) - (I - \gamma \mathcal{P}^1)(1-\beta_1)\epsilon^0.$$

where inequality comes from Lemma 8 with $\alpha = \beta_1, U = U^1, \tilde{U} = U^0$, and previously defined $\bar{U}$.

By induction,

$$U^k - (1-\beta_k)U^{k-1} - \beta_k U^\star$$
$$= \beta_k \left( U^0 - U^\star \right) + (1-\beta_k)(T^\star U^{k-1} - U^{k-1}) + (1-\beta_k)\epsilon^{k-1}$$
$$\leq \beta_k(U^0 - U^\star) - (1-\beta_k)\beta_{k-1}(U^0 - U^\star) + (1-\beta_k)\left( \Pi_{j=1}^{k-1}(1-\beta_j) \right) \left( \Pi_{l=k-1}^0 \gamma \mathcal{P}^l \right) (U^0 - U^\star)$$
$$+ (1-\beta_k)\epsilon^{k-1} - (1-\beta_k)\sum_{i=1}^{k-1} \Pi_{j=i}^{k-1}(1-\beta_j)\Pi_{l=k-1}^{i+1}\gamma \mathcal{P}^l \left( I - \gamma \mathcal{P}^i \right) \epsilon^{i-1}$$
$$\leq (1-\beta_k)\left( \Pi_{j=1}^{k-1}(1-\beta_j) \right) \left( \Pi_{l=k-1}^0 \gamma \mathcal{P}^l \right) (U^0 - U^\star) + (1-\beta_k)\epsilon^{k-1}$$
$$- (1-\beta_k)\sum_{i=1}^{k-1} \Pi_{j=i}^{k-1}(1-\beta_j)\Pi_{l=k-1}^{i+1}\gamma \mathcal{P}^l \left( I - \gamma \mathcal{P}^i \right) \epsilon^{i-1},$$

where the second inequality is from $\beta_k - (1 - \beta_k)\beta_{k-1} \leq 0$ and let $\bar{U}$ be the entire right hand side of inequality. Then, we have

$$
\begin{aligned}
T^\star U^k &- U^k \\
&= T^\star U^k - (1 - \beta_k)T^\star U^{k-1} - \beta_k T^\star U^\star - \beta_k(U^0 - U^\star) - (1 - \beta_k)\epsilon^{k-1} \\
&\leq \gamma\mathcal{P}^k\bigg((1 - \beta_k)\left(\Pi_{j=1}^{k-1}(1 - \beta_j)\right)\left(\Pi_{l=k-1}^0\gamma\mathcal{P}^l\right)(U^0 - U^\star) + (1 - \beta_k)\epsilon^{k-1} \\
&\quad - (1 - \beta_k)\sum_{i=1}^{k-1}\Pi_{j=i}^{k-1}(1 - \beta_j)\Pi_{l=k-1}^{i+1}\gamma\mathcal{P}^l\left(I - \gamma\mathcal{P}^i\right)\epsilon^{i-1}\bigg) - \beta_k(U^0 - U^\star) - (1 - \beta_k)\epsilon^{k-1} \\
&= \Pi_{j=1}^k(1 - \beta_j)\Pi_{l=k}^0\gamma\mathcal{P}^l(U^0 - U^\star) - \sum_{i=1}^k\Pi_{j=i}^k(1 - \beta_j)\Pi_{l=k}^{i+1}\gamma\mathcal{P}^l\left(I - \gamma\mathcal{P}^i\right)\epsilon^{i-1} \\
&\quad - \beta^k(U^0 - U^\star),
\end{aligned}
$$

where the first inequality comes from Lemma 8 with $\alpha = \beta_k, U = U^k, \tilde{U} = U^{k-1}$, and previously defined $\bar{U}$.

For the second inequality in Lemma 18, if $k = 1$,

$$
\begin{aligned}
U^1 - (1 - \beta_1)U^0 - \beta_1\hat{U}^\star &= (1 - \beta_1)\epsilon^0 + \beta_1(U^0 - \hat{U}^\star) + (1 - \beta_1)(T^\star U^0 - U^0) \\
&= (1 - \beta_1)\epsilon^0 + \beta_1(U^0 - \hat{U}^\star) + (1 - \beta_1)(T^\star U^0 - \hat{U}^\star - (U^0 - \hat{U}^\star)) \\
&\geq (1 - \beta_1)\epsilon^0 + \beta_1(U^0 - \hat{U}^\star) - (1 - \beta_1)(U^0 - \hat{U}^\star)
\end{aligned}
$$

where the second inequality is from $U^0 \geq T^\star U^0 \geq \hat{U}^\star$, and let $\bar{U}$ be the entire right hand side of inequality. Then, we have

$$
\begin{aligned}
T^\star U^1 - U^1 &= T^\star U^1 - (1 - \beta_1)T^\star U^0 - \beta_1 U^\star - \beta_1(U^0 - \hat{U}^\star) - (1 - \beta_1)\epsilon^0 \\
&\geq \gamma\mathcal{P}^1((1 - \beta_1)\epsilon^0 + \beta_1(U^0 - \hat{U}^\star) - (1 - \beta_1)(U^0 - \hat{U}^\star)) - \beta_1(U^0 - \hat{U}^\star) - (1 - \beta_1)\epsilon^0 \\
&= (2\beta_1 - 1)\gamma\mathcal{P}^1(U^0 - \hat{U}^\star) - \beta_1(U^0 - \hat{U}^\star) - (I - \gamma\mathcal{P}^1)(1 - \beta_1)\epsilon^0.
\end{aligned}
$$

where inequality comes from Lemma 9 with $\alpha = \beta_1, U = U^1, \tilde{U} = U^0$, and previously defined $\bar{U}$.

By induction,

$$
\begin{aligned}
U^k &- (1 - \beta_k)U^{k-1} - \beta_k\hat{U}^\star \\
&\geq (1 - \beta_k)\sum_{i=1}^{k-1}\left[(\beta_i - \beta_{i-1}(1 - \beta_i))\left(\Pi_{j=i+1}^{k-1}(1 - \beta_j)\right)\left(\Pi_{l=k-1}^i\gamma\hat{\mathcal{P}}^l\right)(U^0 - \hat{U}^\star)\right] \\
&\quad + (\beta_k - (1 - \beta_k)\beta_{k-1})(U^0 - \hat{U}^\star) + (1 - \beta_k)\epsilon^{k-1} \\
&\quad - (1 - \beta_k)\sum_{i=1}^{k-1}\Pi_{j=i}^{k-1}(1 - \beta_j)\Pi_{l=k-1}^{i+1}\gamma\hat{\mathcal{P}}^l\left(I - \gamma\hat{\mathcal{P}}^i\right)\epsilon^{i-1},
\end{aligned}
$$

and let $\bar{U}$ be the entire right hand side of inequality. Then, we have

$$T^\star U^k - U^k$$

$$= T^\star U^k - (1-\beta_k)T^\star U^{k-1} - \beta_k \hat{T}^\star \hat{U}^\star - \beta_k(U^0 - \hat{U}^\star) - (1-\beta_k)\epsilon^{k-1}$$

$$\geq \gamma\hat{\mathcal{P}}^k \left( (1-\beta_k) \sum_{i=1}^{k-1} \left[ (\beta_i - \beta_{i-1}(1-\beta_i)) \left( \Pi_{j=i+1}^{k-1}(1-\beta_j) \right) \left( \Pi_{l=k-1}^i \gamma\hat{\mathcal{P}}^l \right) (U^0 - \hat{U}^\star) \right] \right.$$

$$+ (\beta_k - (1-\beta_k)\beta_{k-1})(U^0 - \hat{U}^\star) - (1-\beta_k) \sum_{i=1}^{k-1} \Pi_{j=i}^{k-1}(1-\beta_j)\Pi_{l=k-1}^{i+1}\gamma\hat{\mathcal{P}}^l \left( I - \gamma\hat{\mathcal{P}}^i \right) \epsilon^{i-1}$$

$$\left. + (1-\beta_k)\epsilon^{k-1} \right) - (1-\beta_k)\epsilon^{k-1} - \beta_k(U^0 - \hat{U}^\star)$$

$$= \sum_{i=1}^{k} \left[ (\beta_i - \beta_{i-1}(1-\beta_i)) \left( \Pi_{j=i+1}^k(1-\beta_j) \right) \left( \Pi_{l=k}^i \gamma\hat{\mathcal{P}}^l \right) (U^0 - \hat{U}^\star) \right] - \beta_k(U^0 - \hat{U}^\star)$$

$$- \sum_{i=1}^{k} \Pi_{j=i}^k(1-\beta_j)\Pi_{l=k}^{i+1}\gamma\hat{\mathcal{P}}^l \left( I - \gamma\hat{\mathcal{P}}^i \right) \epsilon^{i-1},$$

where inequality comes from Lemma 9 with $\alpha = \beta_k, U = U^k, \tilde{U} = U^{k-1}$, and previously defined $\bar{U}$. $\qquad\square$

Now, we prove the second rate in Theorem 6.

*Proof of second rate in Theorem 6.* If we take $\|\cdot\|_\infty$ right side of first inequality in Lemma 18, we have

$$\frac{\left(\gamma^{-1} - \gamma\right)\gamma}{(\gamma^{k+1})^{-1} - \gamma^{k+1}} \left\| U^0 - U^\star \right\|_\infty + \frac{1+\gamma}{1+\gamma^{k+1}} \frac{1-\gamma^k}{1-\gamma} \max_{0 \leq i \leq k-1} \left\| \epsilon^i \right\|_\infty.$$

If we apply second inequality of Lemma 18 and take $\|\cdot\|_\infty$-norm right side, we have

$$\frac{\left(\gamma^{-1} - \gamma\right)\left(1 + \gamma - \gamma^{k+1}\right)}{(\gamma^{k+1})^{-1} - \gamma^{k+1}} \left\| U^0 - \hat{U}^\star \right\|_\infty + \frac{1+\gamma}{1+\gamma^{k+1}} \frac{1-\gamma^k}{1-\gamma} \max_{0 \leq i \leq k-1} \left\| \epsilon^i \right\|_\infty.$$

Therefore, we get

$$\left\| T^\star U^k - U^k \right\|_\infty \leq \frac{\left(\gamma^{-1} - \gamma\right)\left(1 + \gamma - \gamma^{k+1}\right)}{(\gamma^{k+1})^{-1} - \gamma^{k+1}} \left\| U^0 - \hat{U}^\star \right\|_\infty + \frac{1+\gamma}{1+\gamma^{k+1}} \frac{1-\gamma^k}{1-\gamma} \max_{0 \leq i \leq k-1} \left\| \epsilon^i \right\|_\infty,$$

since $\hat{U}^\star \leq U^\star \leq U^0$ implies that

$$\frac{\left(\gamma^{-1} - \gamma\right)\gamma}{(\gamma^{k+1})^{-1} - \gamma^{k+1}} \left\| U^0 - U^\star \right\|_\infty \leq \frac{\left(\gamma^{-1} - \gamma\right)\left(1 + \gamma - \gamma^{k+1}\right)}{(\gamma^{k+1})^{-1} - \gamma^{k+1}} \left\| U^0 - \hat{U}^\star \right\|_\infty.$$

$\qquad\square$

# F  Omitted proofs in Section 6

For the analyses, we first define $\hat{T}^\star_{GS} \colon \mathbb{R}^n \to \mathbb{R}^n$ as

$$\hat{T}^\star_{GS} = \hat{T}^\star_n \cdots \hat{T}^\star_2 \hat{T}^\star_1,$$

where $\hat{T}^\star_j : \mathbb{R}^n \to \mathbb{R}^n$ is defined as

$$\hat{T}^\star_j(U) = \left(U_1, \ldots, U_{j-1}, \left(\hat{T}^\star(U)\right)_j, U_{j+1}, \ldots, U_n\right)$$

for $j = 1, \ldots, n$, where $\hat{T}^\star$ is Bellman anti-optimality operator.

**Fact 3.** *[Classical result, [10, Proposition 1.3.2]] $\hat{T}^\star_{GS}$ is a $\gamma$-contractive operator and has the same fixed point as $\hat{T}^\star$.*

Now, we introduce the following lemmas.

**Lemma 19.** *Let $0 < \gamma < 1$. If $0 \le \alpha \le 1$, then there exist $\gamma$-contractive nonnegative matrix $\mathcal{P}_{GS}$ such that*

$$T^\star_{GS}U - (1-\alpha)T^\star_{GS}\tilde{U} - \alpha T^\star_{GS}U^\star \le \mathcal{P}_{GS}(U - (1-\alpha)\tilde{U} - \alpha U^\star).$$

**Lemma 20.** *Let $0 < \gamma < 1$. If $0 \le \alpha \le 1$, then there exist $\gamma$-contractive nonnegative matrix $\hat{\mathcal{P}}_{GS}$ such that*

$$\hat{\mathcal{P}}_{GS}(U - (1-\alpha)\tilde{U} - \alpha \hat{U}^\star) \le T^\star_{GS}U - (1-\alpha)T^\star_{GS}\tilde{U} - \alpha\hat{T}^\star_{GS}\hat{U}^\star.$$

*Proof of Lemma 19.* First let $U = V, \tilde{U} = \tilde{V}, U^\star = V^\star$. For $1 \le i \le n$, we have

$$T^\star_i V(s_i) - (1-\alpha)T^\star_i \tilde{V}(s_i) - \alpha T^\star_i V^\star(s_i) \le T^{\pi_i}_i V(s_i) - (1-\alpha)T^{\pi_i}_i \tilde{V}(s_i) - \alpha T^{\pi_i}_i V^\star(s_i)$$
$$= \gamma \mathcal{P}^{\pi_i}\left(V - (1-\alpha)\tilde{V} - \alpha V^\star\right)(s_i),$$

where $\pi_i$ is the greedy policy satisfying $T^{\pi_i}V = T^\star V$ and first inequality is from $T^{\pi_i}\tilde{V} \le T^\star\tilde{V}$ and $T^{\pi_i}V^\star \le T^\star V^\star$. Then, define matrix $\mathcal{P}_i$ as

$$\mathcal{P}_i(V) = (V_1, \ldots, V_{i-1}, (\gamma\mathcal{P}^{\pi_i}(V))_i, V_{i+1}, \ldots, V_n)$$

for $i = 1, \ldots, n$. Note that $\mathcal{P}_i$ is nonnegative matrix since $\mathcal{P}^{\pi_i}$ is nonnegative matrix. Then, we have

$$T^\star_i V - (1-\alpha)T^\star_i \tilde{V} - \alpha T^\star_i V^\star \le \mathcal{P}_i(V - (1-\alpha)\tilde{V} - \alpha V^\star).$$

By induction, there exist a sequence of matrices $\{\mathcal{P}_i\}_{i=1,\ldots,n}$ satisfying

$$T^\star_{GS}V - (1-\alpha)T^\star_{GS}\tilde{V} - \alpha T^\star_{GS}V^\star \le \mathcal{P}_n \cdots \mathcal{P}_1(V - (1-\alpha)\tilde{V} - \alpha V^\star)$$

since $T^\star_i V^\star = V^\star$ for all $i$. Denote $P_{GS}$ as $\mathcal{P}_n \cdots \mathcal{P}_1$. Then, $P_{GS}$ is $\gamma$-contractive nonnegative matrix since

$$\sum_{j=1}^n (P_{GS})_{ij} = \sum_{j=1}^n (\mathcal{P}_i \cdots \mathcal{P}_1)_{ij} \le \sum_{j=1}^n (\mathcal{P}_i)_{ij} = \gamma$$

for $1 \le i \le n$, where first equality is from definition of $\mathcal{P}_l$ for $i+1 \le l \le n$, inequality comes from definition of $\mathcal{P}_l$ for $1 \le l \le i-1$, and last equality is induced by definition of $\mathcal{P}_i$. Therefore, this implies that $\|P_{GS}\|_\infty \le \gamma$.

If $U = Q$, with similar argument of case $U = V$, let $\pi_i$ be the greedy policy, define matrix $\mathcal{P}_i$ as

$$\mathcal{P}_i(Q) = (Q_1, \ldots, Q_{i-1}, (\gamma\mathcal{P}^{\pi_i}(Q))_i, Q_{i+1}, \ldots, Q_n),$$

and denote $P_{GS}$ as $\mathcal{P}_n \cdots \mathcal{P}_1$. Then, $P_{GS}$ is $\gamma$-contractive nonnegative matrix satisfying

$$T^\star_{GS}Q - (1-\alpha)T^\star_{GS}\tilde{Q} - \alpha T^\star_{GS}Q^\star \le \mathcal{P}_{GS}(Q - (1-\alpha)\tilde{Q} - \alpha Q^\star).$$

$\square$

*Proof of Lemma 20.* First let $U = V, \tilde{U} = \tilde{V}, \hat{U}^\star = \hat{V}^\star$. For $1 \le i \le n$, we have

$$T^\star_i V(s_i) - (1-\alpha)T^\star_i \tilde{V}(s_i) - \alpha\hat{T}^\star_i \hat{V}^\star(s_i)$$
$$= \sup_{a\in\mathcal{A}}\left\{r(s_i, a) + \gamma\mathbb{E}_{s'\sim P(\cdot\,|\,s_i,a)}[V(s')]\right\} - \sup_{a\in\mathcal{A}}\left\{(1-\alpha)r(s_i, a) + (1-\alpha)\gamma\mathbb{E}_{s'\sim P(\cdot\,|\,s_i,a)}\left[\tilde{V}(s')\right]\right\}$$
$$- \inf_{a\in\mathcal{A}}\left\{\alpha r(s_i, a) + \alpha\gamma\mathbb{E}_{s'\sim P(\cdot\,|\,s_i,a)}\left[\hat{V}^\star(s')\right]\right\}$$
$$\ge \gamma\inf_{a\in\mathcal{A}}\left\{\mathbb{E}_{s'\sim P(\cdot\,|\,s_i,a)}\left[V(s') - (1-\alpha)\tilde{V}(s') - \alpha\hat{V}^\star(s')\right]\right\}.$$

Let $\hat\pi_i(\cdot\,|\,s) = \mathrm{argmin}_{a\in\mathcal{A}}\,\mathbb{E}_{s'\sim P(\cdot\,|\,s,a)}\left[V(s') - (1-\alpha)\tilde{V}(s') - \alpha\hat{V}^\star(s')\right]$ and define matrix $\hat{\mathcal{P}}_i$ as

$$\hat{\mathcal{P}}_i(V) = \left(V_1, \ldots, V_{i-1}, \left(\gamma\mathcal{P}^{\hat\pi_i}(V)\right)_i, V_{i+1}, \ldots, V_n\right)$$

for $i = 1, \ldots, n$. Note that $\hat{\mathcal{P}}_i$ is nonnegative matrix since $\mathcal{P}^{\hat\pi_i}$ is nonnegative matrix. Then, we have

$$\hat{\mathcal{P}}_i(V - (1-\alpha)\tilde{V} - \alpha\hat{V}^\star) \le T_i^\star V - (1-\alpha)T_i^\star\tilde{V} - \alpha T_i^\star\hat{V}^\star.$$

By induction, there exist a sequence of matrices $\{\hat{\mathcal{P}}_i\}_{i=1,\ldots,n}$ satisfying

$$\hat{\mathcal{P}}_n\cdots\hat{\mathcal{P}}_1(V - (1-\alpha)\tilde{V} - \alpha\hat{V}^\star) \le T_{GS}^\star V - (1-\alpha)T_{GS}^\star\tilde{V} - \alpha\hat{T}_{GS}^\star\hat{V}^\star,$$

and denote $\hat{P}_{GS}$ as $\hat{\mathcal{P}}_n\cdots\hat{\mathcal{P}}_1$. With same argument in proof of Lemma 19, $\hat{P}_{GS}$ is $\gamma$-contractive nonnegative matrix.

If $U = Q$, with similar argument, let $\hat\pi_i(\cdot\,|\,s) = \mathrm{argmin}_{a\in\mathcal{A}}\{Q(s,a) - (1-\alpha)\tilde{Q}(s,a) - \alpha\hat{Q}^\star(s,a)\}$ and define matrix $\hat{\mathcal{P}}_i$ as

$$\mathcal{P}_i(Q) = \left(U_1, \ldots, Q_{i-1}, \left(\gamma\mathcal{P}^{\hat\pi_i}(Q)\right)_i, Q_{i+1}, \ldots, Q_n\right).$$

Denote $\hat{P}_{GS}$ as $\hat{\mathcal{P}}_n\cdots\hat{\mathcal{P}}_1$. Then, with same argument in proof of Lemma 19, $\hat{P}_{GS}$ is $\gamma$-contractive nonnegative matrix satisfying

$$\hat{\mathcal{P}}_{GS}(Q - (1-\alpha)\tilde{Q} - \alpha\hat{Q}^\star) \le T_{GS}^\star Q - (1-\alpha)T_{GS}^\star\tilde{Q} - \alpha\hat{T}_{GS}^\star\hat{Q}^\star.$$

$\square$

Next, we prove following key lemma.

**Lemma 21.** *Let $0 < \gamma < 1$. For the iterates $\{U^k\}_{k=0,1,\ldots}$ of* (GS-Anc-VI), *there exist $\gamma$-contractive nonnegative matrices $\{\mathcal{P}_{GS}^l\}_{l=0,1,\ldots,k}$ and $\{\hat{\mathcal{P}}_{GS}^l\}_{l=0,1,\ldots,k}$ such that*

$$
\begin{aligned}
T_{GS}^\star U^k - U^k \le &\sum_{i=1}^{k}\left[(\beta_i - \beta_{i-1}(1-\beta_i))\left(\Pi_{j=i+1}^k(1-\beta_j)\right)\left(\Pi_{l=k}^i\mathcal{P}_{GS}^l\right)(U^0 - U^\star)\right]\\
&- \beta_k(U^0 - U^\star) + \left(\Pi_{j=1}^k(1-\beta_j)\right)\left(\Pi_{l=k}^0\mathcal{P}_{GS}^l\right)(U^0 - U^\star),
\end{aligned}
$$

$$
\begin{aligned}
T_{GS}^\star U^k - U^k \ge &\sum_{i=1}^{k}\left[(\beta_i - \beta_{i-1}(1-\beta_i))\left(\Pi_{j=i+1}^k(1-\beta_j)\right)\left(\Pi_{l=k}^i\hat{\mathcal{P}}_{GS}^l\right)(U^0 - \hat{U}^\star)\right]\\
&- \beta_k(U^0 - \hat{U}^\star) + \left(\Pi_{j=1}^k(1-\beta_j)\right)\left(\Pi_{l=k}^0\hat{\mathcal{P}}_{GS}^l\right)(U^0 - \hat{U}^\star),
\end{aligned}
$$

*where $\Pi_{j=k+1}^k(1-\beta_j) = 1$ and $\beta_0 = 1$.*

*Proof of Lemma 21.* First, we prove first inequality in Lemma 21 by induction.

If $k = 0$,

$$
\begin{aligned}
T_{GS}^\star U^0 - U^0 &= T_{GS}^\star U^0 - U^\star - (U^0 - U^\star)\\
&= T_{GS}^\star U^0 - T_{GS}^\star U^\star - (U^0 - U^\star)\\
&\le \mathcal{P}_{GS}^0(U^0 - U^\star) - (U^0 - U^\star).
\end{aligned}
$$

where inequality comes from Lemma 19 with $\alpha = 1, U = U^0$.

By induction,

$$
\begin{aligned}
&T^{\star}_{GS}U^k - U^k \\
&= T^{\star}_{GS}U^k - (1-\beta_k)T^{\star}_{GS}U^{k-1} - \beta_k T^{\star}_{GS}U^{\star} - \beta_k(U^0 - U^{\star}) \\
&\leq \mathcal{P}^k_{GS}(U^k - (1-\beta_k)U^{k-1} - \beta_k U^{\star}) - \beta_k(U^0 - U^{\star}) \\
&= \mathcal{P}^k_{GS}(\beta_k(U^0 - U^{\star}) + (1-\beta_k)(T^{\star}_{GS}U^{k-1} - U^{k-1})) - \beta_k(U^0 - U^{\star}) \\
&\leq (1-\beta_k)\mathcal{P}^k_{GS}\Bigg( \sum_{i=1}^{k-1} \left[ (\beta_i - \beta_{i-1}(1-\beta_i)) \left(\Pi^{k-1}_{j=i+1}(1-\beta_j)\right) \left(\Pi^i_{l=k-1}\mathcal{P}^l_{GS}\right)(U^0 - U^{\star}) \right] \\
&\qquad - \beta_{k-1}(U^0 - U^{\star}) + \left(\Pi^{k-1}_{j=1}(1-\beta_j)\right)\left(\Pi^0_{l=k-1}\mathcal{P}^l_{GS}\right)(U^0 - U^{\star}) \Bigg) \\
&\qquad + \beta_k \mathcal{P}^k_{GS}(U^0 - U^{\star}) - \beta_k(U^0 - U^{\star}) \\
&= \sum_{i=1}^{k} \left[ (\beta_i - \beta_{i-1}(1-\beta_i)) \left(\Pi^k_{j=i+1}(1-\beta_j)\right)\left(\Pi^i_{l=k}\mathcal{P}^l_{GS}\right)(U^0 - U^{\star}) \right] \\
&\qquad - \beta_k(U^0 - U^{\star}) + \left(\Pi^k_{j=1}(1-\beta_j)\right)\left(\Pi^0_{l=k}\mathcal{P}^l_{GS}\right)(U^0 - U^{\star})
\end{aligned}
$$

where the first inequality comes from Lemma 19 with $\alpha = \beta_k, U = U^k, \tilde{U} = U^{k-1}$, and second inequality comes from nonnegativeness of $\mathcal{P}^k_{GS}$.

First, we prove second inequality in Lemma 21 by induction.

If $k = 0$,

$$
\begin{aligned}
T^{\star}_{GS}U^0 - U^0 &= T^{\star}_{GS}U^0 - \hat{U}^{\star} - (U^0 - \hat{U}^{\star}) \\
&= T^{\star}_{GS}U^0 - \hat{T}^{\star}_{GS}\hat{U}^{\star} - (U^0 - \hat{U}^{\star}) \\
&\geq \hat{\mathcal{P}}^0_{GS}(U^0 - \hat{U}^{\star}) - (U^0 - \hat{U}^{\star}),
\end{aligned}
$$

where inequality comes from Lemma 20 with $\alpha = 1, U = U^0$.

By induction,

$$
\begin{aligned}
&T^{\star}_{GS}U^k - U^k \\
&= T^{\star}_{GS}U^k - (1-\beta_k)T^{\star}_{GS}U^{k-1} - \beta_k \hat{T}^{\star}_{GS}\hat{U}^{\star} - \beta_k(U^0 - \hat{U}^{\star}) \\
&\geq \hat{\mathcal{P}}^k_{GS}(U^k - (1-\beta_k)U^{k-1} - \beta_k \hat{U}^{\star}) - \beta_k(U^0 - \hat{U}^{\star}) \\
&= \hat{\mathcal{P}}^k_{GS}(\beta_k(U^0 - \hat{U}^{\star}) + (1-\beta_k)(T^{\star}_{GS}U^{k-1} - U^{k-1})) - \beta_k(U^0 - \hat{U}^{\star}) \\
&\geq (1-\beta_k)\hat{\mathcal{P}}^k_{GS}\Bigg( \sum_{i=1}^{k-1} \left[ (\beta_i - \beta_{i-1}(1-\beta_i)) \left(\Pi^{k-1}_{j=i+1}(1-\beta_j)\right) \left(\Pi^i_{l=k-1}\hat{\mathcal{P}}^l_{GS}\right)(U^0 - \hat{U}^{\star}) \right] \\
&\qquad - \beta_{k-1}(U^0 - \hat{U}^{\star}) + \left(\Pi^{k-1}_{j=1}(1-\beta_j)\right)\left(\Pi^0_{l=k-1}\hat{\mathcal{P}}^l_{GS}\right)(U^0 - \hat{U}^{\star}) \Bigg) \\
&\qquad + \beta_k \hat{\mathcal{P}}^k_{GS}(U^0 - \hat{U}^{\star}) - \beta_k(U^0 - \hat{U}^{\star}) \\
&= \sum_{i=1}^{k} \left[ (\beta_i - \beta_{i-1}(1-\beta_i)) \left(\Pi^k_{j=i+1}(1-\beta_j)\right) \left(\Pi^i_{l=k}\hat{\mathcal{P}}^l_{GS}\right)(U^0 - \hat{U}^{\star}) \right] \\
&\qquad - \beta_k(U^0 - \hat{U}^{\star}) + \left(\Pi^k_{j=1}(1-\beta_j)\right)\left(\Pi^0_{l=k}\hat{\mathcal{P}}^l_{GS}\right)(U^0 - \hat{U}^{\star})
\end{aligned}
$$

where the first inequality comes from Lemma 20 with $\alpha = \beta_k, U = U^k, \tilde{U} = U^{k-1}$, and nonnegativeness of $\hat{\mathcal{P}}^k_{GS}$. $\qquad\square$

Now, we prove the first rate in Theorem 7.

*Proof of first rate in Theorem 7.* Since $B_1 \leq A \leq B_2$ implies $\|A\|_\infty \leq \sup\{\|B_1\|_\infty, \|B_2\|_\infty\}$ for $A, B \in \mathcal{F}(\mathcal{X})$, if we take $\|\cdot\|_\infty$ right side of first inequality in Lemma 21, we have

$$\sum_{i=1}^{k} |\beta_i - \beta_{i-1}(1-\beta_i)| \left(\Pi_{j=i+1}^k (1-\beta_j)\right) \left\|\left(\Pi_{l=k}^i \mathcal{P}_{GS}^l\right)(U^0 - U^\star)\right\|_\infty$$

$$+ \beta_k \left\|U^0 - U^\pi\right\|_\infty + \left(\Pi_{j=1}^k(1-\beta_j)\right) \left\|\left(\Pi_{l=k}^0 \mathcal{P}_{GS}^l\right)(U^0 - U^\star)\right\|_\infty$$

$$\leq \left(\sum_{i=1}^{k} \gamma^{k-i+1} |\beta_i - \beta_{i-1}(1-\beta_i)| \left(\Pi_{j=i+1}^k(1-\beta_j)\right) + \beta_k + \gamma^{k+1}\Pi_{j=1}^k(1-\beta_j)\right)$$

$$\left\|U^0 - U^\star\right\|_\infty$$

$$= \frac{(\gamma^{-1} - \gamma)(1 + 2\gamma - \gamma^{k+1})}{(\gamma^{k+1})^{-1} - \gamma^{k+1}} \left\|U^0 - U^\star\right\|_\infty,$$

where the first inequality comes from triangular inequality, second inequality is from $\gamma$-contraction of $\mathcal{P}_{GS}^l$, and last equality comes from calculations. If we take $\|\cdot\|_\infty$ right side of second inequality in Lemma 21, we have

$$\sum_{i=1}^{k} |\beta_i - \beta_{i-1}(1-\beta_i)| \left(\Pi_{j=i+1}^k (1-\beta_j)\right) \left\|\left(\Pi_{l=k}^i \hat{\mathcal{P}}_{GS}^l\right)(U^0 - \hat{U}^\star)\right\|_\infty$$

$$+ \beta_k \left\|U^0 - U^\pi\right\|_\infty + \left(\Pi_{j=1}^k(1-\beta_j)\right) \left\|\left(\Pi_{l=k}^0 \hat{\mathcal{P}}_{GS}^l\right)(U^0 - \hat{U}^\star)\right\|_\infty$$

$$\leq \left(\sum_{i=1}^{k} \gamma^{k-i+1} |\beta_i - \beta_{i-1}(1-\beta_i)| \left(\Pi_{j=i+1}^k(1-\beta_j)\right) + \beta_k + \gamma^{k+1}\Pi_{j=1}^k(1-\beta_j)\right)$$

$$= \frac{(\gamma^{-1} - \gamma)(1 + 2\gamma - \gamma^{k+1})}{(\gamma^{k+1})^{-1} - \gamma^{k+1}} \left\|U^0 - \hat{U}^\star\right\|_\infty,$$

where the first inequality comes from triangular inequality, second inequality is from from $\gamma$-contraction of $\hat{\mathcal{P}}_{GS}^l$, and last equality comes from calculations. Therefore, we conclude

$$\left\|T_{GS}^\star U^k - U^k\right\|_\infty \leq \frac{(\gamma^{-1} - \gamma)(1 + 2\gamma - \gamma^{k+1})}{(\gamma^{k+1})^{-1} - \gamma^{k+1}} \max\left\{\left\|U^0 - U^\star\right\|_\infty, \left\|U^0 - \hat{U}^\star\right\|_\infty\right\}.$$

$\square$

For the second rates of Theorem 7, we introduce following lemma.

**Lemma 22.** *Let $0 < \gamma < 1$. For the iterates $\{U^k\}_{k=0,1,\ldots}$ of (GS-Anc-VI), if $U^0 \leq T_{GS}^\star U^0$, then $U^{k-1} \leq U^k \leq T_{GS}^\star U^{k-1} \leq T_{GS}^\star U^k \leq U^\star$ for $1 \leq k$. Also, if $U^0 \geq T_{GS}^\star U^0$, then $U^{k-1} \geq U^k \geq T_{GS}^\star U^{k-1} \geq T_{GS}^\star U^k \geq U^\star$ for $1 \leq k$.*

*Proof.* By Fact 3, $\lim_{m \to \infty} T_{GS}^\star U = U^\star$. By definition, if $U \leq \tilde{U}$, $T_i^\star U \leq T_i^\star \tilde{U}$ for any $1 \leq i \leq n$ and this implies that if $U \leq \tilde{U}$, then $T_{GS}^\star U \leq T_{GS}^\star \tilde{U}$. Hence, with same argument in proof of Lemma 5, we can obtain desired results. $\square$

Now, we prove the second rates in Theorem 7.

*Proof of second rates in Theorem 7.* If $U^0 \leq T_{GS}^\star U^0$, then $U^0 - U^\star \leq 0$ and $U^k \leq T_{GS}^\star U^k$ by Lemma 22. Hence, by Lemma 21, we get

$$0 \leq T_{GS}^\star U^k - U^k$$

$$= \sum_{i=1}^{k} \left[(\beta_i - \beta_{i-1}(1-\beta_i))\left(\Pi_{j=i+1}^k(1-\beta_j)\right)\left(\Pi_{l=k}^i \mathcal{P}_{GS}^l\right)(U^0 - U^\star)\right]$$

$$- \beta_k(U^0 - U^\pi) + \left(\Pi_{j=1}^k(1-\beta_j)\right)\left(\Pi_{l=k}^0 \mathcal{P}_{GS}^l\right)(U^0 - U^\star)$$

$$\leq \sum_{i=1}^{k} \left[(\beta_i - \beta_{i-1}(1-\beta_i))\left(\Pi_{j=i+1}^k(1-\beta_j)\right)\left(\Pi_{l=k}^i \mathcal{P}_{GS}^l\right)(U^0 - U^\star)\right] - \beta_k(U^0 - U^\star),$$

where the second inequality follows from $\left(\Pi_{j=1}^{k}(1-\beta_j)\right)\left(\Pi_{l=k}^{i}\mathcal{P}_{GS}^{l}\right)(U^0 - U^\star) \leq 0$. Taking $\|\cdot\|_\infty$-norm both sides, we have

$$\left\|T_{GS}^{\star}U^k - U^k\right\|_\infty \leq \frac{\left(\gamma^{-1} - \gamma\right)\left(1 + \gamma - \gamma^{k+1}\right)}{\left(\gamma^{k+1}\right)^{-1} - \gamma^{k+1}}\left\|U^0 - U^\star\right\|_\infty.$$

Otherwise, if $U^0 \geq T_{GS}^{\star}U^0$, $U^k \geq T_{GS}^{\star}U^k$ and $U^0 \geq U^\star \geq \hat{U}^\star$ by Lemma 22 and 3. Thus, by Lemma 21, we get

$$0 \geq T_{GS}^{\star}U^k - U^k$$
$$\geq \sum_{i=1}^{k}\left[\left(\beta_i - \beta_{i-1}(1-\beta_i)\right)\left(\Pi_{j=i+1}^{k}(1-\beta_j)\right)\left(\Pi_{l=k}^{i}\hat{\mathcal{P}}_{GS}^{l}\right)(U^0 - \hat{U}^\star)\right] - \beta_k(U^0 - \hat{U}^\star),$$

where the second inequality follows from $0 \leq \left(\Pi_{j=1}^{k}(1-\beta_j)\right)\left(\Pi_{l=k}^{0}\hat{\mathcal{P}}_{GS}^{l}\right)(U^0 - \hat{U}^\star)$. Taking $\|\cdot\|_\infty$-norm both sides, we have

$$\left\|T_{GS}^{\star}U^k - U^k\right\|_\infty \leq \frac{\left(\gamma^{-1} - \gamma\right)\left(1 + \gamma - \gamma^{k+1}\right)}{\left(\gamma^{k+1}\right)^{-1} - \gamma^{k+1}}\left\|U^0 - \hat{U}^\star\right\|_\infty.$$

$\square$

# G   Broader Impacts

Our work focuses on the theoretical aspects of reinforcement learning. There are no negative social impacts that we anticipate from our theoretical results.

# H   Limitations

Our analysis concerns value iteration. While value iteration is of theoretical interest, the analysis of value iteration is not sufficient to understand modern deep reinforcement learning practices.

