# OpenReview forum: "Accelerating Value Iteration with Anchoring"
_NeurIPS.cc/2023/Conference — NeurIPS 2023 poster_

### Official Review · Reviewer_e9mJ · 2023-06-08

**Soundness:** 3 good
**Presentation:** 2 fair
**Contribution:** 3 good
**Rating:** 7
**Confidence:** 4

**Summary:**

New variants of value iterations based on anchoring
Complexity lower-bound that shows that their algorithms is optimal in a worst-case sense
Nice extensions for various important cases ($\gamma =1$, approximate VI, gauss-seidel, infinite state-action spaces).

**Strengths:**

* Strong theorems, able to overcome the shortcomings of the analysis of some prior work that only prove acceleration for linear operators [1, 37]

* Well-written and easy to follow

**Weaknesses:**

* A bit sketchy on the terminology: “rate” is not defined, some inaccurate statements (“O(1) rate for VI when $\gamma \approx 1$”), see my questions below
* No numerical comparisons with existing methods
* No analysis in the model-free setup (using sampling)

**Questions:**

p1, l20: “the optimal rate for the VI setup was not known” : isn’t it Theorem 3 in [37] ? This provides a lower-bound on the rate for both value iteration and value evaluation, and shows that VI achieves this rate.

p1, l26 - contributions: I am a bit puzzled by the formulations of “$O(1/k)$ rate for $\gamma \approx 1$” etc. While this becomes clear when the exact theorems are stated, it may be worth expanding here on the precise formulations of your results.

p2, l45, definition of Bellman operator and Bellman consistency: please mention that you overload the notation. Right now $T^\pi$ is both an operator of $R^{|S|}$ and $R^{|S|\times|A|}$.

Section 1.1: please define what is the rate of an algorithm. There may be inconsistencies since usually rate suggests that the convergence is linear and the rate $\rho$ means that error decreases as $\rho^k$ after k iterations; whereas I believe that the rate of p3, l83 $O(1/k)$ is for the error to decrease as $1/k$ after iterations. Please clarify.

Distinctions between anchoring and Nesterov’s acceleration: some recent works show that in some setup, anchoring and Nesterov’s acceleration are exactly equivalent - they are the same algorithm up to some reformulation [0]. Does this connection hold in your setup?

[0] Tran-Dinh, Q. (2022). The Connection Between Nesterov's Accelerated Methods and Halpern Fixed-Point Iterations. arXiv preprint arXiv:2203.04869.

p4, l122: why not simplify the expression of $\beta_k$ ? The denominator is just a partial geometric sum.

p4, lines 151-152: this is quite sketchy, since “rate” has not been defined (see my previous comment). The comments about “$O(1)$ rate for VI when $\gamma \approx 1$” do not make sense to me. I understand $\gamma \approx 1$ means that VI will converge slowly, but it is more convincing to compare the number of iterations before returning an $\epsilon$ optimal solution.

Choice of U0: to satisfy the inequalities of line 152, one can choose either $U_0 = 0$ or $U_0 = R / (1-\gamma)$, with $R$ an upper bound on the rewards. Would that be a good choice in practice though?

Theorem 1: is your choice of \beta_j the optimal choice?

Theorem 5: is your MDP instance different from the MDP instance from Theorem 3 in [37]? Please clarify.


**Limitations:**


* No significant limitations apart from the absence of numerical simulations. It is not clear if the algorithms proposed in this paper perform well; beating vanilla value iteration is not really hard in practice.

---

> ### Author Rebuttal · Authors · 2023-08-10
>
> We thank the reviewer for the insightful feedback.
>
> Weakness
>
> (i) (+Questions (iv) (vii))
> Thank you for this point. By "rate", we meant the $\mathcal{O}$-dependency of the error. We will define our use of the term "rate" and we will revise the statement "$O(1)$ rate for VI when $\gamma\approx 1$" to make it more precise. (By $\gamma\approx 1$, we meant $\gamma> 1-1/k$.)
>
> (ii) (+limitation)  We performed several numerical experiments for Anc-VI and found that Anc-VI provided a practical acceleration only in some cases. (Our rate is a worst-case rate, so when the MDP is not a worst-case instance, there is no guarantee that the actual rate of Anc-VI is better than regular VI.) However, we were not yet able to find an adequate theoretical or heuristic explanation that tells us when we can expect the anchor to provide a practical acceleration, so we chose not to present them and left this issue to future work.
>
> (iii) In fixed-point theory and minimax optimization literature, to the best of our knowledge, whether anchoring provides an acceleration in stochastic settings is still an open problem. The analysis Anc-VI in a model-free setting is indeed an important direction that we plan to study in future work.
>
> Questions
>
> (i)  Sorry for the confusion. We try to indicate the optimal rate in terms of Bellman error, not the distance to the optimal value function. We will clarify this in revision.
>
> (ii), (iii) We will clarify overload of notation and reflect your comment in revision.
>
> (v) Thank you for pointing out this reference.
> This work does reveal a connection between anchor acceleration and Nesterov's acceleration, but we claim that the two mechanisms are different in the following sense. First, following reformulation scheme of Theorem $3.2$ in this reference, A-VI in [1], a Nesterov-type VI, can not be reformulated to Anchoring type VI since reformulation only covers restricted Nesterov type algorithm. Second, there is line of research for continuous-time models of acceleration, and [2] and [3] showed that anchor acceleration and Nesterov's acceleration have distinct ODE models. It is our view that Nesterov's acceleration and anchor acceleration have a connection but are substantively distinct, but studying the connections and differences between these two acceleration types is certainly an interesting direction.
>
> (vi) Now that we think about it, the reviewer is probably correct that the expression is simpler when we carry out the partial geometric sum. Thank you for pointing this out. We will make that change in the revision.
>
> (viii) Although we believe that $U_0=0$ is a natural choice, it is probably not a "good" choice for all MDPs. In our view, the requirement $U^0\le TU^0$ is not an onerous one, but it can be a cumbersome one. We feel that this requirement is probably an artifact of the analysis, and we will try to relax it in our future work.
>
> (ix) We believe it is likely that there is a better choice of $\beta_j$ that slightly improves the constant. However, our choice $\beta_j$ is a relatively simple one that attains a rate that is optimal up to a constant factor of $4$, so it is the optimal choice in that sense. In terms of our analysis, our choice of $\beta_j$ is the coefficients that optimize our given analysis.
>
> (x) The worst-case MDP in [4] and worst-case MDP in our paper are different primarily in their rewards: $r(s_i, a_1)=\mathbf{1}_{\{i=1\}}$
>
> and $r(s_i, a_1)=\mathbf{1}_{\{i=2\}}$, respectively. Less significantly, our worst-case MDP has one additional state, but the transition probabilities are essentially the same. Both MDPs have only one action. We will clarify this point in our revision.
>
> [1] V. Goyal and J. Grand-Clément. A first-order approach to accelerated value iteration. Operations Research, 71(2):517–535, 2022.
>
> [2] Su, W., Boyd, S., and Cand\`es, E. J.  A
> differential equation for modeling Nesterov’s accelerated gradient method: Theory and insights. Journal of Machine Learning Research, 17(153):1–43, 2016.
>
> [3] J. J. Suh, J. Park, and E. K. Ryu. Continuous-time Analysis of Anchor Acceleration arXiv:2304.00771, 2023.

---

> > ### Comment · Reviewer_e9mJ · 2023-08-11
> >
> > I would like to thank the authors for their very detailed responses. I think that they did a great at addressing my remarks and questions.

---

### Official Review · Reviewer_tQc7 · 2023-06-22

**Soundness:** 3 good
**Presentation:** 3 good
**Contribution:** 3 good
**Rating:** 7
**Confidence:** 3

**Summary:**

The authors focus their study on the theoretical analysis of a (simple) variation of the Value Iteration (VI) algorithm, a classical and grounding tool behind many modern (deep)-RL algorithms.
The variation considered by the authors incorporates anchor acceleration mechanisms leading to what the authors call Anchored Value Iteration (Anc-VI). More specifically, rather than repeatedly applying the bellman consistency/optimality operators $T$ to reach the fixed point (i.e., $x_{k+1} = T x_k$), Anc-VI obtains the next point as a convex combination between the initial point and the result of applying the operator on the previous point, namely $x_{k+1} = \beta_k x_0 + (1-\beta_k) T x_{k}$. Naturally, the sequence $ \beta_k$ is a vanishing sequence.

After introducing the algorithm, the authors proceed with an in-depth analysis of the convergence rates of Anc-VI.

More specifically:
1) They study the convergence rate of the Bellman error (i.e, $||Tx_k - x_k||_{\infty}$) both for the consistency and the optimal operators. The authos show that Anc-VI converges at rate $\mathcal{O}\left( 1/k \right)$ for $\gamma \approx 1$ (while standard VI has rate $\mathcal{O}(1)$).
2) Under mild assumptions, the authors derive a lower bound that shows that the convergence rates of Anc-VI are tight up to a constant factor of 4.
3) The author extends their study to the case in which $\gamma=1$, where VI may not converge to a fixed point even if one exists. Anc-VI, on the other hand, converges to some fixed point asymptotically, and the bellman error shrinks to 0 at a linear rate.
4) Finally, results are extended to the approximated value iteration setting and to the Gauss-Seidel variation of VI.




**Strengths:**

VI is a grounding tool of many modern RL algorithms, and its analysis has gathered the community's attention for a long time. The authors thoroughly review existing works and properly contextualize their results within the field.
Applying the idea of anchoring to Value Iteration is, to the best of my knowledge, novel and leads to some surprising results, which have all been stated in summary above.

More specifically, the most relevant ones are:
1) Optimality of the accelerated rate of Anc-VI (i.e., upper bound matches lower bound up to constant factor).
2) Convergence in the undiscounted setting.

Given the fact that VI is the grounding core of many modern methods, I think these results are of interest to the RL community at NeurIPS. For this reason, I recommend acceptance.

On the clarity. Overall, the paper is a nice read, and not particularly hard to follow.

**Weaknesses:**

Not much.

**Questions:**

1) Do you have any intuitive explanation of why such stronger results can be obtained with anchoring? Why should be want to anchor our search close to the initial point? I think this point might add value for possible future development of RL algorithms that might take inspiration from this work.
2) Why should we anchor only on the initial value rather than doing a, e.g., linear combination of the past value functions? (except for memory requirements).
3) Is there any existing variation of VI whose upper-bound matches the lower-bound?


**Limitations:**

The authors discuss Limitations in Appendix H. Furthermore, the value of each theoretical result is properly discussed after presenting each result.

Potential negative societal impact. The paper deals with foundational research on the convergence rates of Value Iteration. I don't see a direct path to negative applications.

---

> ### Author Rebuttal · Authors · 2023-08-10
>
> We are happy to hear that the reviewer found our work interesting.
>
> Questions
>
> (i) Referring to prior works [1,2,3], we conjectured  that the anchoring mechanism, which pulls the present iterates toward the anchoring point, provides stability and prevents a certain type of cycling behavior. However, we acknowledge that our intuitive understanding of the anchor mechanism is not very strong, and the question of 'why' (beyond the convergence proof) is an interesting direction of future work.
>
> (ii) In fixed-point theory literature, there is a line of research that analyzes the convergence of the anchoring mechanism with arbitrary anchoring points for nonexpansive operators [Section 30.1, 4]. It seems reasonable to consider an anchor that moves, perhaps as a linear combination of past points, and exploring the effectiveness of such variants is an interesting direction.
>
> (iii) As far as we know, there is no prior variant of VI that matched a lower bound. It is worth mentioning that [5] proposed A-VI, a variant of VI that uses Nesterov-type momentum, and showed that it exhibits an accelerated rate in terms of distance to the value function for the Bellman consistency operator. However, this result requires a "reversibility" assumption on the MDP.
>
> [1] T. Yoon and E. K. Ryu. Accelerated algorithms for smooth convex-concave minimax problems with
> $O(1/k^2)$ rate on squared gradient norm. International Conference on Machine Learning, 2021.
>
> [2] J. Park and E. K. Ryu. Exact optimal accelerated complexity for fixed-
> point iterations. International Conference on Machine Learning, 2022.
>
> [3] J. J. Suh, J. Park, and E. K. Ryu. Continuous-time Analysis of Anchor Acceleration arXiv:2304.00771, 2023.
>
> [4] Bauschke, H. H. and Combettes, P. L. Convex Analysis and Monotone Operator Theory in Hilbert Spaces. Springer, second edition, 2017.
>
> [5] V. Goyal and J. Grand-Clément. A first-order approach to accelerated value iteration. Operations Research, 71(2):517–535, 2022.

---

> > ### Comment · Reviewer_tQc7 · 2023-08-10
> > **Ack**
> >
> > I thank the authors for their rebuttal, and for answering my questions. After reading all reviews, I confirm my score.

---

### Official Review · Reviewer_GSn8 · 2023-06-22

**Soundness:** 3 good
**Presentation:** 3 good
**Contribution:** 3 good
**Rating:** 6
**Confidence:** 4

**Summary:**

This paper considers an anchored version of Value Iteration and derives accelerated rates in terms of the Bellman error for both the Bellman consistency and optimality operators. Then, the work addresses the particular case of $\gamma =1$ with a $O(1/k)$ rate that VI fails to guarantee via the standard contraction argument. A complexity lower bound matching the upper-bound (up to a constant numerical factor) is then established. The paper further proposes an error propagation analysis of approximate anchored VI, extending the results of the exact case and a Gauss-Seidel version of the algorithm is also analyzed.


**Strengths:**

- I find the fact that the Anc-VI algorithm is able to address the case $\gamma  =1$ interesting and the anchoring mechanism seems to be well suited for this case unlike VI.
It is interesting to see that the anchoring idea which was recently investigated in depth in optimization is also interesting in the DP setting. While the idea may seem natural as DP involves fixed point iterations, the execution requires to address several technical challenges that are more specific to the DP setting.

- The contributions of the paper are solid and somehow comprehensive with accelerated convergence rates, a lower bound and extensions to inexact and Gauss-Seidel variants.

- The anchoring mechanism can also be used for further extensions in DP/RL for the design of learning algorithms beyond the deterministic setting.

- The paper is well-written and the contributions are overall clearly stated.



**Weaknesses:**

1. Even if the Bellman error is a natural quantity as a performance measure for fixed point iteration schemes, it would be nice if the paper can comment more if possible on the motivation to consider convergence guarantees in terms of this performance measure compared to the distance to the optimal value function. It is mentioned in l. 18 that the distance to the optimum is not computable and that the Bellman error can be monitored which I find a valid and fair interesting point. However, as a matter of fact, the lower bound provided in [37, Theorem 3] (which is mentioned in the paper) in terms of distance to the optimal value function is actually achieved for Value Iteration. If we would like to approximate the optimal value function as fast as possible to find an optimal policy in a control task for instance, why would we use anchored VI instead of VI? See also the related question 5 in the ‘Questions’ section below.

2.  It is not discussed how to find a near-optimal policy using Anchored VI.  I guess you can just output a policy that is greedy with respect to the output of the Anchored VI algorithm after a certain number of iterations as it is usually done with VI for which there are policy error bounds. Given that Section 1.1 mentions optimal policies, it would be nice to mention that the Anchored VI could be further enhanced to find near-optimal policies as an approximate planning algorithm.

3. The paper only discusses guarantees in terms of the Bellman error comparing with the translated result for VI from distance to optimal value function $\|\|U^k - U^{\star}\|\|_{\infty}$ to Bellman error. It is also similarly possible to translate the Bellman error guarantee to a distance to optimality result. See question 5 below.

4. Case $\gamma = 1$: How can the (action) state value functions be defined in that case? Definitions of Section 1.1 may not be relevant anymore. While the paper states that that ‘a full treatment of undiscounted MDPs is beyond the scope of this paper’, I think at least adding references would be useful to give a meaning to the problem and hint to the fact that fixed points can be guaranteed to exist (as currently assumed in the paper) under some technical assumptions. This would justify that the assumption is reasonable and allows to avoid some technicalities.

5. I think the paper can comment on the advantages of the Gauss-Seidel Anc-VI and why this extension is considered. Section 6 does not discuss the motivation for considering such an algorithm. Is it for the possibility to perform asynchronous updates via its coordinate-wise update rule?

**Minor:**

- I find the terminology ‘linear operator’ for the Bellman (consistency) operator a little confusing even if Puterman 2005 (p. 144, Eq. (6.1.7)) uses for e.g. the terminology ‘linear transformation’. A linear operator T would satisfy T(0) = 0 according to the standard mathematical definition of a linear operator, which is not the case for the Bellman consistency operator. Maybe affine would be more appropriate.

- l. 137-141: I would say the result for the Bellman ‘consistency’ operator can also be relevant for policy evaluation which is useful as a subroutine for several algorithms in RL beyond value iteration.

- As a suggestion, the paper could add a figure for the hard MDP instance to ease the reading (see for e.g., Figure 1 in [37]). It also seems that the hard MDP instance is the same as the one in [37], it may be worth it to mention this in that case.

- There is sometimes some redundancy in the proofs. For instance the Hahnn-Banach argument is used identically 2 times in l. 568, l. 590. It is again used for Q functions in l. 578 and l.  600. This also happens for some inductions throughout the proofs which are very similar.

- Lemma 15 in appendix: in the proof, ‘by definition W, there exist W’, I guess you mean there exists $k$.

- l. 513 and later: $\lim …  \to $,  I guess $\to$ can be replaced by $=$ here.

- Several grammatical articles are missing in writing, especially in the appendix: l. 225 (a complexity …), l. 507 (a nonexpansive), 510, 515, 520, 664, 700, 759, 760, 764 and several other places.

**Typos:**
l. 111, 113: ‘nonexpensive’;
l. 116: ‘operator’.


**Questions:**

Besides the comments above, please find some questions below for clarifications:

1. l. 149 - 151: is the comparison of the rates with VI straightforward from the rates or does it require some algebraic manipulation? It does not seem to be immediate from the expression, please provide more details to clarify in the appendix if needed.

2. It is mentioned in the paper that the anchoring mechanism is ‘distinct from Nesterov’s acceleration’. The latter mechanism has inspired the work [37] which requires for instance some reversibility conditions as discussed in related works. A recent work (Tran-Dinh 2022) draws connections between Nesterov’s accelerated methods and Halpern fixed point iterations. Could you comment on this?

Quoc Tran-Dinh, 2022. The Connection Between Nesterov’s Accelerated Methods and Halpern Fixed-Point Iterations.
https://arxiv.org/pdf/2203.04869.pdf

3. In the abstract, it is mentioned that ‘the optimal rate for the VI setup was not known’. Do you mean for the Bellman error (as the rest of the abstract and the paper state results for the Bellman error)?  Please precise in that case, I found this a bit confusing at first read.

4. How do you come up with the specific way you set the parameter $\beta_t$ in l. 122? It seems to match the usual $\beta_t = 1/(t+1)$ used for anchoring/Halpern iterations in the case where $\gamma = 1$. Could you provide more intuition about this?

5. Results on the Bellman error can also be translated to the optimality measure $\|\|U^k - U^{\star}\|\|$
 using the triangle inequality at least for $0<\gamma<1$
to obtain $\|\|U^k - U^{\star}\|\| \leq \frac{1}{1-\gamma} \|\|U^k - T U^k\|\|$. How would this result compare to VI? Is it meaningful to include a comment along these lines for comprehensiveness given that you discuss translating the result for VI to control the Bellman error? I understand though that the anchoring mechanism is better suited to control the Bellman error.

6. l. 290 - 292: How would the GS update rule would even be defined in infinite dimensions even before talking about the Hahn-Banach theorem for the analysis? Why Hahn-Banach theorem would not be applicable?

7. Minor comment, l. 240: for the upper bound, it seems that one can obtain a constant equal to 2 using the following derivations:
$$\frac{(\gamma^{-1} - \gamma)(1+\gamma - \gamma^{k+1})}{(\gamma^{k+1})^{-1} - \gamma^{k+1}}
= \frac{\gamma^k (1-\gamma) (1+\gamma - \gamma^{k+1}) }{1-\gamma^{2(k+1)}}
= \gamma^k \frac{1}{\sum_{i=0}^{2k+1} \gamma^i} (1+\gamma - \gamma^{k+1})
\leq \frac{2 \gamma^k}{\sum_{i=0}^k \gamma^i}.$$

**Proof related questions for clarifications:**

 8. About the Hahn-Banach argument (for infinite state action space): Could you please provide clarifications regarding the following questions?
- Lemma 8, 9 in the appendix: What are $U$, $\tilde{U}$, $\bar{U}$ in the statements? If these are arbitrary (as they are instantiated later in the proofs of Lemma 10,11 for example), please precise the statement of the lemmas for clarity.

- The definition of the operator $\mathcal{P}$ (l. 600 for e.g., and others) is not very clear to me. Do you define it only for multiples of $\bar{Q}$ where $\bar{Q}$ is defined in the lemma statement or do you define it for every $Q$? Why do you introduce $c$ in the definition? I guess this is because you need homogeneity to invoke the Hahn-Banach theorem but I would expect that this would be verified once you define the sublinear function for the application of the theorem. Also, it is not clear to me why is the $\mathcal{P}$ as defined a linear functional on M (because of the inf) with $\|\|\mathcal{P}\|\|$ = 1, which norm is this? Is it the operator norm (please define it in this case)? The notation $\bar{Q}$ is a little confusing. If $\mathcal{P}$ is defined by the inf (without c) for any $Q$, then you can show sublinearity (or rather ‘superlinearity’ with the inf) and homogeneity. The restriction to the span of $\bar{Q}$ would be a linear functional (as you mention it) and then you can conclude I guess. If this is the reasoning you conduct, please clarify. I suggest to state the Hahn-Banach theorem and its application or at least to clarify what is the sublinear function $p$ used and what is the linear functional dominated by $p$, especially the use of the notation $\bar{Q}$ for defining $\mathcal{P}$.

- Minor, l. 568, l. 590: ‘if action space is infinite’? Is it rather state space here? Same comment for other occurrences.

9. Case $\gamma =1$: l. 672, 673 in the proofs in appendix, it is stated that the $O(1/(k+1))$ rate is obtained after taking the infinity norm in Lemmas 4 and 10. Does the $O(1/(k+1))$ rate just follow from using $\beta_k = 1/(k+1)$ or is there any particular upper bounding manipulation? I think more clarifications would be useful for the reader.

**Limitations:**

The paper does list the fact that the analysis of VI is not sufficient to understand modern deep RL (in appendix H in the end of the paper) and comments on the potential of future work in the conclusion.

---

> ### Author Rebuttal · Authors · 2023-08-10
>
> We appreciate the reviewer for the highly detailed and constructive feedback.
>
> Weakness
>
> (i) As an analogy in the optimization literature, the recent discovery of OGM [1] and OGM-G [2] demonstrate that considering a different performance measure leads to a different optimal algorithm. In this setting, we show that Anc-VI is an optimal algorithm when we consider optimally reducing the Bellman error, but we are not claiming that the Bellman error is "better" measure compared to the usual distance to the optimal value function. Our point is that if we consider the Bellman error (which is also natural) Anc-VI is an optimal algorithm. However, one argument for the Bellman error is that we can get a meaningful rate and a point convergence result when $\gamma\approx1$ or $\gamma=1$ only when we consider the Bellman error. We will incorporate this discussion in the revision. We will comment on this point in our revision.
>
> (ii) Thank you for your comment. We will define a near-optimal policy and explain how it can be derived from Anc-VI in revision.
>
> (iii) (+ Question $5$)
> Yes, we can. If we translate Bellman error of Anc-VI to the distance to optimal value function using $\|U^k-U^{\star}\| \le \frac{1}{1-\gamma}\|TU^k-U^k\|$, Anc-VI shows same convergence rate $O(\gamma^k)$ with VI but constant factor slower by
> \begin{align*}
>     \gamma^{k}(1+\gamma)\frac{1+2\gamma-\gamma^{k+1}}{1-\gamma^{2k+2}}   \ge \gamma^{k}(1+\gamma).
> \end{align*}
> We will add this argument in the revision.
>
> (iv) We could not find prior works about technical condition of well definiteness of value function when $\gamma=1$, But we could think of the following MDP instance: If an undiscounted MDP has bounded reward and the probability of transitioning to a terminal state is larger than some fixed positive constant (for all current state and action) this undiscounted MDP has finite value function since $\sum^{\infty}_{i=0}nx^n$ converges for $|x|<1$. This is one condition for well definiteness of value function, and pursuing a systematic study of such a condition may be an interesting direction.
>
> (v) We were careful in commenting on what we did not yet prove, but, yes, we think Gauss-seidel Anc-VI is a stepping stone to analyzing asynchronous coordinate update version of Anc-VI, and we plan to study this direction in our future work. We will comment on this in revision.
>
> Minor and typos
>
> (i) We agree with your point on the definition of a linear operator. Although we followed the convention of [3], we will update the terminology in our revision.
>
> (ii) Thank you for this point on the usefulness of the results for the policy evaluation setup. Our intented point of line 137-141 was that the Bellman optimality setting is more difficult due to the nonlinearity and that we are able to provide an acceleration in both the consistency and optimality setups while prior works don't. We will adjust our wording.
>
> (iii) The worst-case MDP in [4] and worst-case MDP in our paper are different primarily in their rewards: $r(s_i, a_1)=\mathbf{1}_{\{i=1\}}$
>
> and $r(s_i, a_1)=\mathbf{1}_{\{i=2\}}$, respectively. Less significantly, our worst-case MDP has one additional state, but the transition probabilities are essentially the same. Both MDPs have only one action. We will clarify this point, and we will add a figure to illustrate the hard MDP instance.
>
> (iv) As the Hahn-Banach argument is abstract and delicate, we tried to make the argument precise and explicit for every case. However, we will revise the proofs to reduce the redundancy.
>
> (v, vi, vii, Typos)  Thank you for the detailed corrections. We will correct the errors in our revision.
>
> Questions
>
> (i) Comparison in line 149-151 comes from direct calculations, but we agree with that algebraic manipulation is not immediate. We clarify this in revision.
>
> (ii) Thank you for pointing out this reference. This work does reveal a connection between anchor acceleration and Nesterov's acceleration, but we claim that the two mechanisms are different in the following sense. First, following reformulation scheme of Theorem $3.2$ in this reference, A-VI in [4], a Nesterov-type VI, can not be reformulated to Anchoring type VI since reformulation only covers restricted Nesterov type algorithm. Second, there is line of research for continuous-time models of acceleration, and [5] and [6] showed that anchor acceleration and Nesterov's acceleration have distinct ODE models. It is our view that Nesterov's acceleration and anchor acceleration have a connection but are substantively distinct, but studying the connections and differences between these two acceleration types is certainly an interesting direction.
>
> (iii) Sorry for the confusion. Yes, we mean the optimal rate in terms of Bellman error.  We will clarify this in revision.
>
> (iv) [7] studied anchor acceleration for $\gamma$-contractive non linear operator respect to $l_2$ norm in Hilbert space. Inspired by this paper, we tested several candidates of $\beta_n$, $\frac{1}{\sum^n_{i=0} \gamma^{ki}}$ for $k=1,2,3$,
> and we found that the choice $k=2$ gives analytically simple and accelerated rate as in Theorem $1$ and $2$.
>
> (vi) Sorry for the confusion. We can consider block GS update by dividing action space into finite disjoint sets, and `is not applicable' might not be proper expression. We briefly mention that even if we apply Hahn-Banach theorem on block GS update setting, it does not lead us to valid convergence result since argument of Lemma $8$ is not valid for product of multiple operators $\mathcal{P}_n \dots \mathcal{P}_1$ which appears in proof of Lemma $19$. We will clarify this in revision.
>
> (vii) We believe there is little mistake on your calculation since
> \begin{align*}
>     \frac{(\gamma^{-1}-\gamma)(1+2\gamma-\gamma^{k+1})}{(\gamma^{k+1})^{-1}-\gamma^{k+1}}
>     =\gamma^{k}\frac{(1-\gamma^2)(1+2\gamma-\gamma^{k+1})}{1-\gamma^{2k+2}}.
> \end{align*}
>
> **Rebuttal continues in common response**

---

> > ### Comment · Reviewer_GSn8 · 2023-08-15
> > **post rebuttal**
> >
> > I thank the authors for their rebuttal which addressed my questions in details, I maintain my positive score.
> >
> > Regarding Bellman error vs distance to optimal value, I wanted to stress that while in optimization first order stationary is a natural quantity to look at in the non convex setting since distance to optimum is not even necessarily well-defined, in RL (and also in the present paper setting), actually both Bellman error and distance to optimal value are meaningful. This is the reason why I am somehow questioning the importance of finding an optimal algorithm for the Bellman error rather than for the distance to optimal value for which value iteration is known to be optimal. I acknowledge that the discussion about $\gamma = 1$ provides some motivation for the Bellman error though (if the distance to optimal value function is not meaningful anymore).

---

### Official Review · Reviewer_8tEg · 2023-07-07

**Soundness:** 3 good
**Presentation:** 3 good
**Contribution:** 2 fair
**Rating:** 6
**Confidence:** 3

**Summary:**

The paper introduces an accelerated version of the Value Iteration (VI) algorithm, called Anc-VI, based on the anchoring mechanism.  The proposed method achieves faster reduction in the Bellman error compared to standard VI even when the discounting factor is close to 1. Meanwhile, a complexity lower bound is also provided which match with the upper bound up to a constant 4. Meanwhile, this work also shows the benefits of anchoring mechanism in approximate VI and Gauss-Seidel VI.

**Strengths:**

1. This work proposed the acceleration of VI by leveraging the anchoring mechanism
2. The proposed Anc-VI has provable faster convergence rate comparing with classic VI
3. This work provide the acceleration rate for both the bellman optimality operator and consistency operator

**Weaknesses:**

1. The performance of anchoring mechanism has reliance on the starting point $U_0$ and discounting factor. In order to have the fast acceleration, we need to set $\gamma\approx1$. However, one of the reason that we set discounting factor $\gamma < 1$ is for faster computing in practice.  The author need to clarify whether the anchoring mechanism still can benefit the learning acceleration in the practical case, e.g., in Theorem 1, when $\gamma \to 0$, how does the upper bound show the acceleration. Same question apply to Theorem 2
2. In the Apx-Anc-VI, when $U_0$ is far from optimal, according to Theorem 6, the first few iteration does not necessary results in policy improvement (e.g., by setting $k=1$, $\gamma=1$). Thus, what does Theorem 6 can tell on the benefits of using Anchoring mechanism?

**Questions:**

1.  If it is possible to derive any practical algorithm based on the theoretical findings in this work?
2. What is the computing complexity of the proposed method comparing with VI?
3. What is the guidance on choosing $\gamma$ in practice in order to achieve faster convergence, e.g., let $\gamma \approx 1$?

**Limitations:**

The authors addressed the limitations in the Appendix.

---

> ### Author Rebuttal · Authors · 2023-08-10
>
> We thank the reviewer for the thoughtful comments.
>
> Weakness
>
> (i) (+Question (iii)) Although in some practical setups, discount factor $\gamma$ could be chosen freely, we assumed that environment and MDP and $\gamma$ are given constants.
>
> If $1/2 \le \gamma<1$, Anc-VI exhibits a provably faster convergence rate (First rates of Theorems 1 and 2) than the standard rate of VI, since
> \begin{align*}
>     \frac{(\gamma^{-1}-\gamma)(1+2\gamma-\gamma^{k+1})}{(\gamma^{k+1})^{-1}-\gamma^{k+1}}
>     =\gamma^{k}\frac{(1-\gamma^2)(1+2\gamma-\gamma^{k+1})}{1-\gamma^{2k+2}}
>     \le \gamma^{k}(1+\gamma).
> \end{align*}
> If $0<\gamma<1/2$, these rates don't guarantee acceleration, but if $TU_0 \le U_0$ or $ U_0 \le TU_0$, the second rates of Theorems 1 and 2 are faster than the standard rate of VI for all $0 < \gamma <1$. (Both rates are decreasing functions of $\gamma$.)
>
> (ii) For Apx-Anc-VI and Apx-VI, we didn't include the case $\gamma=1$ since the error diverges as the iteration number increases. If $0<\gamma<1/2$, Apx-Anc-VI exhibits a provably faster convergence rate than the rate of Apx-VI by same argument in (i).
>
> Questions.
>
> (i) As VI serves as the foundational basis of practical RL algorithms such as fitted value iteration and temporal difference learning, we expect that Anc-VI will give insight for designing new practical algorithms or improving existing ones by incorporating the anchoring mechanism. This is certainly an interesting direction that we plan to pursue in our future work.
>
> (ii) Computing complexity of Anc-VI and VI *per iteration* are basically the same; the operation of adding an anchor term is a vector-vector operator, and is therefore negligible compared the evaluation of the Bellman operator, which often involves a matrix-vector operation.

---

> > ### Comment · Reviewer_8tEg · 2023-08-10
> > **Ack**
> >
> > I thank the authors' effort on the rebuttal.
> >
> > I think the discussion on the discounting factor when introducing the theorems can be helpful for the general audiences to be more aware of the limitations of the theoretical results.
> >
> > I will keep my original score.

---

### Author Rebuttal · Authors · 2023-08-10

# Common Response

First of all, we thank the reviewers for their constructive and detailed feedback. We were excited to see that all the reviewers found our work valuable. Indeed, as reviewer e9mJ and tQc7 mentioned, acceleration of Anc-VI is guaranteed by ''strong theorem'' and ''leads to some surprising results'', and we expect the anchoring mechanism of Anc-VI to be applicable to more practical setups. Specifically, Reviewer GSn8 and e9mJ mentioned using Anc-VI in model-free setups and with asynchronous coordinates. We believe these are all interesting future directions.

$ $

$ $

$ $

$ $


# Rebuttal of reviewer GSn8 continued

Hahn-Banach argument

(i) $U, \\tilde{U}, \\bar{U}$ are arbitrary function satisfying condition of Lemmas. We will clarify this in revision.

(ii) $\\bar{Q}$ is defined in the Lemma and we introduce $c$ for homogeneity of $\\mathcal{P}$ to invoke the Hahn-Banach theorem as you pointed out. Then, $\\mathcal{P}$ is linear functional in $M$ where $M$ is linear space spanned by $\\bar{Q}$ with $\\|\\cdot\\|_{\\infty}$

-norm.  About norm of $\\mathcal{P}$, we apologize for unclear notation and typo. Norm should be operator norm, and in line 601, '$\\|\\mathcal{P}\\| = 1$' should be modified to '$\\|\\mathcal{P}\\| \\le 1$'. This is true since $\\frac{ |c \\inf_{(s',a') \\in \\mathcal{S} \\times \\mathcal{A}}  \\bar{Q}(s',a')|}{\\|c\\bar{Q}\\|_{\\infty}} \\le 1 $. We will clarify this and reflect your suggestion in our revision.

(iii) If action space is finite, greedy policy satisfying $T^{\\pi}V=T^{\\star}V$ is well defined and it directly leads to Lemma $8$ and $9$ as we showed in proofs. But if action space is infinite, we can not guarantee existence of greedy policy since maximizer may not exist in action space. In this case, we solved this issue by Hahn-Banach argument.

(iv) If we plug $\\beta_k=\\frac{1}{k+1}$ in Lemma $4$ and $10$, we get $O(1/k+1)$ rate by simple calculation. We will clarify this in our revision.

[1] D. Kim and J. A. Fessler. Optimized first-order methods for smooth convex minimization. Mathematical Programming, 159(1–2):81–107, 2016.

[2] D. Kim and J. A. Fessler. Optimizing the efficiency of first-order methods for decreasing the gradient of smooth convex functions. Journal of Optimization Theory and Applications, 188(1):192–219, 2021

[3] M. L. Puterman. Markov Decision Processes: Discrete Stochastic Dynamic Programming. John Wiley and Sons, 1994

[4] V. Goyal and J. Grand-Clément. A first-order approach to accelerated value iteration. Operations Research, 71(2):517–535, 2022.

[5] Su, W., Boyd, S., and Candès, E. J.  A differential equation for modeling Nesterov’s accelerated gradient method: Theory and insights. Journal of Machine Learning Research, 17(153):1–43, 2016.

[6] J. J. Suh, J. Park, and E. K. Ryu. Continuous-time Analysis of Anchor Acceleration arXiv:2304.00771, 2023.

[7] J. Park and E. K. Ryu. Exact optimal accelerated complexity for fixed-point iterations. International Conference on Machine Learning, 2022.

---

### Decision · Program_Chairs · 2023-09-21

**Decision:**

Accept (poster)

**Comment:**

This paper applied the anchoring mechanism rooted in fixed point theory to Value Iteration and showed improved rates in terms of the Bellman errors. While the results are perhaps not too surprising from optimization perspectives, all reviewers find the contributions solid and valuable given the importance of value iteration in RL.

Therefore, acceptance is recommended, though I strongly suggest the authors to explicitly discuss the key limitations of this work as identified by reviewers in the revision : (1) lack of empirical justification of the advantages of anchored VI , (2) worse bounds in terms of distance to optimal values, (3) challenges for stochastic extensions, etc. Please also be rigorous in terms of the notion of rate, optimality of algorithms.